# McIdas localizes to centrioles and controls centriole numbers through PLK4-dependent phosphorylation

Marina Arbi [1✉], Margarita Skamnelou[1], Lydia Koufoudaki[1], Vasiliki Bakali [1], Spyridoula Bournaka[1], Sihem Zitouni [2], Stavroula Tsaridou[1], Ozge Karayel [3], Catherine G Vasilopoulou[3], Aikaterini C Tsika [4], Nikolaos N Giakoumakis [5], Ourania Preza[1], Georgios A Spyroulias [4], Matthias Mann [3,6], Mónica Bettencourt-Dias [2], Stavros Taraviras [7] & Zoi Lygerou [1]

## Abstract

The centriole duplication cycle must be tightly controlled and coordinated with the chromosome cycle. Aberrations in centriole biogenesis can cause developmental disorders, ciliopathies and cancer, yet the molecular determinants controlling centriole numbers and the link between the two cycles remain poorly characterized. Here, we demonstrate that McIdas, previously implicated in cell cycle regulation and multiciliogenesis, plays a critical role in maintaining proper centriole numbers. McIdas localizes to centrioles, where it exhibits dynamic localization throughout the cell cycle, dependent upon a nuclear export signal (NES) in its coiled-coil domain. Overexpression of McIdas induces centriole overduplication, whereas its depletion perturbs daughter centriole biogenesis and SAS6 recruitment. An NES mutant of McIdas that fails to localize to centrioles does not induce centriole amplification. Moreover, McIdas depletion reduces PLK4-induced centriole amplification. McIdas interacts with and is phosphorylated by PLK4, which is critical for its role in centriole number control. Overall, our results demonstrate that in addition to its known nuclear localization, McIdas also localizes to centrioles, affecting centriole duplication. This novel, direct role of McIdas in centriole duplication connects its functions in cell cycle regulation and multiciliogenesis.

**Keywords** Centriole Amplification; Centrosome; Geminin Superfamily; Multicilin; Polo-like kinase 4
**Subject Categories** Cell Adhesion, Polarity & Cytoskeleton; Cell Cycle; Post-translational Modifications & Proteolysis

## Introduction

Centrioles are highly conserved, microtubule-based cylindrical organelles that contribute to the biogenesis of centrosomes and primary or motile cilia in cells (Jana, 2021; Paz and Luders, 2018; Vasquez-Limeta and Loncarek, 2021; Wu and Akhmanova, 2017). Two orthogonally arranged centrioles, one mother and one daughter, surrounded by the pericentriolar material (PCM), are the major components of centrosomes. Centrosomes play critical roles in cell division, cell migration, polarization and signaling. In many non-dividing cells, the centrosome migrates to the apical cell membrane, where the mature mother centriole docks and acting as basal body nucleates a primary, sensory immotile cilium (Arslanhan et al, 2020; Mill et al, 2023). However, in terminally differentiated multiciliated cells centrioles are massively amplified and act as templates for the generation of multiple motile cilia (Meunier and Azimzadeh, 2016; Spassky and Meunier, 2017).

Centrioles duplicate once and only once during the cell cycle, in S phase. Licensing, duplication and segregation are key steps that control the centriole duplication cycle (Firat-Karalar and Stearns, 2014; Fu et al, 2015; Gomes Pereira et al, 2021). Upon exit from M phase and during G1 phase, the two centrioles are disengaged, licensing a new round of centriole duplication. During S phase, one new centriole, termed a procentriole, is orthogonally generated next to each pre-existing centriole. The newly generated centrioles are elongated during G2 phase, while mother centrioles mature. During mitosis, the two pairs of centrioles are separated to form the poles of the mitotic spindle, ensuring accurate sister chromatid segregation to the two daughter cells.

PLK4 (Polo-like kinase 4), STIL (SCL/TAL1 interrupting locus protein) and SAS6 (Spindle assembly abnormal protein 6 homolog) are critical regulators for the initiation of centriole assembly (Arquint and Nigg, 2016). PLK4 is first recruited as a ring-like structure at the proximal region of mother centrioles (Bettencourt-Dias et al, 2005; Habedanck et al, 2005; Kim et al, 2013; Ohta et al, 2014). PLK4 binds and phosphorylates STIL, resulting in the

[1]Department of General Biology, School of Medicine, University of Patras, Basic Medical Sciences Building, 1 Asklepiou Str., University Campus, 26504 Rio Patras, Greece. [2]Gulbenkian Institute for Molecular Medicine, Rua da Quinta Grande 6, Oeiras 2780-156, Portugal. [3]Department of Proteomics and Signal Transduction, Max Planck Institute of Biochemistry, 82152 Martinsried, Germany. [4]Department of Pharmacy, School of Health Sciences, University of Patras, New Pharmacy Building, 1 Asklepiou Str., University Campus, 26504 Rio Patras, Greece. [5]Advanced Digital Microscopy Facility, Institute for Research in Biomedicine, 10 Baldiri Reixac, 08028 Barcelona, Spain. [6]NNF Center for Protein Research, 2200 Copenhagen, Denmark. [7]Department of Physiology, School of Medicine, University of Patras, Basic Medical Sciences Building, 1 Asklepiou Str., University Campus, 26504 Rio Patras, Greece. ✉E-mail: marmpi@upatras.gr; marmpi@ed.ac.uk

subsequent loading of SAS6 (Kratz et al, 2015; Moyer et al, 2015; Ohta et al, 2014). This leads to the assembly of the cartwheel, a macromolecular hub nucleating the formation of the ninefold symmetrical microtubule triplets of the new centriole (Gonczy, 2012).

The expression levels and the interactions between PLK4, STIL and SAS6 must be strictly regulated with their overexpression leading to the formation of multiple daughter centrioles around a mother centriole in rosette-type structures. PLK4 is regulated through multiple post-transcriptional and post-translational mechanisms (Cunha-Ferreira et al, 2013; Cunha-Ferreira et al, 2009; Guderian et al, 2010; Holland et al, 2012; Holland et al, 2010; Lopes et al, 2015; Nakamura et al, 2013; Phan et al, 2022; Rogers et al, 2009; Ryniawec and Rogers, 2022), whereas STIL and SAS6 are degraded in late M and early G1 phase by the anaphase-promoting complex/cyclosome associated with Cdh1 (APC/C$^{Cdh1}$) (Arquint and Nigg, 2014; Arquint et al, 2012; Strnad et al, 2007).

The centriole cycle must be closely coordinated with the chromosome cycle, as defects in this coordination will lead to improper chromosome segregation, causing genomic instability and tumorigenesis (Gonczy, 2015; Nigg and Holland, 2018; Raff and Basto, 2017). Earlier studies showed that cyclin-dependent kinases are required for centrosome duplication (Hinchcliffe et al, 1999; Lacey et al, 1999; Matsumoto et al, 1999; Meraldi et al, 1999; Zitouni et al, 2016). Moreover, replication proteins such as MCM5, Orc1, Geminin and Cdc6 are recruited at centrosomes through interactions with S phase cyclins and are implicated in centriole duplication (Ferguson and Maller, 2008; Ferguson et al, 2010; Hemerly et al, 2009; Tachibana et al, 2005; Xu et al, 2017). However, how the centriole and chromosome cycles are inter-linked remains poorly characterized.

Recent studies proposed McIdas as a protein with roles both in cell cycle control and in differentiation decisions in cells (Arbi et al, 2018). McIdas is a nuclear, vertebrate-specific coiled-coil protein (Pefani et al, 2011), phylogenetically related to the cell cycle regulators Geminin (McGarry and Kirschner, 1998) and GemC1 (Balestrini et al, 2010; Caillat et al, 2015). McIdas binds to Geminin and inhibits its association with Cdt1, affecting DNA replication licensing (Caillat et al, 2013; Pefani et al, 2011). Its protein levels drop during anaphase of mitosis, and it has been suggested as an APC/C target. McIdas loss affects normal cell cycle progression and leads to the accumulation of cells in S phase, whereas its overexpression causes abnormal multipolar spindles (Pefani et al, 2011). In addition, several studies identified both McIdas and GemC1 as early regulators of multiciliogenesis (Arbi et al, 2016; Boon et al, 2014; Kyrousi et al, 2015; Loukas et al, 2021; Stubbs et al, 2012; Terre et al, 2016; Zhou et al, 2015). During multiciliogenesis, McIdas acts through the formation of complexes with transcription factors, including E2F4/5 and DP1, switching on a gene expression program essential for centriole amplification and cilia formation (Lewis et al, 2023; Ma et al, 2014).

Here, we show that McIdas is a novel regulator of the centriole duplication cycle in cycling cells. McIdas, via a nuclear export signal in its coiled-coil domain, localizes to centrioles in a cell cycle-specific manner. McIdas interacts with and is phosphorylated by PLK4 and is required for the maintenance of correct centriole numbers. Our data suggest that the roles of McIdas in the chromosome cycle and multiciliogenesis may be interconnected through this novel, direct role in centriole duplication.

# Results

## McIdas localizes to centrosomes

McIdas was initially characterized as a nuclear protein that binds to Geminin and inhibits its association with Cdt1 (Pefani et al, 2011). Overexpression of McIdas leads to abnormal mitotic spindle formation (multipolar spindles), indicating that it may affect centrosome numbers in cycling cells. Moreover, McIdas was identified as an important regulator of centriole and cilia formation in post-mitotic multiciliated cells (Stubbs et al, 2012). It exhibits a highly specific expression pattern in ciliated epithelia, but it is also expressed in proliferating cells, albeit at lower levels (Arbi et al, 2016). Using a previously validated antibody (Pefani et al, 2011) in human osteosarcoma U2OS cells and methanol fixation, which preserves centriolar structures, we observed that McIdas was detected at centrosomes throughout the cell cycle (Fig. 1A). The known, nuclear localization of McIdas was not detectable under methanol fixation, unless McIdas was over-expressed (Fig. EV1C,D). We examined the localization of McIdas at centrosomes throughout the cell cycle by following EdU incorporation, DNA staining and the distal centriolar marker Centrin to monitor different cell cycle phases. G1 cells can be distinguished as EdU-negative cells with two Centrin dots which are closer together in early G1 phase and move further apart as G1 phase progresses. McIdas appeared as a single dot in early G1 cells, localizing adjacent to the two Centrin foci (Fig. 1A), consistent with a centrosomal localization. Later in G1, McIdas was detected as two foci adjacent to the Centrin-marked centrioles (Fig. EV1A,B). As cells progressed through S phase (EdU-positive cells), G2 phase (EdU-negative cells with four Centrin foci) and early mitosis (Fig. 1B), McIdas was detected at centrosomes with increased intensity, while its levels dropped during telophase (Fig. 1A,B). This is consistent with a previous study showing that total McIdas protein levels drop in anaphase (Pefani et al, 2011).

A GFP-tagged McIdas protein was also detected at centrosomes in different cell lines, either stably expressed in HeLa cells (Fig. 1C) or as an exogenously overexpressed protein (Fig. EV1C). Accumulation of Centrin was evident upon GFP-McIdas overexpression in U2OS or hTERT RPE-1 cells (Fig. EV1C, marked by an asterisk). Notably, the addition of the proteasome inhibitor MG132 in HeLa cells stably expressing either GFP-tagged McIdas or GFP alone as a control, caused McIdas accumulation at centrosomes in mitosis, consistent with a proteasome-dependent degradation during mitosis (Fig. EV1D). The specificity of McIdas localization at centrosomes was confirmed using two different siRNA oligos (Fig. EV2). Both siRNA oligos efficiently reduced McIdas mRNA (Fig. EV2A,F) and protein (Fig. EV2B–E,G–J) levels at centrosomes and were therefore used interchangeably in the following experiments. The above data show that McIdas is localized to centrosomes and its expression is regulated during the cell cycle. Interestingly, McIdas temporal expression pattern is reminiscent of the expression pattern followed by known centriole regulators, such as STIL and SAS6 (Arquint and Nigg, 2014; Arquint et al, 2012; Strnad et al, 2007).

## McIdas localizes to the central region of the centriole and exhibits a differential localization pattern during the cell cycle

Given that McIdas can be detected at the centrosome as only one dot during part of the G1 phase, we next examined in which one of

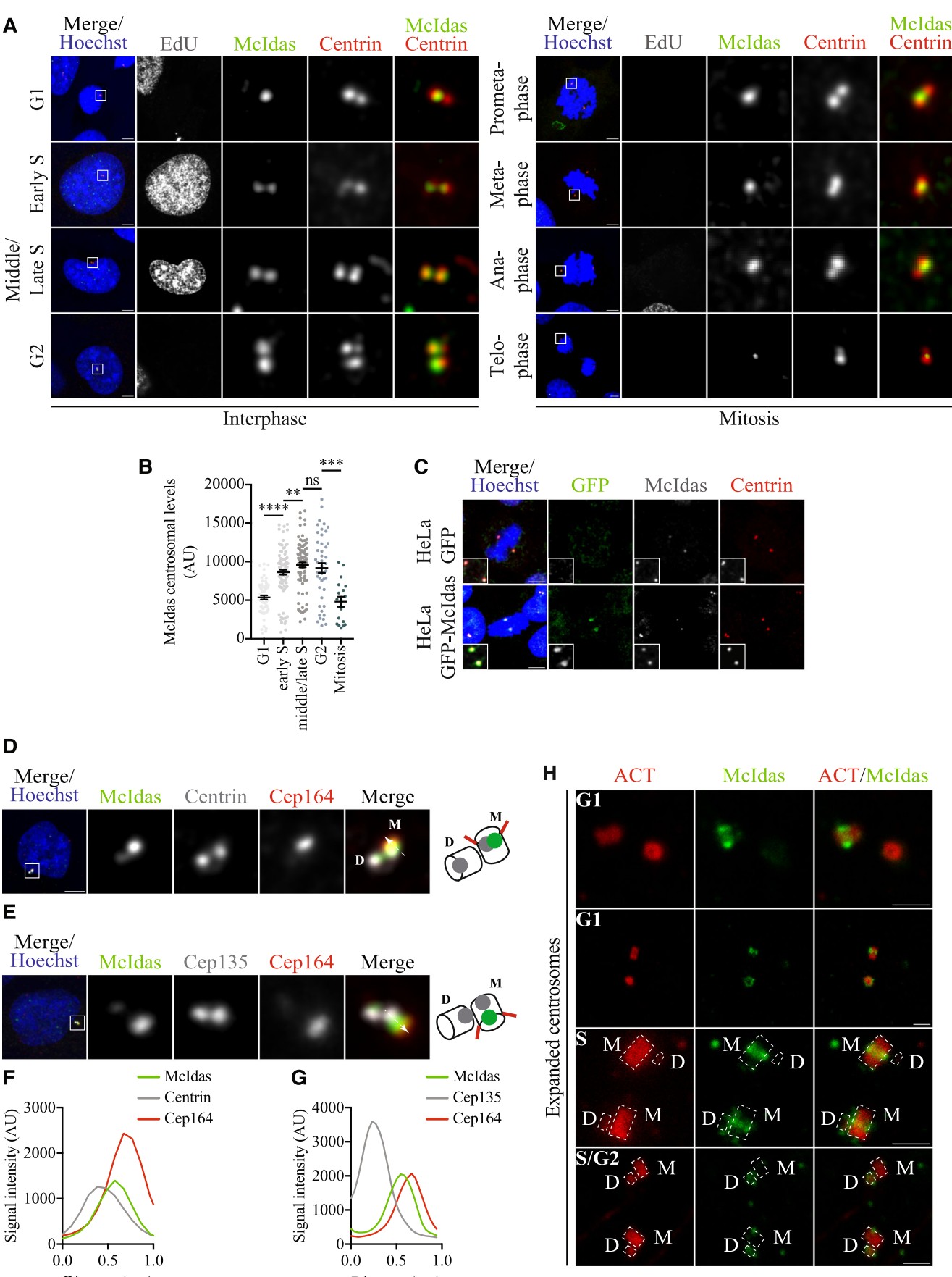

◄ **Figure 1. McIdas localizes to centrosomes in a cell cycle specific manner.**

(A) U2OS cells were immunostained with antibodies against McIdas and Centrin (a distal lumen centriole marker). EdU (marks S phase cells) and Hoechst (for DNA staining) were used to monitor the different cell cycle stages. (B) Quantification of McIdas centrosomal levels across the different cell cycle stages shown in (A). Data are from three independent experiments for each cell cycle stage. Error bars indicate ± SEM. *P*-values were calculated by the nonparametric two-tailed Mann-Whitney test: **$P$ <0.01, ***$P$ <0.001, ****$P$ <0.0001 and ns, not significant ($P$ values: G1-early S, $P < 0.0001$; early S-middle/late S, $P = 0.0042$, middle/late S-G2, $P = 0.4862$ and G2-Mitosis, $P = 0.0001$). (C) HeLa cells were generated to stably express either GFP-tagged McIdas or GFP alone as a control. Cells were pre-extracted and immunostained with antibodies against GFP to mark the transfected cells, endogenous McIdas, and Centrin. (D, E) hTERT RPE-1 cells were stained for McIdas and centriole markers, including Centrin, Cep135 (proximal centriole marker), and Cep164 (distal appendage centriole marker). Schematic representations indicate McIdas localization with respect to the centriole markers used, consistent with this analysis. (F, G) Corresponding fluorescence intensity plots of the centrosomal signals shown in (D) and (E) for McIdas, Cep164 and Centrin or Cep135, respectively. The position and direction of the line used to generate the intensity profiles as a function of distance are indicated at (D) and (E) by the white line and the arrow, respectively. (H) ExM images from hTERT RPE-1 cells at different cell cycle phases stained with antibodies against acetylated α-tubulin (ACT, centriolar microtubule marker) and McIdas. Centriole numbers and acetylated α-tubulin staining were used to identify the different cell cycle phases. Mother and newly synthesized centrioles are outlined with dashed lines. White boxes indicate regions shown as higher-magnification images. DNA was stained with Hoechst. Scale bars, 5 μm (A, C, D and E) and 2 μm (H). ExM ~ 4x. ACT acetylated α-tubulin, M mother centriole, D daughter centriole, AU arbitrary units. Source data are available online for this figure.

the two centrioles it can be found. To that end, we performed an immunofluorescence analysis in hTERT RPE-1 cells, stained for McIdas along with Cep164, which marks the distal appendages of mother centriole. As shown in Fig. 1D, when McIdas was detected as one dot, this coincided with the Cep164-positive centriole, consistent with McIdas localization to the mother centriole in early G1 phase. To study McIdas sub-centrosomal localization, further analysis was performed by comparing McIdas localization with additional centriole markers, such as Centrin (distal centriole lumen marker) and Cep135 (proximal marker) (Fig. 1D,E). McIdas-, Cep135- and Cep164-signal intensities were plotted relative to distance and revealed that McIdas localization at the centrosome was proximal to that of Centrin and Cep164 (Fig. 1F) and distal to Cep135 (Fig. 1G). McIdas is partially co-localized with Cep164 at the distal part of centrioles.

To gain better insight into McIdas centriolar localization we performed expansion microscopy (ExM), marking centrioles with acetylated α-tubulin (Fig. 1H). In line with our previous analysis, McIdas is localized to the central part of the mother centriole and exhibits one- or two-dot staining in G1-phase cells. The observed localization is consistent with McIdas distribution on the outer surface of centrioles. During late S and G2 phases of the cell cycle, higher resolution images demonstrated a gradual enrichment of McIdas at newly synthesized centrioles, with decreased levels on mother centrioles. Consistently, McIdas is detected prominently at the newly synthesized centrioles during early mitosis (Fig. EV1E–G).

We therefore conclude that McIdas localizes to the central region of centrioles, displaying redistribution from mother to daughter centrioles as the cell cycle progresses.

## McIdas regulates daughter centriole biogenesis

Given the centrosomal localization of McIdas, we next tested whether McIdas has a direct role in centriole biogenesis. McIdas was initially characterized as a cell cycle regulator (Pefani et al, 2011). To study its direct involvement in centriole biogenesis, we performed a centrosome duplication assay in U2OS cells, where the centrosome cycle can be experimentally uncoupled from the DNA replication cycle by inhibiting DNA synthesis (Balczon et al, 1995; Meraldi et al, 1999). This uncoupling ensures that effects observed on the centrosome cycle are not secondary to DNA replication cycle effects. U2OS cells were treated with high concentration of

hydroxyurea (HU), which blocks DNA replication, arrests cells in S phase and results in multiple rounds of centriole duplication. First, to examine whether McIdas is sufficient to induce centriole amplification we treated cells with HU and overexpressed either a GFP-McIdas fusion protein or GFP alone as a control. The number of centrioles was determined by staining with markers for the proximal and distal ends of centrioles using Cep135 and CP110, respectively (Fig. 2A). We observed that 34% of the control GFP-overexpressing cells possessed more than 4 centrioles (CP110 dots per cell) following HU treatment, whereas 60% of the GFP-McIdas overexpressing cells had more than 4 centrioles (Fig. 2B). These results indicate that increased McIdas expression is sufficient to directly induce centriole amplification.

Next, we depleted endogenous McIdas from U2OS cells, treated with hydroxyurea (HU), and quantified centriole number. As shown in Fig. 2C,D, nearly half of the control cell population had more than 4 centrioles, whereas only 22% of McIdas-depleted cells showed this phenotype. The centriole overduplication phenotype was rescued in McIdas-depleted cells by overexpression of an RNAi-resistant *GFP-MCIDAS* construct, but not by GFP alone, confirming the specificity of the phenotype (Fig. 2E,F). Overall, these results demonstrate that McIdas is necessary for HU-induced centriole overduplication.

Given that *PLK4* overexpression induces centriole amplification, we next examined whether McIdas also affects PLK4-mediated centriole amplification. To that end, we depleted McIdas in U2OS cells inducibly overexpressing PLK4 (Fig. EV3A–C) and measured the number of centrioles per cell, by using either CP110 or Centrin as markers. As expected, PLK4-overexpressing cells exhibited an increased number of centrioles organized in rosette-like structures (Fig. EV3D,E). However, this phenotype was severely reduced upon McIdas knockdown, as revealed by CP110 (Fig. EV3F,G) and Centrin staining (Fig. EV3H,I). Moreover, we examined the effect of McIdas depletion on cartwheel assembly in PLK4-overexpressing cells. In control cells, McIdas, PLK4 and SAS6 were co-recruited to rosette-like structures (Fig. EV3D,E). On the contrary, McIdas-depleted cells showed a significant increase in the number of cells lacking SAS6 foci, highlighting the requirement of McIdas for PLK4-induced centriole amplification (Fig. 2G,H).

We further tested whether McIdas is required for canonical centriole duplication by depleting its expression in asynchronous U2OS cells (Fig. 3). We focused our analysis on S phase cells where centriole duplication occurs, identified by EdU incorporation, and

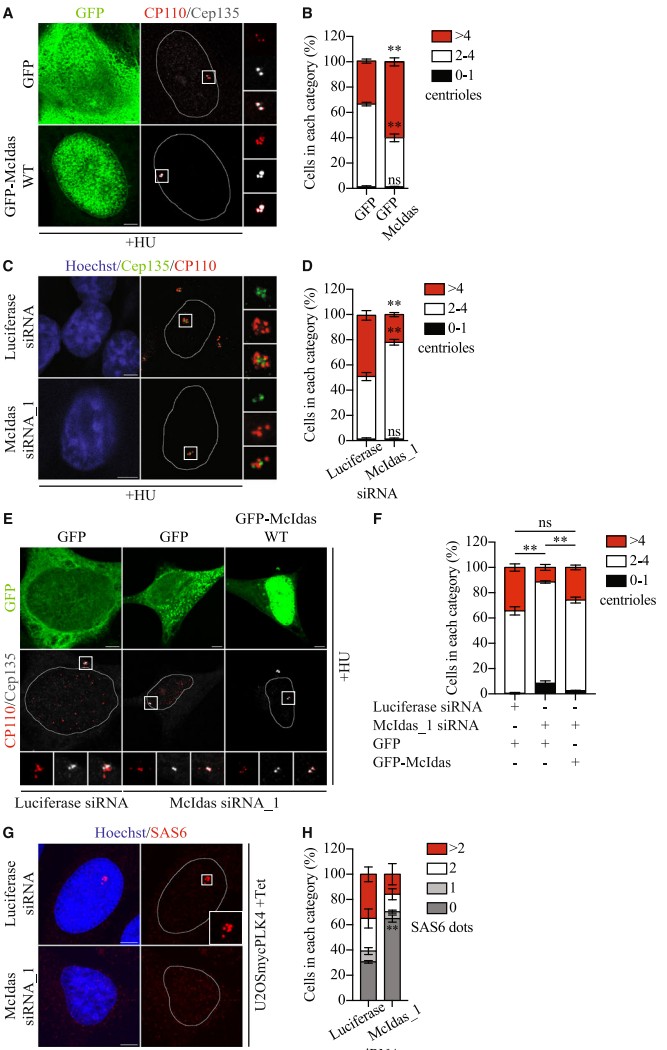

Figure 2. McIdas is essential for centriole amplification.

(A) U2OS cells were treated with hydroxyurea (HU) and 24 h later were transfected with vectors expressing GFP-tagged McIdas or GFP alone as a control. Cells were fixed with methanol 48 h after transfections and centriole numbers were counted using antibodies against Cep135 and CP110, to mark the proximal and distal parts of the centrioles, respectively. (B) Quantification of centriole numbers per cell in GFP- and GFP-McIdas-overexpressing cells. The number of CP110 dots corresponds to the number of centrioles. Data are presented as the mean values of three independent experiments and error bars indicate ± SEM. In each experiment at least 100 cells were counted per condition. P-values were calculated using two-tailed Student's t-test: **P <0.01 and ns, not significant (P values: 0–1 centrioles, P = 0.9528; 2–4 centrioles, P = 0.0012 and >4 centrioles, P = 0.0019). (C) U2OS cells were transfected with McIdas siRNA_1 or control siRNA oligos. 24 h later, after a second transfection with the indicated siRNAs, cells were treated with HU for an additional 48 h. Cells were fixed, and centriole numbers were determined by staining with antibodies against Cep135 and CP110. (D) Quantification of centriole numbers per cell in control and McIdas-depleted cells. The number of CP110 dots corresponds to the number of centrioles. Data are presented as the mean values of three independent experiments and error bars indicate ± SEM. In each experiment at least 100 cells were counted per condition. P-values were calculated using two-tailed Student's t-test: **P <0.01 and ns, not significant (P values: 0–1 centrioles, P = 0.9785; 2–4 centrioles, P = 0.0022 and >4 centrioles, P = 0.0031). (E) U2OS cells were transfected twice with McIdas siRNA_1 or control siRNA oligos. 24 h after the second siRNA transfection, cells were transfected with vectors expressing either GFP-tagged McIdas or GFP alone as a control and then treated with HU for an additional 48 h. Cells were fixed, and centriole numbers were determined by immunostaining with antibodies against Cep135 and CP110. (F) Quantification of centriole numbers per cell. The number of CP110 dots corresponds to the number of centrioles. Data are presented as the mean values of three independent experiments and error bars indicate ± SEM. In each experiment, at least 100 cells were counted per condition. Statistical values shown refer to the >4 centrioles category. P-values were calculated using two-tailed Student's t-test: **P <0.01 and ns, not significant (P values: Luciferase siRNA/GFP-McIdas siRNA/GFP, P = 0.0035; McIdas siRNA/ GFP-McIdas siRNA/GFP-McIdas, P = 0.0087; and Luciferase siRNA/GFP-McIdas siRNA/GFP-McIdas, P = 0.0649). (G) U2OS cells expressing myc-PLK4 under a tetracycline-dependent promoter were transfected with McIdas siRNA_1 or control siRNA oligos. A second transfection followed 24 h later, and then cells were treated with tetracycline to induce myc-PLK4 overexpression. Cells were fixed and immunostained for SAS6. (H) Quantification of SAS6 foci in control and McIdas-depleted cells overexpressing PLK4. Data are presented as the mean values of two independent experiments and error bars indicate ± SEM. At least 100 cells were counted per condition in each experiment. P-value was calculated using two-tailed Student's t-test: **P <0.01 (P value: 0 SAS6 dots, P = 0.0077). Nuclear boundaries are outlined and white boxes indicate regions shown as higher-magnification images. DNA was stained with Hoechst. Scale bars, 5 μm. HU hydroxyurea, Tet tetracycline, ns not significant. Source data are available online for this figure.

counted the number of Centrin foci, which correspond to individual centrioles. Cep192, a structural component of the pericentriolar material (PCM), was used as a second centrosomal marker. Foci were defined as centrosomes only if they were positive for both markers. As expected, most control S-phase cells possessed 4 centrioles. In contrast, McIdas-depleted cells showed a significant increase in cells with only 2 Centrin dots, highlighting unduplicated centrosomes (Fig. 3A,B). To further investigate whether McIdas is involved in cartwheel assembly, we examined SAS6 recruitment to centrosomes in McIdas-depleted asynchronous U2OS cells. Cells were co-immunostained with SAS6 and Cep170, a centriole component of subdistal appendages, and SAS6 foci were quantified in EdU-positive cells. To accurately quantify SAS6 recruitment, we defined foci as centrioles only if they were positive for both SAS6 and Cep170, since background or non-centrosomal signal was occasionally observed for both markers in some cells. Upon McIdas depletion, we observed a significant increase in cells with 0 or 1 SAS-6 focus, compared to control cells, where most EdU-positive cells exhibited two SAS6 foci (Fig. 3C,D).

Altogether, these results strongly support that McIdas is essential for maintaining correct centriole numbers in cells,

affecting SAS6 recruitment, an essential component for cartwheel assembly.

## A nuclear export signal in McIdas mediates its centrosomal localization and supports centriole biogenesis

As McIdas is known to possess a nuclear localization signal and localize to the nucleus (Pefani et al, 2011), we next investigated how McIdas is exported to the cytoplasm for centrosomal localization. Nuclear export signal (NES) sequences are highly diverse, but one of the most well-characterized motifs is a leucine- or isoleucine-rich sequence containing characteristically spaced conserved hydrophobic residues: LX{2,3}[L,I,V,F,M]X{2,3}LX[L,I], where X represents any amino acid (Kosugi et al, 2008). Analysis of the human McIdas amino acid sequence revealed two potential NES-like motifs at residues

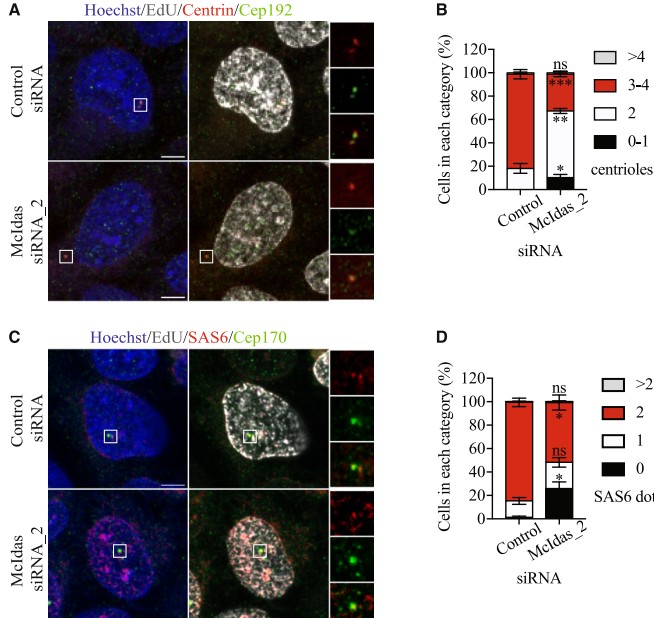

**Figure 3. McIdas is required for daughter centriole biogenesis in cycling cells.**

(A, C) U2OS cells were transfected with McIdas siRNA_2 or control siRNA oligos and immunostained with antibodies against Centrin and Cep192 (a PCM structural component) to count centriole numbers (A) or SAS6 and Cep170 (a centriole component of subdistal appendages) (C). (B) Quantification of centriole numbers per cell in control and McIdas-depleted cells. The number of Centrin dots corresponds to the number of centrioles. Data are presented as the mean values of three independent experiments and error bars indicate ± SEM. At least 100 cells were counted per condition in each experiment. P-values were calculated using two-tailed Student's t-test: *P < 0.1, **P < 0.01, ***P < 0.001 and ns, not significant (P values: 0–1 centrioles, P = 0.0173; 2 centrioles, P = 0.0011; 3–4 centrioles, 0.0004 and >4 centrioles, P = 0.9395). (D) Quantification of SAS6 foci in control and McIdas-depleted cells. Data are presented as the mean values of three independent experiments and error bars indicate ± SEM. At least 100 cells were counted per condition in each experiment. P-values were calculated using two-tailed Student's t-test: *P < 0.1 and ns, not significant (P values: 0 SAS6 dots, P = 0.0137; 1 SAS6 dots, P = 0.1658; 2 SAS6 dots, P = 0.0108 and >2 SAS6 dots, P = 0.8276). White boxes indicate regions shown as higher-magnification images. DNA was stained with Hoechst. Scale bars, 5 μm. ns not significant. Source data are available online for this figure.

71–82 (NES1) and 234–243 (NES2) (Fig. 4A). Multiple sequence alignments of these motifs in human, mouse and Xenopus showed that both sequences are highly conserved, particularly NES2, located at the end of McIdas coiled coil domain (Fig. 4A).

To determine whether these sequences function as NES, we generated three mutants in which the leucine or isoleucine residues were replaced with alanine: one targeting the NES1 sequence in the N-terminal region of McIdas (GFP-McIdas NES1mut), one targeting the NES2 sequence at the end of the coiled-coil domain (GFP-McIdas NES2mut), and a third mutant in which both sites were mutated (GFP-McIdas NES1/2mut), as shown in Fig. 4B. U2OS cells were transfected with either GFP-McIdas WT or the GFP-McIdas NES mutants, and 48 h later, cells were fixed and stained for GFP. The ratio of cytoplasmic to nuclear GFP signal intensity was calculated for each condition. The GFP-McIdas NES2 and the double NES1/2 mutants exhibited a significantly reduced cytoplasmic signal compared to GFP-McIdas WT and GFP-McIdas

NES1, respectively (Fig. EV4A,B), highlighting the importance of NES2 for McIdas translocation from the nucleus to the cytoplasm.

We then examined whether the centrosomal localization of GFP-McIdas depends on these NES sequences. U2OS cells were transfected with GFP-McIdas WT or the GFP-McIdas NES mutants, and 48 h later, treated with MG132 for 4 h to enhance McIdas signal at centrioles (Fig. 4C). Quantification of centrosomal to nuclear GFP signal revealed that the NES2 mutant consistently exhibited significantly reduced centrosomal localization compared to WT or the NES1 mutants (Fig. 4D), highlighting the importance of the NES2 sequence for centrosomal recruitment.

To examine whether centrosomal localization of McIdas is important for centriole duplication, U2OS were treated with HU and transfected with GFP-McIdas WT, NES mutants or GFP as a control (Fig. 4E). As expected, GFP-McIdas WT overexpression induced centriole overduplication (number of Centrin dots per cell) compared to the control GFP-overexpressing cells (Fig. 4F). In contrast, overexpression of GFP-McIdas NES2mut or GFP-McIdas NES1/2mut, both of which showed impaired centrosomal localization, failed to induce centriole overduplication. Notably, GFP-McIdas NES1mut induced centriole amplification to a level comparable to GFP-McIdas WT.

Taken together, these results demonstrate that McIdas contains a novel nuclear export signal at the end of its coiled-coil domain that is critical for its centrosomal localization and function in centriole biogenesis.

## McIdas interacts with and is phosphorylated by PLK4

To investigate how McIdas acts molecularly during centriole duplication, we first asked whether McIdas can interact with key components of the centriole duplication machinery, including PLK4, STIL, and SAS6. To that end, we performed reciprocal immunoprecipitation experiments in U2OS cells expressing myc-PLK4 under a tetracycline-dependent promoter, transfected with either GFP-McIdas or GFP alone as a control. PLK4 was found in McIdas immunoprecipitates and vice versa (Figs. 5A and EV5A). Furthermore, we validated the interaction between endogenously expressed proteins: endogenous McIdas co-immunoprecipitated with endogenous PLK4, supporting the specificity of this interaction (Fig. 5B). In contrast, we did not detect interactions between McIdas and either STIL (Fig. EV6A,B) or SAS6 (Fig. EV6C,D).

Given the specific interaction between McIdas and the kinase PLK4, we next examined whether McIdas could be a substrate of PLK4. First, in vitro kinase assays were performed by incubating immunoprecipitated GFP-McIdas from HEK293T cells with recombinant Xenopus PLK4 (PLX4) and $^{32}$P-radiolabeled ATP. Our analysis showed incorporation of $^{32}$P-radiolabeled ATP into McIdas in the presence of PLK4 (Fig. 5C,D). To test whether PLK4 kinase activity is required for its interaction with McIdas, we treated cells with centrinone, a PLK4 inhibitor. Since PLK4 protein levels are regulated by auto-phosphorylation-induced degradation (Holland et al, 2010), centrinone-mediated inhibition of PLK4 kinase activity resulted in increased PLK4 protein levels (Fig. EV5B). Notably, centrinone treatment had no effect on the McIdas-PLK4 interaction, indicating that PLK4 kinase activity is not required for McIdas binding (Fig. EV5C). Taken together, these data indicate that McIdas interacts with PLK4 and is a novel PLK4 substrate.

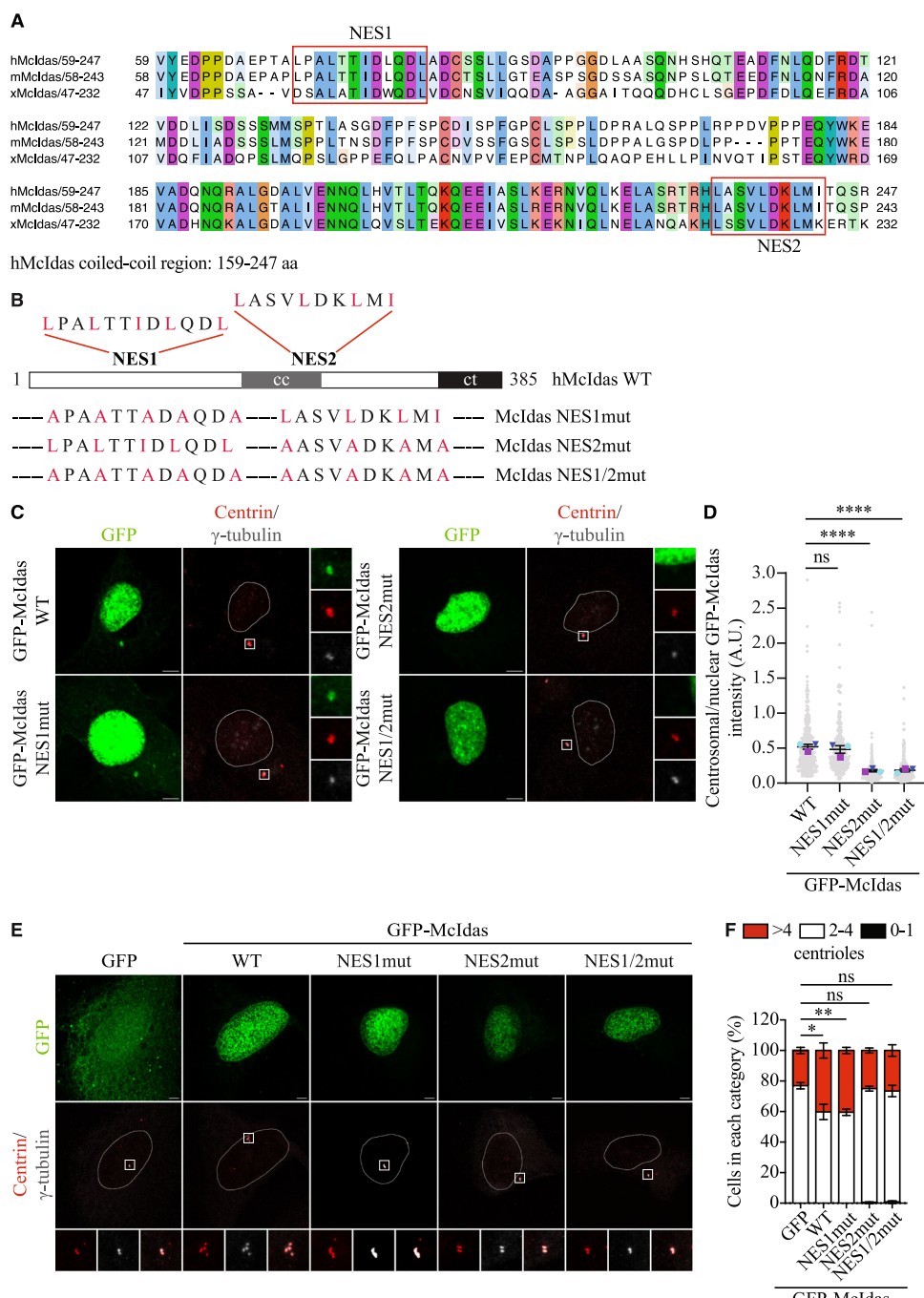

hMcIdas coiled-coil region: 159–247 aa

To determine which region of McIdas mediates its phosphorylation by PLK4, we performed in vitro kinase assays using three truncated recombinant McIdas proteins: an N-terminal fragment (1–179 aa), a fragment containing the central coiled-coil domain (101–284 aa) and a fragment containing the conserved C-terminal region (314–385 aa). Each one of these truncated proteins was incubated with purified PLK4 in the presence of $^{32}$P-radiolabeled ATP. Consistently, we observed that McIdas is indeed phosphorylated by PLK4, but in multiple sites (Fig. 5E). As a negative control, an irrelevant recombinant protein that could not be phosphorylated was included and showed no signal. All the above results

suggest that PLK4 binds to and phosphorylates McIdas at multiple sites.

Next, we performed tandem mass spectrometry analysis to map PLK4-specific phosphorylation sites on the McIdas protein. In vitro-phosphorylated McIdas fragments were subjected to mass spectrometry, revealing 10 serine phosphorylation sites (Fig. 6A). Among these, 4 sites (S55, S58 and S86/87) were located in the N-terminal region, 3 on both sides of the central coiled-coil region (S167, S246, S271), whereas the last 3 in the C-terminal region (S333, S338, S340) (Fig. 6B). We mutated these 10 serine residues to alanine (GFP-McIdas 10A) or aspartic acid (GFP-McIdas 10D),

    

Figure 4. McIdas localizes to centrosomes via a nuclear export signal at the end of its coiled-coil domain.

(A) Multiple sequence alignments of McIdas putative NES sequences in human, mouse, and Xenopus. (B) Schematic representation of the McIdas NES mutant constructs generated in this study. (C) U2OS cells were transfected with vectors expressing GFP-McIdas WT or GFP-tagged NES mutants. 48 h after the transfection, cells were treated with the proteasome inhibitor MG132 for 4 h, fixed with methanol, and immunostained with antibodies against GFP, Centrin, and γ-tubulin. (D) Quantification of centrosomal/nuclear GFP-McIdas fluorescence intensity in each condition. Small symbols represent individual cells, large symbols represent the mean values from three independent experiments. Error bars indicate ± SEM. At least 100 cells were analyzed per condition in each experiment. P-values were calculated using two-tailed Student's t-test: ****$P$ <0.0001 and ns, not significant ($P$ values: WT-NES1mut, $P = 0.1857$; WT-NES2mut, $P < 0.0001$ and WT-NES1/2mut, $P < 0.0001$). (E) U2OS cells were treated with hydroxyurea (HU) and 24 h later were transfected with vectors expressing GFP-McIdas WT, GFP-tagged NES mutants or GFP alone as a control. Cells were fixed with methanol 48 h after transfections and centriole numbers were counted using antibodies against Centrin and γ-tubulin. (F) Quantification of centriole numbers per cell in GFP- and GFP-McIdas WT or NES mutant-overexpressing cells. The number of Centrin dots corresponds to the number of centrioles. Data are presented as the mean values of three independent experiments and error bars indicate ± SEM. In each experiment at least 100 cells were counted per condition. Statistical values shown refer to the >4 centrioles category. P-values were calculated using two-tailed Student's t-test: *$P < 0.1$, **$P < 0.01$ and ns, not significant ($P$ values: GFP-WT, $P = 0.0335$; GFP-NES1mut, $P = 0.0040$; GFP-NES2mut, $P = 0.5067$ and GFP-NES1/2mut, $P = 0.4576$). Nuclear boundaries are outlined and white boxes indicate regions shown as higher-magnification images. DNA was stained with Hoechst. Scale bars, 5 μm. NES Nuclear export signal, cc coiled-coil domain, ct c-terminal domain, AU arbitrary units, ns not significant. Source data are available online for this figure.

generating phospho-dead and phospho-mimetic mutants of McIdas, respectively. HEK293T cells overexpressing either GFP-McIdas WT or GFP-McIdas 10A were subjected to in vitro kinase assay using purified PLK4 and ³²P-radiolabeled ATP. The phospho-dead mutant of McIdas exhibited a markedly reduced incorporation of ³²P-radiolabeled ATP, suggesting the importance of the identified sites for PLK4-mediated phosphorylation (Fig. 6C,E). Interestingly, a slower mobility band was observed for GFP-McIdas WT, suggesting a phosphorylated form, whereas the faster mobility band of GFP-McIdas 10 A represents a non-phosphorylated McIdas form (Fig. 6D). Consistently, incubation of GFP-McIdas WT-, 10A- or 10D-containing extracts with lambda protein phosphatase caused all forms to shift to faster mobility, suggesting that phosphorylation is responsible for the observed differences (Fig. EV7A). Notably, both the phospho-dead (10 A) and phospho-mimetic (10D) mutants also exhibited a mobility shift after phosphatase treatment, indicating that McIdas carries additional phosphorylation sites beyond the identified PLK4-dependent ones. To narrow down the serine residues that could be important for McIdas phosphorylation by PLK4, we generated partial phospho-dead mutants targeting either the N-terminal region (GFP-McIdas 4A_N), the coiled-coil region (3A_cc) or the C-terminal region (3A_C) (Fig. EV7B). This analysis showed that PLK4 preferentially phosphorylates serine residues in the N-terminal and coiled-coil regions (Fig. EV7C–E).

In conclusion, our findings demonstrate that McIdas interacts with PLK4 and is a novel PLK4 substrate. Our data identify multiple PLK4-specific phosphorylation sites on McIdas that could be important for its function at centrosomes.

## PLK4-specific phosphorylation of McIdas is essential for daughter centriole biogenesis

To examine the significance of McIdas phosphorylation by PLK4 in centriole duplication, U2OS cells were treated with HU and transfected with plasmid vectors expressing either GFP-McIdas WT or the 10 A and 10D mutants (Fig. 7A). Quantification of centrioles using the CP110 marker revealed that the GFP-McIdas 10 A phospho-dead mutant was not sufficient to induce centriole amplification upon HU treatment to the same extent as GFP-McIdas WT (Fig. 7B). Western blot analysis confirmed that there were no significant differences in their expression levels (Fig. 7C). The RNAi-resistant GFP-McIdas 10 A phospho-dead mutant was

also unable to rescue the centriole overduplication phenotype when overexpressed in HU-treated U2OS cells depleted of endogenous McIdas (Fig. EV8A,B), unlike the GFP-McIdas WT (as previously shown in Fig. 2E,F), highlighting the specificity of this phenotype. This analysis supports that McIdas phosphorylation by PLK4 is a regulatory mechanism essential for centriole biogenesis. Interestingly, the phospho-mimetic mutant GFP-McIdas 10D, which mimics a continuously phosphorylated protein, was also unable to induce centriole amplification to the same degree as GFP-McIdas WT. This suggests that constant phosphorylation of McIdas by PLK4 inhibits centriole biogenesis. Notably, both the GFP-McIdas 10 A and 10D mutants retained their ability to interact with PLK4 (Fig. EV8C). We therefore conclude that both phosphorylation and dephosphorylation of McIdas are important mechanisms that act antagonistically to control centriole number in cells.

Given that during multiciliogenesis McIdas forms complexes with the transcription factors E2F4/5 and DP1 to drive the multiciliogenesis transcriptional program (Ma et al, 2014), we next asked whether McIdas directly affects centriole numbers or whether its function is mediated through its previously characterized transcriptional role. To separate these functions, we analyzed the ability of GFP-McIdas WT and the phospho-dead or phospho-mimetic mutants to induce the expression of known transcriptional targets, including FoxJ1 and cMyb. As expected, McIdas was able to induce the expression of both FoxJ1 and c-Myb (Fig. 7D,F), particularly when co-expressed with E2F5 in U2OS cells (Fig. 7E,G). Interestingly, the phospho-dead McIdas mutant also efficiently induced expression of both FoxJ1 and c-Myb (Fig. 7D–F). In contrast, neither McIdas WT nor the phospho-mutants induced the expression of genes involved in centriole duplication, such as PLK4 and SAS6 (Fig. EV9A–D). These findings suggest that although the phospho-dead mutant of McIdas is unable to promote centriole amplification, it retains its ability to activate the transcription of factors critical for multiciliogenesis. This highlights a functional separation between McIdas cytoplasmic and transcriptional roles, suggesting that its role in centriole duplication is independent of its transcriptional function.

## Discussion

In this study, we demonstrate that McIdas has a novel, direct role in centriole duplication. McIdas was originally characterized as a

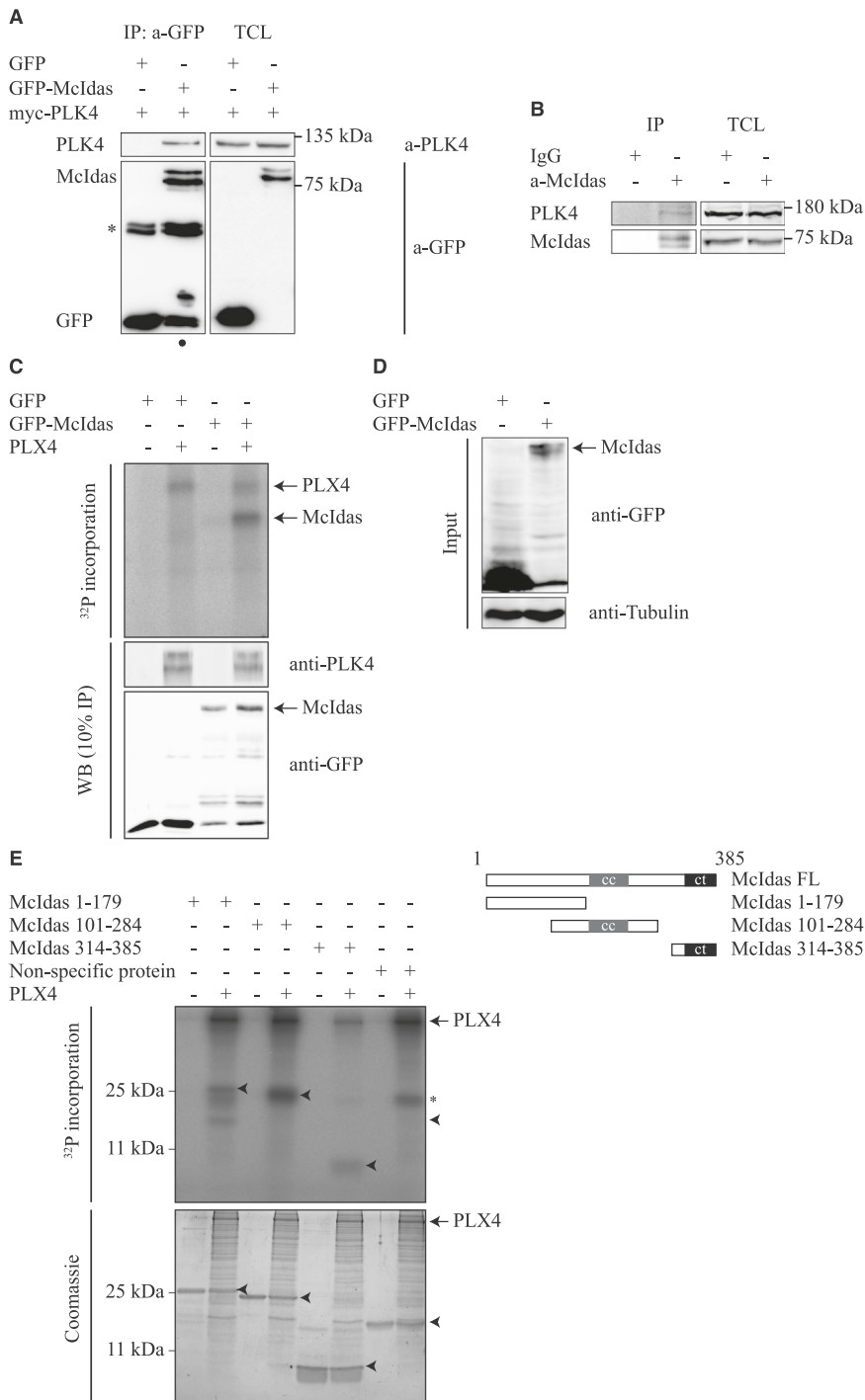

nuclear protein involved in DNA replication initiation (Pefani et al, 2011), interacting with Geminin to regulate its affinity for the licensing factor Cdt1. Here, we show that McIdas also exhibits cytoplasmic localization, mediated by a leucine-rich motif located C-terminal to the McIdas coiled-coil, which is conserved in McIdas homologues and resembles a nuclear export signal (NES). We found that this signal is also essential for McIdas centrosomal localization, suggesting that it may function as a centrosomal localization signal (CLS). The sequence follows the classical NES consensus LX{2,3}[L,I,V,F,M]X{2,3}LX[L,I] (Kosugi et al, 2008), while its mutation disrupts centrosomal targeting. Interestingly, similar leucine-rich motifs are present in other proteins, such as CDC6, BRCA1 and BRCA2, which have also been reported to localize at centrosomes (Fabbro and Henderson, 2003; Han et al, 2008; Kim et al, 2015; Thompson et al, 2005). Centrosomal localization signals have been well-characterized in centrosomal scaffolding proteins, such as pericentrin and AKAP450, which contain a conserved Pericentrin-AKAP450 Centrosomal Targeting

**Figure 5.  McIdas interacts with and is phosphorylated by PLK4 at multiple sites.**

(A) GFP-tagged McIdas or GFP alone as a control were transfected into an inducible U2OS myc-PLK4 overexpressing cell line. Immunoprecipitation was performed against GFP. The asterisk and circle indicate the heavy and light chains of the IgGs, respectively. (B) Western blot analysis of total U2OS cell extracts subjected to immunoprecipitation using an anti-McIdas antibody. IgG immunoprecipitation was used as a control. (C) In vitro kinase assays were performed using control IP or GFP-McIdas, incubated with recombinant Xenopus PLK4 (PLX4) and [γ-$^{32}$P] ATP. Samples were analyzed by autoradiography. A western blot analysis of 10% of each sample used in the kinase assays was performed by staining with antibodies against PLK4 and GFP to verify protein loading. (D) Western blot analysis of the total cell extracts used in (C) for the in vitro kinase assays, stained with antibodies against GFP and α-tubulin. (E) In vitro kinase assays were performed using three recombinant truncated McIdas proteins, as indicated. Each reaction was incubated with recombinant Xenopus PLK4 (PLX4) and [γ-$^{32}$P] ATP and analyzed by autoradiography. Coomassie staining was used to assess protein loading. A non-specific protein was included as a negative control. Arrowheads indicate the expected positions of the recombinant proteins, while the asterisk represents a non-specific band. IP immunoprecipitation, TCL total cell extracts, cc coiled-coil domain, ct c-terminal domain. Source data are available online for this figure.

(PACT) domain (Gillingham and Munro, 2000). This domain is necessary and sufficient for centrosomal anchoring and is distinct from NES motifs. In contrast, ninein, a protein involved in microtubule anchoring at the subdistal appendages, lacks a defined CLS such as the PACT domain, yet it is critical for centrosomal function (Delgehyr et al, 2005). Given the importance of leucine-rich hydrophobic interactions in centrosomal localization (Malik et al, 2016), further studies are needed to determine whether the McIdas NES-like motif mediates direct binding to centrosomal components or functions through a transport mechanism.

McIdas is specifically detected at the central region of centrioles and its centrosomal localization is regulated during the cell cycle. During G1 phase, McIdas is initially present at the mother centriole and is then detected at both centrioles. This is consistent with an early recruitment of McIdas at centrioles prior to the initiation of centriole duplication. As the cell cycle progresses, centrosomal McIdas levels increase and by G2 phase, its presence at the mother centriole diminishes, while it becomes significantly enriched at the daughter centriole (see schematic representations at Fig. 1 and Fig. EV1). It would be interesting to investigate whether McIdas differential localization is linked with processes that take place during G2 phase, including block to reduplication, elongation, and maturation of centrioles or separation of centrosomes. McIdas is lost from centrioles during telophase, at the end of mitosis. A previous study showed that total McIdas protein levels follow a similar pattern (Pefani et al, 2011). Its levels drop during anaphase of mitosis and rise again in early G1 phase. This is reminiscent of the expression pattern followed by other centrosomal regulators, such as STIL and SAS6, which are degraded in late M and early G1 phases by the anaphase-promoting complex/cyclosome associated with Cdh1 (APC/C$^{Cdh1}$). Sequence analysis of the McIdas protein revealed a RALL motif in its N-terminal region, resembling the RXXL-type destruction box consensus found in APC/C substrates (Pefani et al, 2011). We can speculate that regulation of McIdas expression may also be important to prevent untimely initiation of centriole duplication until the next G1/S phase transition.

Centriole duplication is tightly regulated to prevent reduplication within the same cell cycle. Excess centrioles can lead to the formation of multipolar spindles, resulting in chromosome mis-segregation during mitosis, and therefore to aneuploidy (Raff and Basto, 2017). Our findings indicate that the regulation of McIdas expression is important for the maintenance of proper centriole number. Ectopic expression of McIdas is sufficient to increase centriole numbers, while its loss reduces daughter centriole biogenesis. Moreover, McIdas is required for PLK4-induced centriole amplification. Our data suggest that McIdas functions in co-operation with PLK4: immunoprecipitation experiments revealed an interaction between McIdas and PLK4, and we identified McIdas as a novel substrate of this kinase. On the contrary, we were unable to detect binding of McIdas to STIL or SAS6. This could be explained either by low or no affinity of McIdas for these proteins or transient protein-protein interactions. However, SAS6 recruitment to centrioles was significantly reduced upon McIdas depletion. Tandem mass spectrometry analysis revealed multiple PLK4-specific phosphorylation sites on serine residues of McIdas. Importantly, ectopic expression of a McIdas mutant that cannot be phosphorylated by PLK4, failed to induce centriole amplification, unlike the wild-type protein. This demonstrates that PLK4-mediated phosphorylation of McIdas is important for its role in centriole duplication. Surprisingly, the phospho-mimetic McIdas mutant also did not induce centriole over-duplication. This highlights a sophisticated mechanism of McIdas function during centriole duplication cycle, where both its phosphorylation and de-phosphorylation could be important. Together, these findings support a model in which McIdas cooperates with PLK4 during the initiation of centriole duplication. It will be interesting to determine whether McIdas is required for the recruitment or stabilization of PLK4 at the proximal end of the mother centriole together with the CEP192 and CEP63/CEP152 complex (Brown et al, 2013; Cizmecioglu et al, 2010; Dzhindzhev et al, 2014; Hatch et al, 2010; Kim et al, 2013; Sir et al, 2011; Sonnen et al, 2013). Notably, Cep63, which interacts with CEP152 to recruit PLK4 during centriole biogenesis in cycling cells, has a paralogue in multiciliated cells, DEUP1, which is a transcriptional target of McIdas (Ma et al, 2014; Zhao et al, 2013). Furthermore, our data show that McIdas-PLK4 binding is independent of PLK4-mediated phosphorylation, raising the interesting possibility that these two events contribute to McIdas function through distinct mechanisms. Whether McIdas phosphorylartion directly facilitates procentriole assembly and how these processes are coordinated temporally during the centriole duplication cycle, remain important questions for future investigation.

Previous work showed that McIdas forms a stable complex with Geminin and prevents Geminin's binding to Cdt1, suggesting a possible role of McIdas in DNA replication licensing (Pefani et al, 2011). Geminin is a well-known negative regulator of DNA replication initiation, inhibiting re-licensing of DNA replication (McGarry and Kirschner, 1998). Interestingly, Geminin has also been shown to undergo regulated nucleo-cytoplasmic shuttling, which is important for its function in licensing control (Dimaki et al, 2013), but also localizes to centrosomes and contributes to

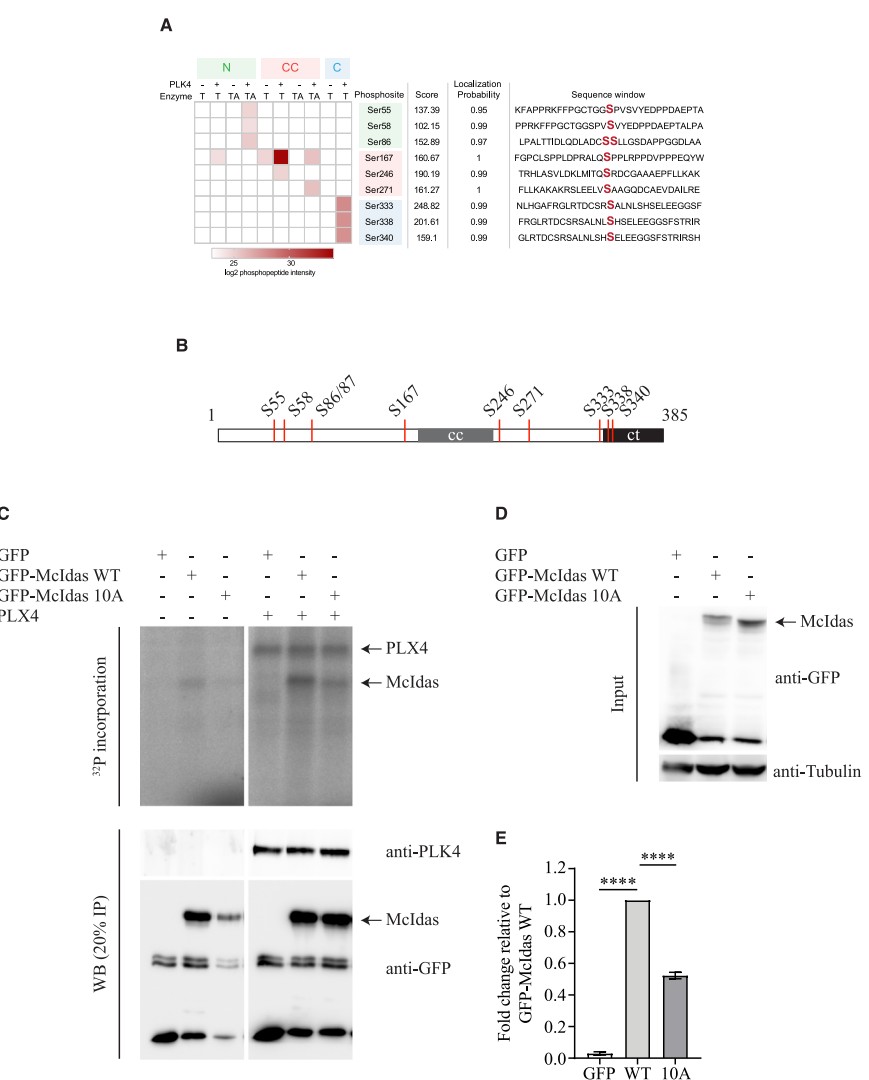

**Figure 6. PLK4 phosphorylates McIdas at 10 serine residues.**

(A) In vitro phosphorylated recombinant McIdas proteins were subjected to tandem mass spectrometry analysis and 10 PLK4-specific serine phosphorylation sites were identified. (B) Schematic representation of the PLK4-phosphorylated serine residues identified on the McIdas protein. (C) In vitro kinase assays were performed using control IP, GFP-McIdas WT, or the phospho-dead GFP-McIdas 10 A mutant. Each IP was incubated with recombinant Xenopus PLK4 (PLX4) and [γ-$^{32}$P] ATP. Samples were analyzed by autoradiography. Western blot analysis of 20% of each reaction was performed using antibodies against PLK4 and GFP to verify protein input. (D) Western blot analysis of the total cell extracts used in (C) for the in vitro kinase assays was performed using antibodies against GFP and α-tubulin. (E) Quantification of McIdas WT and 10 A mutant phosphorylation from (C) was performed using Image Lab software. Values represent phosphorylation signal intensity relative to McIdas WT (set as 1). Data are the mean values of three independent experiments and error bars indicate ± SEM. P-values were calculated using two-tailed Student's t-test: ****$P < 0.0001$ (P values: GFP-WT, $P < 0.0001$ and WT-10A, $P < 0.0001$). Source data are available online for this figure.

once-per-cell-cycle centrosome duplication (Lu et al, 2009; Tachibana et al, 2005; Tachibana and Nigg, 2006). Other components of the pre-replicative complex, including MCM5, Orc1 and Cdc6, have similarly been shown to localize at centrosomes through interactions with S phase cyclins, controlling centriole duplication (Ferguson and Maller, 2008; Ferguson et al, 2010; Hemerly et al, 2009; Xu et al, 2017). Despite these observations, the precise molecular mechanisms linking DNA replication proteins to centriole duplication remain poorly understood. It is intriguing to speculate that McIdas could be a factor that links the centrosome and chromosome cycles, and its function

could be mediated through cooperation with other DNA replication licensing factors.

McIdas has also been identified as a key regulator of multiciliogenesis in Xenopus skin and kidney, as well as in mouse airway and brain ependyma (Arbi et al, 2016; Boon et al, 2014; Kaplani et al, 2024; Kyrousi et al, 2015; Stubbs et al, 2012). Functioning as a transcriptional regulator, McIdas promotes centriole assembly and ciliogenesis, through the formation of complexes with the transcription factors E2F4/5 and DP1 (Ma et al, 2014). Transcriptomic analysis of skin progenitors from Xenopus embryos revealed that the McIdas/E2F4/DP1 complex activates not only ciliogenesis-

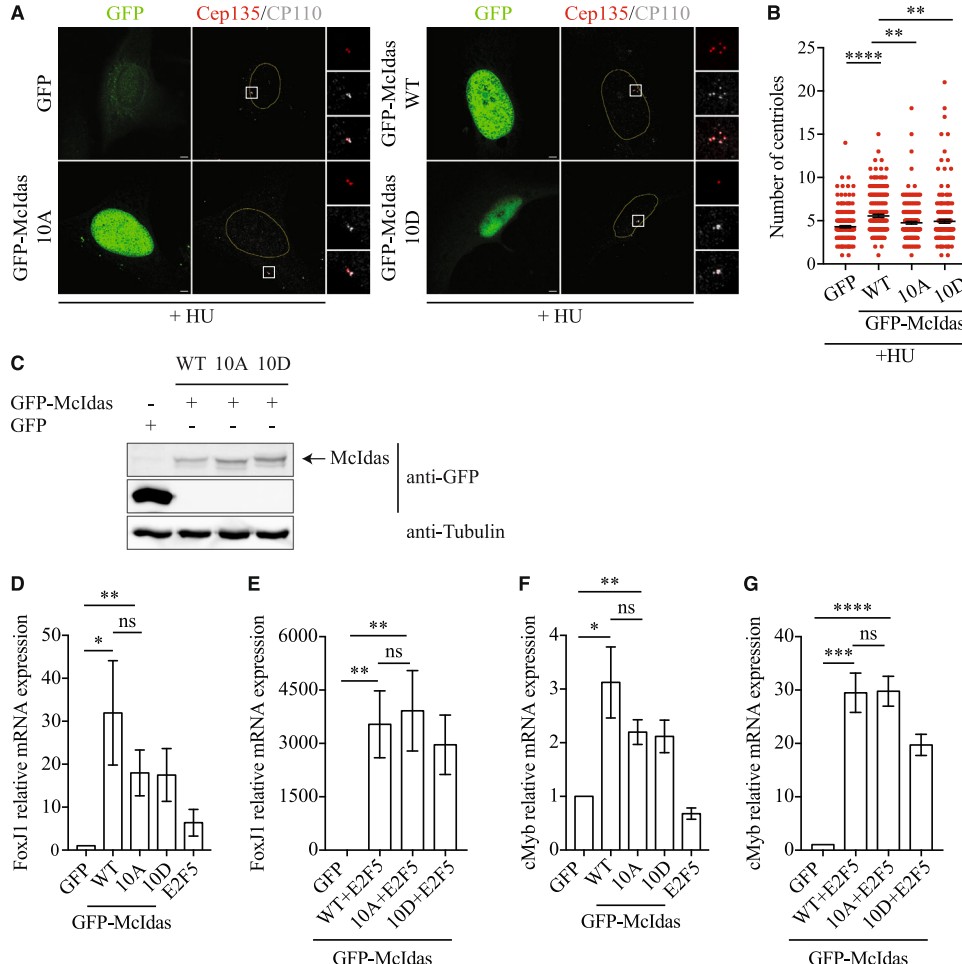

**Figure 7.  PLK4-specific phosphorylation of McIdas is essential for daughter centriole biogenesis.**

(A) U2OS cells were treated with HU and 24 h later transfected with vectors expressing GFP alone as a control, GFP-McIdas WT, GFP-McIdas 10 A (phospho-dead mutant) or GFP-McIdas 10D (phospho-mimetic mutant). Cells were fixed 48 h after transfection and centriole numbers were counted after immunostaining with antibodies against GFP to mark transfected cells, Cep135 and CP110. (B) Quantification of centriole numbers per cell for each condition shown in (A). The number of CP110 dots corresponds to the number of centrioles. Data are presented as individual values from two independent experiments and error bars indicate ± SEM. At least 100 cells were counted per condition in each experiment. *P*-values were calculated using the nonparametric two-tailed Mann–Whitney test: **$P$ <0.01, ***$P$ <0.001, ****$P$ <0.0001 ($P$ values: GFP-WT, $P$<0.0001; WT-10A, $P = 0.0018$ and WT-10D, $P = 0.0001$). (C) Western blot analysis of total cell extracts from U2OS cells transfected with GFP, GFP-McIdas WT or the 10 A and 10D mutants using an anti-GFP antibody. (D–G) U2OS cells were transfected with vectors expressing the indicated genes and mRNA levels of FoxJ1 (D, E) or cMyb (F, G) were assessed by quantitative real-time PCR (qPCR). Data are shown as the mean values of at least three independent experiments and error bars indicate ± SEM. *P*-values were calculated using two-tailed Student's t-test: *$P$ < 0.05, **$P$ < 0.01, ***$P$ < 0.001, ****$P$ < 0.0001 and ns, not significant (FoxJ1 $P$ values: GFP-WT, $P = 0.0288$; GFP-10A, $P = 0.0098$; WT-10A, $P = 0.3169$ and GFP-WT + E2F5, $P = 0.0055$; GFP-10A + E2F5, $P = 0.0086$; WT + E2F5-10A + E2F5, $P = 0.8022$. cMyb $P$ values: GFP-WT, $P = 0.0183$; GFP-10A, $P = 0.0020$; WT-10A, $P = 0.2344$ and GFP-WT + E2F5; $P = 0.0002$; GFP-10A + E2F5, $P$<0.0001; WT + E2F5-10A + E2F5, $P = 0.9494$). Nuclear boundaries are outlined and white boxes indicate regions shown as higher-magnification images. Scale bars, 5 μm. HU hydroxyurea, ns not significant. Source data are available online for this figure.

related transcription factors, but also genes essential for centriole assembly, such as *PLK4, Cep152, STIL* and *Deup1* (Ma et al, 2014). Kim et al analyzed the ability of McIdas to induce multiciliate cell differentiation in primary mouse embryonic fibroblasts (Kim et al, 2018). They showed that McIdas overexpression alone significantly increased centriole number, and this effect was further enhanced upon co-expression with an active form of E2F4. However, a direct role of McIdas in centriole assembly was not examined. Our findings now support a direct role. Notably, a phospho-dead mutant of McIdas, which cannot be efficiently phosphorylated by PLK4, fails to promote centriole amplification when overexpressed,

in contrast to wild-type McIdas. Yet, this mutant retains its ability to activate the expression of cilia-related genes, such as *FoxJ1* and *c-Myb*. This result suggests a separation between the transcriptional and centrosomal functions of McIdas. In line with this, recent studies have revealed an unexpected, non-transcriptional role of E2F4 -McIdas' transcriptional partner- in multicilogenesis (Hazan et al, 2021; Mori et al, 2017). Specifically, E2F4 undergoes nucleocytoplasmic shifting early in multiciliated cell differentiation, a process required for organizing centers of centriole nucleation in the airway epithelium. Furthermore, in parallel with our study, Lu et al reported that McIdas similarly undergoes nucleus-to-

cytoplasm translocation to organize massive de novo centriole biogenesis in multiciliated cells, in cooperation with E2F4/5 (Lu et al, 2025).

Multiciliated cells exit the cell cycle and exhibit an S-phase-like state in which they massively amplify their centrioles (Lewis and Stracker, 2021; Spassky and Meunier, 2017). Recent studies have shown that these cells repurpose core cell cycle regulators, including cyclin-dependent kinases and their associated cyclins, to drive centriole amplification without DNA replication or mitosis (Choksi et al, 2024; Serizay et al, 2025). Yet, the mechanisms determining how a cell decides to duplicate its centrioles once or hundreds of times remain elusive. Centriole biogenesis during multiciliate cell differentiation is accomplished through the same regulators that are required for centriole duplication during the cell cycle. Al Jord et al demonstrated that proteins involved in cell cycle progression are also crucial to drive multiciliated cell differentiation (Al Jord et al, 2017). More recently, LoMastro et al showed that PLK4 protein and its kinase activity are essential for centriole amplification in multiciliated cells (LoMastro et al, 2022), supporting the idea that the early steps of centriole assembly are conserved across cycling and multiciliated cells. Here, we propose that McIdas may act as a regulator influencing the decision between producing two centrioles or many. McIdas function in canonical or non-canonical centriole duplication cycles could be mediated by its expression levels, subcellular localization or differential interactions with specific partners.

In conclusion, our findings identify McIdas as a regulator of centriole duplication cycle in proliferating cells. Given McIdas previously characterized roles in both the cell cycle and multiciliogenesis, we propose that McIdas has a novel role that links its seemingly unrelated functions.

# Methods

### Reagents and tools table

| Reagent/Resource | Reference or Source | Identifier or Catalog Number |
|---|---|---|
| **Experimental models** | | |
| U2OS | ATCC | HTB-96 |
| HeLa | ATCC | CCL-2 |
| 293 T | ATCC | CRL-3216 |
| hTERT RPE-1 | ATCC | CRL-4000 |
| U2OS myc-PLK4 | Kleylein-Sohn et al, 2007 | N/A |
| HeLa GFP | This study | N/A |
| HeLa GFP-McIdas | This study | N/A |
| **Recombinant DNA** | | |
| pLVDest GFP | Arbi et al, 2016 | N/A |
| pENTR1ACmR GFP-McIdas (mouse) | This study | N/A |
| pLVDest GFP-McIdas (mouse) | This study | N/A |
| pENTR1ACmR GFP- McIdas (human) | This study | N/A |
| pLVDest GFP-McIdas (human) | This study | N/A |

| Reagent/Resource | Reference or Source | Identifier or Catalog Number |
|---|---|---|
| pENTR1ACmR GFP-McIdas NES1mut | This study | N/A |
| pLVDest GFP-McIdas NES1mut | This study | N/A |
| pENTR1ACmR GFP-McIdas NES2mut | This study | N/A |
| pLVDest GFP-McIdas NES2mut | This study | N/A |
| pENTR1ACmR GFP-McIdas NES1/2mut | This study | N/A |
| pLVDest GFP-McIdas NES1/2mut | This study | N/A |
| pENTR1ACmR GFP-McIdas 10 A | This study | N/A |
| pLVDest GFP-McIdas 10 A | This study | N/A |
| pENTR1ACmR GFP-McIdas 10D | This study | N/A |
| pLVDest GFP-McIdas 10D | This study | N/A |
| pENTR1ACmR GFP-McIdas 4A_N | This study | N/A |
| pLVDest GFP-McIdas 4A_N | This study | N/A |
| pENTR1ACmR GFP-McIdas 3A_cc | This study | N/A |
| pLVDest GFP-McIdas 3A_cc | This study | N/A |
| pENTR1ACmR GFP-McIdas 3A_C | This study | N/A |
| pLVDest GFP-McIdas 3A_C | This study | N/A |
| psPAX2 | Addgene | Cat #12260 |
| pMD2.G | Addgene | Cat #12259 |
| pcDNA3.1 HA | Pefani et al, 2011 | N/A |
| pcDNA3.1 McIdas-HA | Pefani et al, 2011 | N/A |
| pCMV HA-STIL | Ohta et al, 2014 | N/A |
| pEBTet HsSAS6-GFP | Keller et al, 2014 | N/A |
| pEBTet HsSAS6-SNAP-FLAG | Keller et al, 2014 | N/A |
| pCMV HA-E2F5 | Addgene (gift from Kristian Helin) | Cat #24213 |
| pET20b(+) McIdas C-terminal region (314–385 aa) | This study | N/A |
| **Antibodies** | | |
| rabbit anti-hMcIdas | Pefani et al, 2011 | N/A |
| rabbit anti-CP110 | Jiang et al, 2012 | N/A |
| rabbit anti-PLK4 | Mónica Bettencourt-Dias lab | N/A |
| rabbit anti-γ-tubulin | Sigma-Aldrich | Cat #T5192 |
| rabbit anti-Cep192 | Proteintech | Cat #18832-1-AP |
| rabbit anti-Cep170 | Abcam | Cat #ab72505 |
| rabbit anti-STIL/SIL | Abcam | Cat #ab89314 |
| mouse anti-Centrin (clone 20H5) | Merck Millipore | Cat #04-1624 |
| mouse anti-SAS6 | Santa Cruz Biotechnology | Cat #sc-81431 |
| mouse anti-PLK4 (clone 6H5) | Merck Millipore | Cat #MABC544 |
| mouse anti-acetylated α-tubulin (clone 6-11B-1) | Sigma-Aldrich | Cat #T6793 |
| mouse anti-GFP (clones 7.1 & 13.1) | Roche | Cat #11814460001 |

| Reagent/Resource | Reference or Source | Identifier or Catalog Number |
|---|---|---|
| mouse anti-HA (clone F-7) | Santa Cruz Biotechnology | Cat #sc-7392 |
| mouse anti-myc (clone 4A6) | Merck Millipore | Cat #05-724 |
| mouse anti-Flag (clone M2) | Sigma-Aldrich | Cat #F-3165 |
| mouse anti-α-tubulin | Sigma-Aldrich | Cat #T5168 |
| chicken anti-GFP | Aves Labs | Cat #GFP-1010 |
| goat anti-Cep164 | Santa Cruz Biotechnology | Cat #sc-240226 |
| rat anti-Cep135 | Mónica Bettencourt-Dias lab | N/A |
| Alexa Fluor 488 goat anti-Rabbit | Invitrogen | Cat #A11034 |
| Alexa Fluor 488 goat anti-Chicken | Invitrogen | Cat #A11039 |
| Alexa Fluor 488 goat anti-Rat | Invitrogen | Cat #A11006 |
| Alexa Fluor 568 donkey anti-Goat | Invitrogen | Cat #A11057 |
| Alexa Fluor 568 goat anti-Mouse | Invitrogen | Cat #A11004 |
| Alexa Fluor 647 donkey anti-Rabbit | Invitrogen | Cat #A31573 |
| goat anti-Rabbit IgG-HRP | Merck Millipore | Cat #12-348 |
| goat anti-Mouse IgG-HRP | Merck Millipore | Cat #12-349 |
| Click-iT EdU Cell Proliferation Kit for Imaging, Alexa Fluor 647 dye | Invitrogen | Cat #C10340 |
| **Oligonucleotides and other sequence-based reagents** | | |
| PCR primers | This study | Table EV1 |
| McIdas siRNA_1 | This study | Table EV2 |
| Luciferase siRNA | This study | Table EV2 |
| McIdas siRNA_2: MCIDAS Silencer™ Select Pre-designed siRNA | Thermo Fisher Scientific | siRNA ID s51151 |
| Silencer™ Select Negative Control No. 1 siRNA | Thermo Fisher Scientific | Cat #4390843 |
| **Chemicals, Enzymes and other reagents** | | |
| Hoechst 33258 | Sigma-Aldrich | Cat #94403 |
| DAPI | AppliChem | Cat #A1001 |
| Mowiol 4-88 | Merck Millipore | Cat #475904 |
| Acryloyl-X | Thermo Fisher Scientific | Cat #A20770 |
| Sodium Acrylate | Merck Millipore | Cat #408220 |
| Proteinase K | Invitrogen | Cat #25530-015 |
| Polyethylenimine (PEI) | Polysciences | Cat #23966 |
| Tetracycline | Sigma-Aldrich | Cat #A39246 |
| Protein A agarose bead suspension | Merck Millipore | Cat #IP02 |
| PVDF membrane | Merck Millipore | Cat #IPVH00010 |
| Clarity Western ECL Substrate | Bio-Rad | Cat #1705061 |
| Isopropyl β-D-1-thiogalactopyranoside (IPTG) | UBPBio | P1010 |
| Lambda protein phosphatase, Biolabs | New England Biolabs | Cat #P0753 |

| Reagent/Resource | Reference or Source | Identifier or Catalog Number |
|---|---|---|
| Recombinant Xenopus PLK4 (PLX4) | Dr. Monica Bettencourt-Dias Lab | N/A |
| SteadyRed Adenosine 5'-[γ-P32] triphosphate, 6000 Ci/mmol, 250 μCi | Hartmann Analytic | Cat #SRP-501 |
| Nucleospin RNA II kit | Macherey-Nagel | Cat #740955 |
| M-MLV Reverse Transcriptase | Invitrogen | Cat #28025013 |
| Lipofectamine RNAiMAX Transfection Reagent | Invitrogen | Cat #13778075 |
| Lipofectamine 2000 Transfection Reagent | Invitrogen | Cat #11668027 |
| Turbofect | Thermo Fisher Scientific | Cat #R0531 |
| MG132 | Sigma-Aldrich | Cat #474790 |
| Centrinone | MedChemExpress | Cat #HY-18682 |
| Polybrene | Sigma-Aldrich | Cat #H9268 |
| Hydroxyurea | Sigma-Aldrich | Cat #H8267 |
| EdU (5-ethynyl-2'-deoxyuridine) | Invitrogen | Cat #C10340 |
| Gateway LR Clonase II enzyme mix | Invitrogen | Cat #11791 |
| **Software** | | |
| Fiji: ImageJ | Open source | https://imagej.net/Fiji |
| CellProfiler | Open source | https://cellprofiler.org |
| Image Lab Software | Bio-Rad | N/A |
| GraphPad Prism | GraphPad Software | N/A |
| **Other** | | |
| μ-slide 2-well glass-bottom | Ibidi | Cat #80287 |
| ChemiDoc Imaging System | Bio-Rad | N/A |
| Amicon® Ultra centrifugal filter, 3 kDa MWCO | Merck Millipore | Cat #UFC9003 |
| HisTrap FF column | Cytiva | Cat #17525501 |

## Plasmids

The control GFP plasmid (pLVDest GFP) used in this study was previously described (Arbi et al, 2016). To generate a GFP-tagged mouse McIdas construct, the full-length mouse McIdas cDNA (Pefani et al, 2011) was cloned with an N-terminal GFP tag into the KpnI/XhoI restriction sites of the pENTR1AminusCmR vector. For the human GFP-McIdas construct, the full-length human McIdas cDNA (GenScript, OHu00715) was cloned with an N-terminal GFP tag into the EcoRI/XhoI sites of the pENTR1AminusCmR vector. NES mutants of McIdas were obtained from GenScript. Briefly, all leucine or isoleucine residues within the NES1-like sequence (71–82), NES2-like sequence (234–243), or both were replaced with alanine. NES1, NES2 or NES1/2 mutant sequences were cloned with an N-terminal GFP tag into pENTR1AminusCmR using PmlI/SacI (NES1 and NES1/2) or KasI/PmlI (NES2)

restriction sites, generating the constructs GFP-McIdas NES1mut, GFP-McIdas NES2mut, and GFP-McIdas NES1/2mut. Phospho-dead and phospho-mimetic McIdas mutants were also obtained from GenScript. Ten serine residues (S55, S58, S86, S87, S167, S246, S271, S333, S338 and S340) were mutated either to alanine (phospho-dead mutant, McIdas 10 A) or to aspartic acid (phospho-mimetic mutant, McIdas 10D). Each mutant was cloned with an N-terminal GFP tag into the EcoRI/XhoI sites of the pENTR1A-minusCmR vector to generate the GFP-McIdas 10 A and GFP-McIdas 10D constructs. The GFP-McIdas 10 A plasmid was used to generate partial phospho-dead mutants (GFP-McIdas 4A_N, 3A_cc & 3A_C). Subcloning into the GFP-McIdas WT backbone was performed using SacII/MscI (for 4A_N), MscI/KasI (for 3A_cc) and KasI/XhoI (for 3A_C) restriction sites. Final expression vectors were generated by LR recombination using the Gateway LR clonase II enzyme mix (Invitrogen, 11791) between attL-containing entry clones and attR-containing pLVDest-CAG destination vectors to produce the constructs: pLVDest GFP-McIdas (mouse), pLVDest GFP-McIdas (human), pLVDest GFP-McIdas NES1mut, pLVDest GFP-McIdas NES2mut, pLVDest GFP-McIdas NES1/2mut, pLVDest GFP-McIdas 10 A, pLVDest GFP-McIdas 10D, pLVDest GFP-McIdas 4A_N, pLVDest GFP-McIdas 3A_cc and pLVDest GFP-McIdas 3A_C.

Additional plasmids used in this study include: pcDNA3.1 McIdas-HA and pcDNA3.1 HA control vectors (Pefani et al, 2011), pCMV HA-STIL (Ohta et al, 2014), pEBTet HsSAS6-GFP and pEBTet HsSAS6_SNAP-FLAG, kindly provided by Dr. Pierre Gonczy (1 μg/ml tetracycline was used for efficient induction (Keller et al, 2014) and pCMVHA-E2F5, a gift from Kristian Helin (Addgene plasmid #24213).

## Cell culture and treatments

U2OS (ATCC, HTB-96), HeLa (ATCC, CCL-2) and 293 T (ATCC, CRL-3216) cells were cultured in DMEM (Gibco) supplemented with 10% fetal bovine serum (FBS) in 5% $CO_2$ at 37 °C. hTERT RPE-1 cells (ATCC, CRL-4000) were cultured in DMEM/F12 (Sigma-Aldrich) supplemented with 10% FBS under the same conditions. U2OS cells expressing myc-PLK4 under a tetracycline-dependent promoter (Kleylein-Sohn et al, 2007) were grown in DMEM supplemented with 10% tetracycline-free FBS (PAN-Biotech) in 5% $CO_2$ at 37 °C. For efficient myc-PLK4 over-expression, cells were cultured in medium containing 100 ng/ml tetracycline (Sigma-Aldrich).

For the generation of the stable cell lines, HeLa cells were infected with lentiviral particles expressing either GFP or GFP-McIdas. Lentiviral particles were produced using a second-generation packaging system, as described previously (Arbi et al, 2016). Briefly, 293 T cells were co-transfected with either the GFP (pLVDest GFP) or GFP-MCIDAS (pLVDest GFP-McIdas) expression vector, together with the two helper plasmids psPAX2 (packaging vector, Addgene 12260) and pMD2.G (envelope vector, Addgene 12259), using Turbofect (Fermentas) as the transfection reagent. Supernatants were harvested 48 h post-transfection, filtered, and used to infect HeLa cells in the presence of 5 μg/ml hexadimethrine bromide (polybrene, Sigma-Aldrich). Limiting dilution and single-colony picking were used to establish stable cell lines.

To inhibit proteasome activity, Hela cells stably expressing GFP or GFP-McIdas were treated with MG132 (100 μM, Sigma-Aldrich) for 4 h. For PLK4 kinase activity inhibition, U2OS cells expressing myc-PLK4 under a tetracycline-dependent promoter were cultured overnight in the presence of 100 nM centrinone (MedChemExpress, HY-18682) or DMSO (Sigma-Aldrich) as a control.

## HU-induced centriole amplification assay and transfections

U2OS cells were seeded in 6-well plates at 80% confluency and treated with 4 mM HU. After 24 h, cells were transfected with 1 μg of the indicated plasmids using polyethylenimine (PEI, 3 μg/ml, Polysciences, 23966). Cells were fixed and processed for immunostaining 48 h post-transfection.

## RNA interference

For McIdas RNAi experiments, custom STEALTH siRNA oligos (McIdas siRNA_1; Invitrogen) were used (Table EV1), as described previously (Pefani et al, 2011). A luciferase siRNA was used as a control (Table EV1). A second siRNA oligo targeting McIdas (McIdas siRNA_2; s51151, Silencer® Select siRNA, Thermo Fisher Scientific) was also used. The Silencer® Select No. 1 siRNA (4390843, Thermo Fisher Scientific) was used as a negative control. U2OS or U2OS-mycPLK4 cells were seeded onto coverslips at 80% confluency in 6-well plates and transfected with the indicated siRNA oligos using RNAiMAX (Life Technologies), according to the manufacturer's instructions. A second siRNA transfection was performed 24 h later. McIdas siRNA_1 or luciferase control siRNAs were used at a final concentration of 80 nM, while McIdas siRNA_2 or the Silencer® Select control siRNA oligos were used at 40 nM. For HU-induced centriole amplification, 5 h after the second transfection, cells were treated with 4 mM HU. For PLK4-induced centriole amplification, U2OS-mycPLK4 cells were cultured in medium containing 100 ng/ml tetracycline 5 h after the second transfection. To label S phase cells, EdU (10 μM, Invitrogen) was added to the culture medium for 1 h before fixation. Cells were either analyzed by qPCR or western blotting to quantify McIdas mRNA and protein levels, or fixed and immunostained 48 h after the second transfection.

## Immunofluorescence and expansion microscopy (ExM)

Cells were grown on coverslips, washed twice with 1x PBS, and fixed with pre-cooled methanol (−20 °C) for 10 min at −20 °C. For the detection of GFP-tagged McIdas at centrosomes, pre-extraction was performed prior to methanol fixation. Briefly, cells were treated with CSK buffer (10 mM PIPES pH 7.9, 300 mM sucrose, 100 mM NaCl and 3 mM $MgCl_2$) for 1 min at room temperature, followed by incubation in CSK buffer containing 0.5% Triton X-100, 1 mM PMSF, and 1x protease inhibitor cocktail for an additional 1 min. To visualize centrioles with acetylated α-tubulin for ExM, cells were incubated on ice for 30 min to depolymerize cytoplasmic micro-tubules before fixation. S-phase cells were stained using the Click-iT Alexa Fluor 647 Imaging Kit (Invitrogen) after fixation, according to the manufacturer's instructions. Cells were subse-quently blocked in a solution containing 10% FBS and 3% BSA, in

1x PBS for 1 h, followed by overnight incubation at 4 °C with primary antibodies diluted in blocking solution. The primary antibodies used were: rabbit anti-hMcIdas (1:250) (Pefani et al, 2011), mouse anti-Centrin (1:500–1:1000, Merck Millipore), chicken anti-GFP (1:1000, Aves Labs), rat anti-Cep135 (1:500, Mónica Bettencourt-Dias lab, produced against human CEP135 by Metabion, aa 270–371, expressed in *E. coli* as an N-terminal 10x His-tagged fusion protein), goat anti-Cep164 (1:500, Santa Cruz Biotechnology), mouse anti-SAS6 (1:500, Santa Cruz Biotechnology), rabbit anti-CP110 ((1:250, Jiang et al, 2012), rabbit anti-PLK4 (1:500, Mónica Bettencourt-Dias lab, produced against the c-terminal region, aa 510–970 of human PLK4 and purified using a membrane-bound c-terminal region of PLK4), rabbit anti-γ-tubulin (1:1000, Sigma-Aldrich), rabbit anti-Cep192 (1:500, Proteintech) and rabbit anti-Cep170 (1:500, Abcam). After three washes with 1x PBS containing 0.1% Tween-20, Alexa Fluor-conjugated secondary antibodies (Invitrogen) were used at 1:500 in blocking solution and incubated for 1 h at room temperature: Alexa Fluor 488 goat anti-Rabbit, Alexa Fluor 568 goat anti-Mouse, Alexa Fluor 488 goat anti-Chicken, Alexa Fluor 488 goat anti-Rat, Alexa Fluor 568 donkey anti-Goat and Alexa Fluor 647 donkey anti-Rabbit. DNA was stained with Hoechst 33258 (1:1000, Sigma-Aldrich) or DAPI (1:1000, AppliChem). Coverslips were mounted in Mowiol 4-88 (Merck Millipore) or further processed for ExM (Asano et al, 2018).

For ExM, cells were incubated with 0.1 mg/ml Acryloyl-X (Thermo Fisher Scientific) in 1x PBS at room temperature overnight. The following day, cells were washed twice with 1x PBS for 15 min and incubated in monomer solution (900 mM Sodium Acrylate, 2.6% Acrylamide 37.5:1, 0.15% N, N'-Methylenebisacrylamide, 2 M NaCl, PBS 1x) containing 4-HT (0.01%), APS (0.2%) and TEMED (0.2%) in a 47:1:1:1 ratio for 2 h at 37 °C. Following polymerization, gels were digested with 8 U/ml Proteinase K (Thermo Fisher Scientific) in a solution containing 50 mM Tris-HCl pH 8, 1 mM EDTA, 1 M NaCl and 0.5% Triton X-100 for 1 h at 37 °C. Gels were expanded in Milli-Q water (3 × 20 min washes) and imaged on poly-L-lysine-coated μ-slide 2-well glass-bottom (Ibidi). The expansion factor was approximately 4x. The primary and secondary antibodies used for the expanded samples were: rabbit anti-hMcIdas (1:100) (Pefani et al, 2011), mouse anti-acetylated α-tubulin (1:250, Sigma), Alexa Fluor 488 goat anti-Rabbit (1:250, Invitrogen) and Alexa Fluor 568 goat anti-Mouse (1:250, Invitrogen).

Images were acquired using a Leica TCS SP5 confocal system with a 63x NA 1.4 oil immersion objective. XY was set at either 512 × 512 or 1024 × 1024 pixels, using a bidirectional scanning speed of 400 Hz. Z-stacks were collected with a step size of 0.2 μm. For expanded samples, images were acquired frame-by-frame to minimize gel drift.

Digital image processing was carried out using ImageJ (Rueden et al, 2017). Image intensity quantifications were performed on SUM projections of Z-stacks using a custom ImageJ macro. Multichannel intensity profiles, as a function of distance on a line connecting the local maximal intensity points marking the centrioles, were also calculated using a custom ImageJ macro.

Quantification of cytoplasmic versus nuclear GFP-McIdas (WT or NES mutants) signal was performed using a custom pipeline in CellProfiler. Briefly, a mask was generated around the nucleus using the DAPI channel, and the mean GFP intensity within this mask

(nuclear intensity) was measured. The mask was then expanded by 5 pixels, and the original nuclear mask was subtracted to define a 5-pixel cytoplasmic ring around the nucleus. Mean GFP intensity within this ring (cytoplasmic intensity) was measured, and the cytoplasmic/nuclear intensity ratio was calculated per cell. For centrosomal versus nuclear intensity quantification, a custom macro in ImageJ was used. First, a mask was generated around the nucleus using the Hoechst channel, and the mean GFP intensity within this mask (nuclear intensity) was measured on SUM projections of the Z-stacks. A second mask was generated around the centrosome using the γ-tubulin channel, which was then applied to the GFP channel to measure mean centrosomal GFP intensity. The centrosomal/nuclear intensity ratio was calculated for each cell.

## Co-immunoprecipitation and western blot analysis

Immunoprecipitation was performed as previously described (Lalioti et al, 2019). U2OS cells expressing myc-PLK4 under a tetracycline-dependent promoter or 293 T cells were transiently transfected in 10 cm plates, using polyethylenimine (PEI) with a total of 6 μg of the indicated plasmids. For induction of myc-PLK4 overexpression, 100 ng/ml tetracycline (Sigma-Aldrich) was added to the growth medium 5 h post-transfection. Cells were collected 48 h after transfection, washed twice with ice-cold PBS (1x) and incubated for 10 min in lysis buffer (50 mM Tris-HCl pH 8.2, 150 mM NaCl, 5 mM EDTA, 5 mM MgCl$_2$, 0.5% Triton X-100) supplemented with fresh protease and phosphatase inhibitors (1 mM PMSF, 0.1 mM Na$_3$VO$_4$ and 1x PIC - Roche). Lysates were passed through a 1 ml syringe for mechanical disruption and centrifuged at 13,000 rpm for 10 min at 4 °C. In parallel, for each IP reaction, 50 μl of Protein A agarose bead suspension (Merck Millipore, IP02) was incubated with 1.5 μg of the indicated antibodies for 2 h on a rotating wheel at 4 °C. Beads were washed three times with lysis buffer and incubated with 1–2 mg of total protein from whole-cell lysates for 3 h at 4 °C with rotation. Beads were washed again with lysis buffer, and proteins were eluted in Laemmli buffer by boiling for 5 min. Eluted proteins were analyzed by SDS-PAGE and transferred onto PVDF membrane (Merck Millipore). Membranes were washed with 1x PBS containing 0.1% Tween-20 (PBS-T), blocked in 5% milk in PBS-T for 1 h at room temperature, and incubated overnight at room temperature with primary antibodies. After washing, membranes were incubated with HRP-conjugated secondary antibodies for 1 h at room temperature and visualized either on X-ray film (Santa Cruz Biotechnology) or using the ChemiDoc Imaging System (Bio-Rad) following incubation with Clarity ECL substrate (Bio-Rad). The antibodies used for immunoprecipitation were: mouse anti-GFP (clones 7.1 & 13.1, Roche), mouse anti-HA (clone F-7, Santa Cruz Biotechnology), mouse anti-myc (clone 4A6, Merck) and rabbit anti-hMcIdas (Pefani et al, 2011). The primary and secondary antibodies used for western blot analysis were: mouse anti-GFP (1:1000, Roche), mouse anti-myc (1:1000, clone 4A6, Merck), mouse anti-PLK4 (1:500, Merck), rabbit anti-STIL (1:500, Abcam), mouse anti-HA (1:1000, Santa Cruz Biotechnology), mouse anti-Flag (1:1500, Sigma-Aldrich), rabbit anti-hMcIdas (Pefani et al, 2011), goat anti-Rabbit IgG-HRP (1:1000–1:3000, Merck Millipore) and goat anti-Mouse IgG-HRP (1:1000–1:3000, Merck Millipore).

## Protein expression and purification

For the in vitro kinase assays, three truncated McIdas recombinant proteins were used: an N-terminal fragment (1–179 aa), a fragment containing the central coiled-coil domain (101–284 aa) and a fragment containing the conserved C-terminal region of McIdas (314–385 aa). McIdas recombinant proteins containing the N-terminal region (similar to McIdas 1–127 aa) and the central coiled-coil domain (101–284 aa) were described previously (Pefani et al, 2011). The C-terminal region (314–385 aa) was amplified from a synthetic, codon-optimized gene purchased from GenScript, and cloned into the pET20b(+) vector. Primer sequences are shown in Table EV2. The obtained construct was verified by DNA sequencing. The final derived polypeptide was expressed fused with a C-terminal His$_6$-tag.

For the expression of the C-terminal region of McIdas (314–385 aa), *Escherichia coli* BL21 (DE3) Singles™ Competent Cells (Novagen) were transformed with the vector. Cells were grown in LB medium containing ampicillin until reaching an O.D. value 0.6–0.8. Expression was induced with 1 mM isopropyl β-D-1-thiogalactopyranoside (IPTG) for 4 h at 37 °C. Cells were harvested and resuspended in buffer containing 50 mM Tris-HCl pH 8, 300 mM NaCl, 0.1% Triton X-100. After two repeated cycles of sonication and centrifugation, the derived pellet was resuspended in 50 mM Tris-HCl pH 8, 500 mM NaCl, 10 mM Imidazole, 2 mM β-mercaptoethanol and 6 M urea, followed by 2 h agitation and centrifugation at 14,000 rpm. The supernatant was loaded onto a HisTrap column (GE Healthcare) and the C-terminal region of McIdas was purified in the presence of urea, followed by buffer exchange in 50 mM Tris-HCl pH 8, 300 mM NaCl and 2 mM β-mercaptoethanol. The desired concentration was achieved using Amicon® Ultra centrifugal filter units (3 kDa cutoff – Merck Millipore).

## Kinase assays

293 T cells were transiently transfected in 10 cm plates using polyethylenimine (PEI) with 6 μg of the indicated plasmids. Cells were collected 48 h after transfection, washed twice with ice-cold 1x PBS and incubated for 10 min in lysis buffer without phosphatase inhibitors (50 mM Tris-HCl pH 8.2, 150 mM NaCl, 0.5% Triton X-100), supplemented with fresh protease inhibitors (1 mM PMSF and 1x PIC - Roche). Cell lysates were passed through a 1 ml syringe for mechanical disruption and centrifuged at 13,000 rpm for 10 min. To release phosphate groups from phosphorylated residues, cell lysates were subjected to phosphatase treatment for 30 min at 30 °C, according to the manufacturer's instructions (Lambda protein phosphatase, Biolabs). In parallel, for each IP reaction, 50 μl of Protein A agarose bead suspension (Merck Millipore, IP02) was incubated with 1.5 μg of anti-GFP antibody (mouse anti-GFP, clones 7.1 & 13.1, Roche, 11814460001) for 2 h on a rotating wheel at 4 °C. Beads were washed three times with lysis buffer without phosphatase inhibitors and then incubated with 1–2 mg of total protein from whole-cell lysates for 3 h at 4 °C, with gentle rotation. Beads were washed three times with lysis buffer (50 mM Tris-HCl pH 8.2, 150 mM NaCl, 5 mM EDTA, 5 mM MgCl$_2$, 0.5% Triton X-100) supplemented with fresh protease and phosphatase inhibitors (1 mM PMSF, 0.1 mM Na$_3$VO$_4$ and 1x PIC - Roche), followed by three washes with kinase buffer (25 mM Tris-

HCl pH 7.5, 25 mM NaCl, 10 mM MgCl$_2$ and 1 mM DTT). Each IP sample was incubated with 500 ng of recombinant Xenopus PLK4 (PLX4, provided by Dr. Mónica Bettencourt-Dias) and 100 μM [γ-$^{32}$P] ATP for 30 min at 30 °C. Samples were analyzed by SDS-PAGE followed by autoradiography. Western blot analysis was performed in the 10–20% of each sample, as well as in the total cell extracts used for the kinase assays. The primary and secondary antibodies used were: mouse anti-GFP (1:1000, Roche), rabbit anti-PLK4 (1:500), mouse anti-α-tubulin (1:6000, Sigma-Aldrich), goat anti-Rabbit IgG-HRP (1:1000–1:3000, Merck Millipore) and goat anti-Mouse IgG-HRP (1:1000–1:3000, Merck Millipore). Kinase reactions using recombinant McIdas fragments were performed similarly: 1 μg of each McIdas recombinant protein was incubated with 500 ng recombinant Xenopus PLK4 (PLX4) in the presence of 100 μM $^{32}$P-radiolabeled ATP for 30 min at 30 °C. An irrelevant recombinant protein was used as a negative control. Samples were analyzed by SDS-PAGE followed by autoradiography and Coomassie staining or subjected to tandem mass spectrometry analysis.

## LC-MS/MS analysis and data processing

Recombinant proteins were diluted 1:1 in 100 mM Tris-HCl (pH 8), containing 10 mM TCEP and 40 mM 2-Chloroacetamide to the final concentrations of 10 mM and 40 mM, respectively. Trypsin alone or in combination with AspN was then added to samples which were subsequently incubated for 4 h at 37 °C with agitation (1500 rpm). Next day, peptides were loaded onto SDB-RPS StageTips after acidifying with 10% TFA to ~1% and desalted as described previously (Kulak et al, 2014). Briefly, the StageTips were centrifuged at 1000 × g for washing with 2% ACN/0.2% TFA twice and at 500 × g for elution with 80% ACN/0.1% TFA. The eluate was evaporated to dryness using a vacuum centrifuge and peptides were resuspended in MS loading buffer (2% ACN/0.2% TFA). Equal amount of peptides was subjected to LC-MS/MS analysis.

Peptides were separated on a 50 cm reversed-phase column (75 μm inner diameter, packed in-house with ReproSil-Pur C18-AQ 1.9 μm resin [Dr. Maisch GmbH]) with a binary buffer system of buffer A (0.1% formic acid (FA)) and buffer B (80% acetonitrile plus 0.1% FA) over 60 min gradient (steps: (1) 5–30% of buffer B for 35 min, (2) 30–65% for 5 min, (3) 65–95% for 5 min and (4) washout for 15 min) using the EASY-nano LC 1200 system (Thermo Fisher Scientific) with a flow rate of 300 nL/min. Column temperature was maintained at 60 °C. The nano LC system was coupled to Orbitrap Exploris 480 mass spectrometer (Thermo Fisher Scientific). The instrument is operated in Top12 DDA mode. We acquired full scans (300–1650 *m/z*, maximum injection time 25 ms, resolution 60,000 at 200 *m/z*, charges included 2–5 and dynamic exclusion of 30 ms) at a target of 3e6 ions. The 12 most intense ions were isolated and fragmented with higher-energy collisional dissociation (HCD) (target 1e5 ions, maximum injection time 28 ms, isolation window 1.4 *m/z*, NCE 28%) and detected in the Orbitrap (resolution 15,000 at 200 *m/z*).

Raw MS files were processed within the MaxQuant environment (version 1.6.7.0) with the integrated Andromeda search engine with FDR < 0.01 at the protein and peptide levels (Cox et al, 2014; Cox and Mann, 2008; Cox et al, 2011). We included methionine (M) oxidation and acetylation (protein N-term) as variable and carbamidomethyl (C) as fixed modifications in the search. We allowed up to 2 missed cleavages for tryptic and AspN digestion

and considered peptides with at least six amino acids for identification. "Match between runs" was enabled with a matching time window of 0.7 min to allow the quantification of MS1 features which were not identified in each single measurement. Peptides and proteins were identified using a UniProt FASTA database from *Homo sapiens* (2015) containing 21,051 entries.

## Phosphatase treatment

293 T cells were transfected with vectors expressing either GFP-McIdas WT or the mutant forms GFP-McIdas 10 A and GFP-McIdas 10D. Cell lysates were collected 48 h post-transfection and treated with lambda protein phosphatase for 30 min at 30 °C, according to the manufacturer's instructions (Lambda protein phosphatase, Biolabs). Western blot analysis was then performed using an antibody against GFP (1:1000, mouse anti-GFP, Roche) and goat anti-Mouse IgG-HRP (1:3000, Merck Millipore).

## RNA purification and real-time PCR

Total RNA was isolated from either U2OS cells following McIdas RNAi or from transfected U2OS cells using the Nucleospin RNA II kit (Macherey-Nagel). 1 µg of RNA was converted to cDNA using M-MLV transcriptase (Invitrogen). McIdas, FoxJ1, cMyb, PLK4 and SAS6 mRNA expression levels were assessed by quantitative real-time PCR (Applied Biosystems StepOne), using the Kapa SYBR Fast qPCR kit (KapaBiosystems), according to the manufacturer's instructions. YWHAZ mRNA expression levels were used for normalization. Primer sequences are shown in Table EV2. qPCR data analysis was performed with the REST-MCS beta software.

## Data availability

The datasets produced in this study are available in the following database: LC-MS/MS proteomics data: PRIDE PXD037043.

The source data of this paper are collected in the following database record: biostudies:S-SCDT-10_1038-S44319-026-00697-5.

## Peer review information

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

## Acknowledgements

We thank the Advanced Light Microscopy Facility of the University of Patras for assistance with microscopy. We are grateful to Dr. Pierre Gönczy for kindly providing the SAS6 vectors. We also thank Prof. Constantinos Stathopoulos and Dr. Ilias Skeparnias for their support with kinase assays and for helpful discussions. Additionally, we acknowledge the members of Mónica Bettencourt-Dias's laboratory, especially Catarina Peneda and Dr. Mariana Lince-Faria. We also thank Prof. Pleasantine Mill, Dr. Emma Hall, Dr. Dafni-Eleftheria Pefani, Dr. Ioannis Loukas, Dr. Nibal Badra-Fajardo, Dr. Meletios Verras and the members of Zoi Lygerou's and Stavros Taraviras's laboratories for insightful discussions regarding this work. This work was supported by the Hellenic Foundation for Research and Innovation (HFRI) under grant agreement No. 1302 to MA and No. 2728 to ZL; the project GoMedPrecision (TAEDR-0539180) implemented within the framework of the Action "Actions in interdisciplinary scientific areas with special interests for the connection with the productive fabric", Greece 2.0 - National Recovery and Resilience Plan to ZL; the European Union's Horizon Europe (2021-2027) "ESPERANCE" ERA

Chair program (GA 101087215) and the European Research Council (ERC-Co-683258; CentrioleBirthDeath) to MBD. The publication fees of this manuscript have been financed by the Research Council of the University of Patras.

## Author contributions

**Marina Arbi**: Conceptualization; Data curation; Formal analysis; Supervision; Funding acquisition; Investigation; Visualization; Methodology; Writing—original draft; Project administration; Writing—review and editing. **Margarita Skamnelou**: Validation; Investigation; Methodology; Writing—review and editing. **Lydia Koufoudaki**: Data curation; Investigation. **Vasiliki Bakali**: Data curation; Validation; Investigation; Writing—review and editing. **Spyridoula Bournaka**: Investigation; Writing—review and editing. **Sihem Zitouni**: Investigation; Methodology. **Stavroula Tsaridou**: Investigation; Writing—review and editing. **Ozge Karayel**: Data curation; Formal analysis; Investigation; Visualization; Methodology. **Catherine G Vasilopoulou**: Methodology. **Aikaterini C Tsika**: Investigation; Methodology. **Nikolaos N Giakoumakis**: Investigation; Methodology. **Ourania Preza**: Methodology. **Georgios A Spyroulias**: Resources. **Matthias Mann**: Resources. **Mónica Bettencourt-Dias**: Resources; Supervision; Methodology; Writing—review and editing. **Stavros Taraviras**: Resources; Supervision. **Zoi Lygerou**: Conceptualization; Resources; Supervision; Funding acquisition; Project administration; Writing—review and editing.

Source data underlying figure panels in this paper may have individual authorship assigned. Where available, figure panel/source data authorship is listed in the following database record: biostudies:S-SCDT-10_1038-S44319-026-00697-5.

## Disclosure and competing interests statement

The authors declare no competing interests.

# Expanded View Figures

**Figure EV1.  McIdas localizes to centrosomes.**

(**A**) U2OS cells were immunostained with antibodies against McIdas and Centrin (a distal lumen centriole marker). G1-phase cells were identified as EdU-negative and exhibiting two Centrin dots. (**B**) Quantification of G1-phase centrosomes showing either one (light green) or two (dark green) McIdas dots at centrosomes. Data are from two independent experiments and at least 20 G1-phase centrosomes counted in each experiment. Error bars indicate ± SEM and *P*-values were calculated using a two-tailed Student's t-test: ns, not significant (*P* value: $P = 0.2020$). (**C**) U2OS and hTERT RPE-1 cells were transfected with vectors expressing either GFP-tagged McIdas or GFP alone as a control. Cells were pre-extracted and immunostained with antibodies against GFP (to mark transfected cells), McIdas and Centrin. Arrows indicate centrosomes; asterisks show Centrin accumulation observed upon GFP-McIdas overexpression. Maximum intensity projection images are shown. Protein aggregates observed in GFP-McIdas overexpressing cells are not considered specific. (**D**) HeLa cells stably expressing GFP-tagged McIdas or GFP alone were treated with the proteasome inhibitor MG132 for 4 h, fixed and immunostained with antibodies against GFP, endogenous McIdas and Cep135 (a proximal centriole marker). (**E**) Representative image of a prophase hTERT RPE-1 cell showing duplicated centrosomes stained with antibodies against McIdas, Centrin and Cep164. (**F, G**) Corresponding fluorescence intensity plots of the centrosomal signals shown in (**E**) for McIdas, Cep164 and Centrin. The position and direction of the line used to generate the intensity profiles as a function of distance are indicated in (**E**) by the white line and arrow, respectively. Schematic representation in (**E**) shows McIdas localization relative to the centriole markers, consistent with the intensity profile analysis. White boxes indicate regions shown as higher-magnification images. DNA was stained with Hoechst. Scale bars, 5 µm. ns not significant, AU arbitrary units.

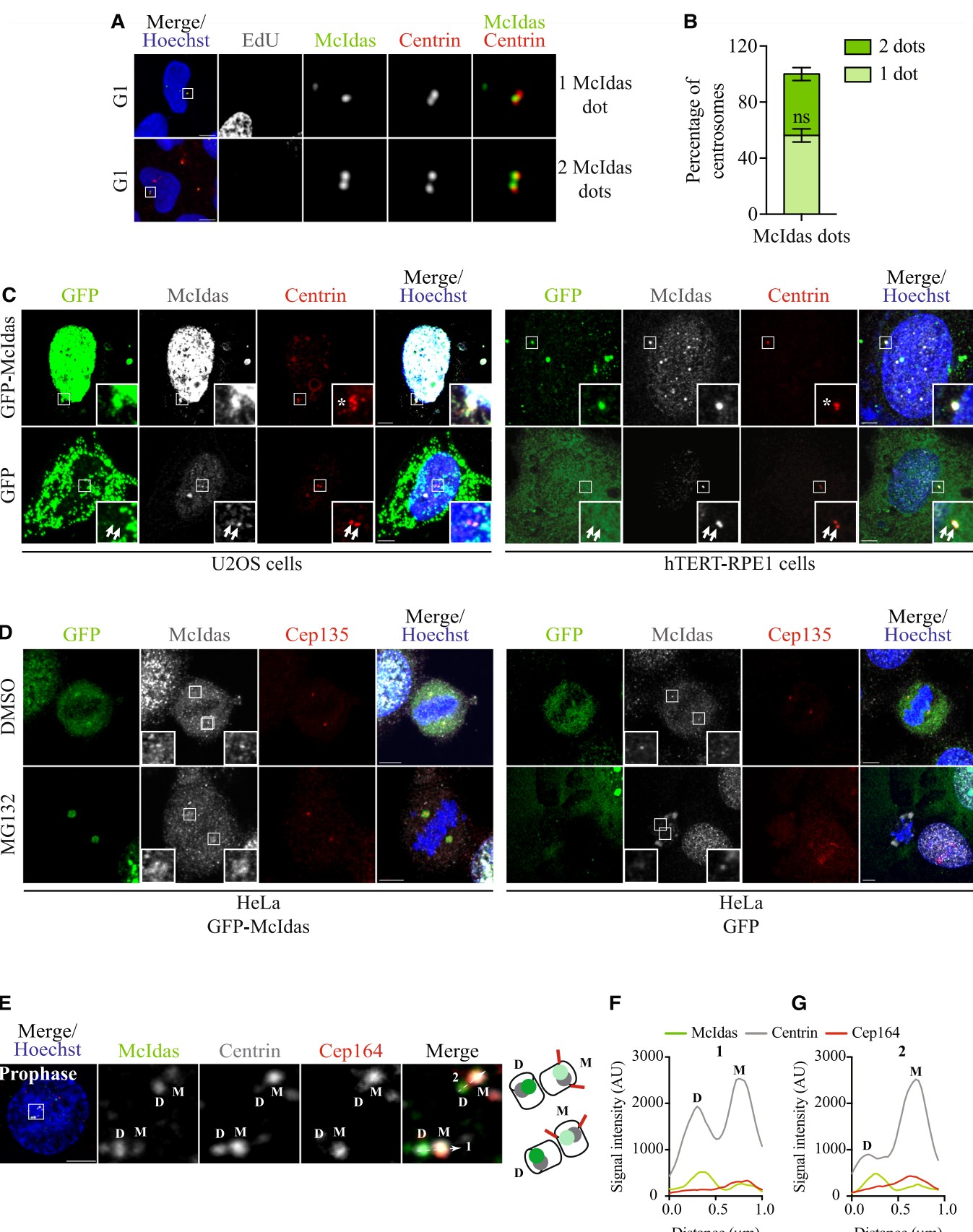

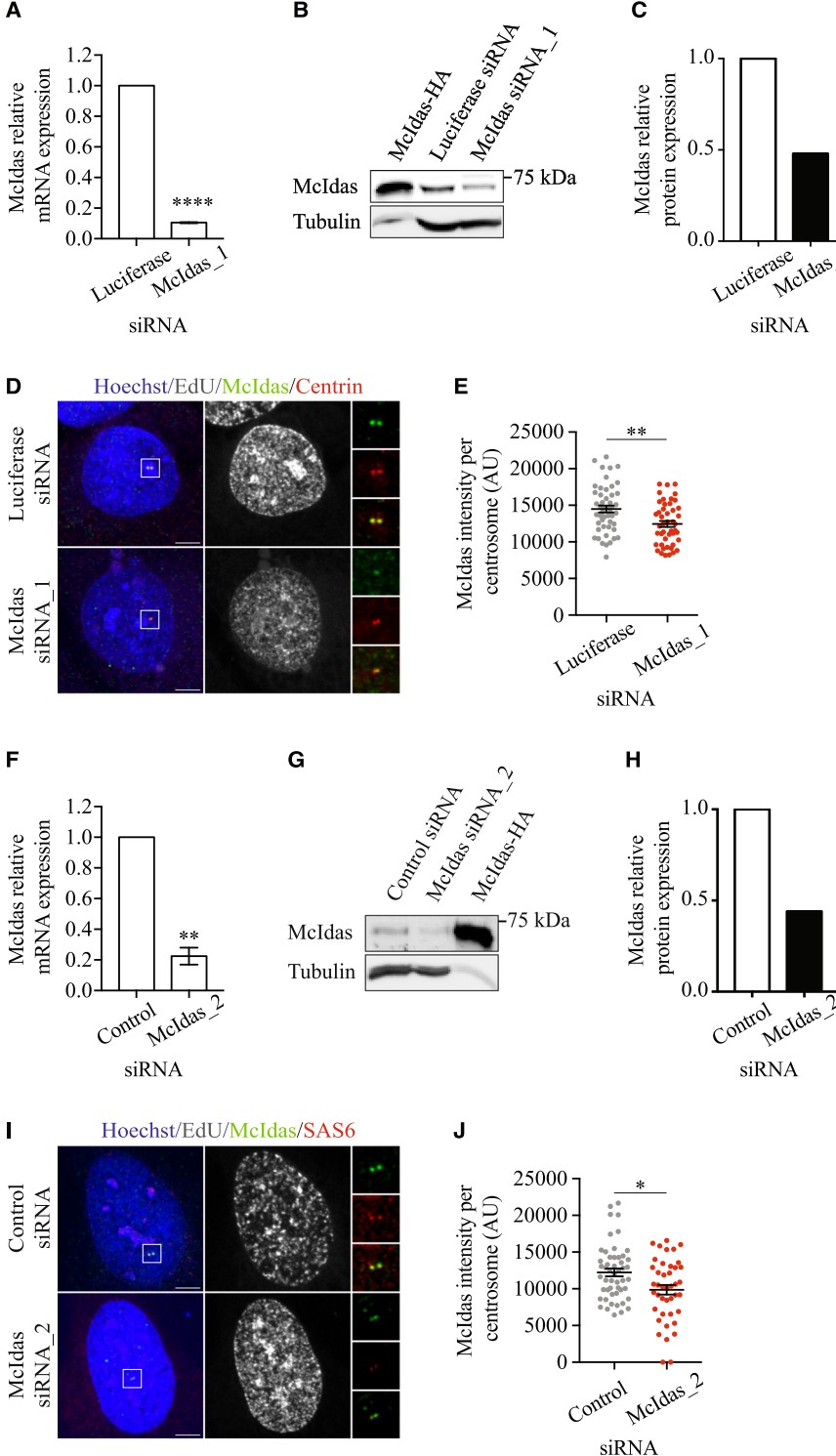

**Figure EV2. Verification of McIdas localization to centrosomes.**

Two different McIdas siRNA oligos (1 and 2) were used to verify the specificity of McIdas localization to centrosomes. U2OS cells were transfected with luciferase or McIdas siRNA_1 (**A–E**) and control or McIdas siRNA_2 (**F–J**). Cells were collected and McIdas mRNA (**A, F**) and protein levels (**B, C** and **G, H**) were assessed by qPCR and western blot analysis, respectively. Cells were fixed and immunostained with antibodies against McIdas (green) and Centrin (red, a distal lumen centriole marker) (**D**) or with antibodies against McIdas (green) and SAS6 (red, a centriole marker) (**I**). McIdas fluorescence intensity at centrosomes (**E, J**) was quantified. For the quantifications of McIdas centrosomal levels at least 40 centrosomes were counted per condition. Two independent experiments were conducted for all data shown. Error bars indicate ± SEM. *P*-values in (**A**) (*P* < 0.0001) & (**F**) (*P* = 0.0068) were calculated using two-tailed Student's t- and *P*-values in (**E**) (*P* = 0.0034) and (**J**) (*P* = 0.0274) were calculated by the nonparametric two-tailed Mann-Whitney test: \**P* < 0.1, \*\**P* < 0.01, \*\*\*\**P* < 0.0001. White boxes indicate regions shown as higher-magnification images. DNA was stained with Hoechst. Scale bars, 5 μm. AU arbitrary units.

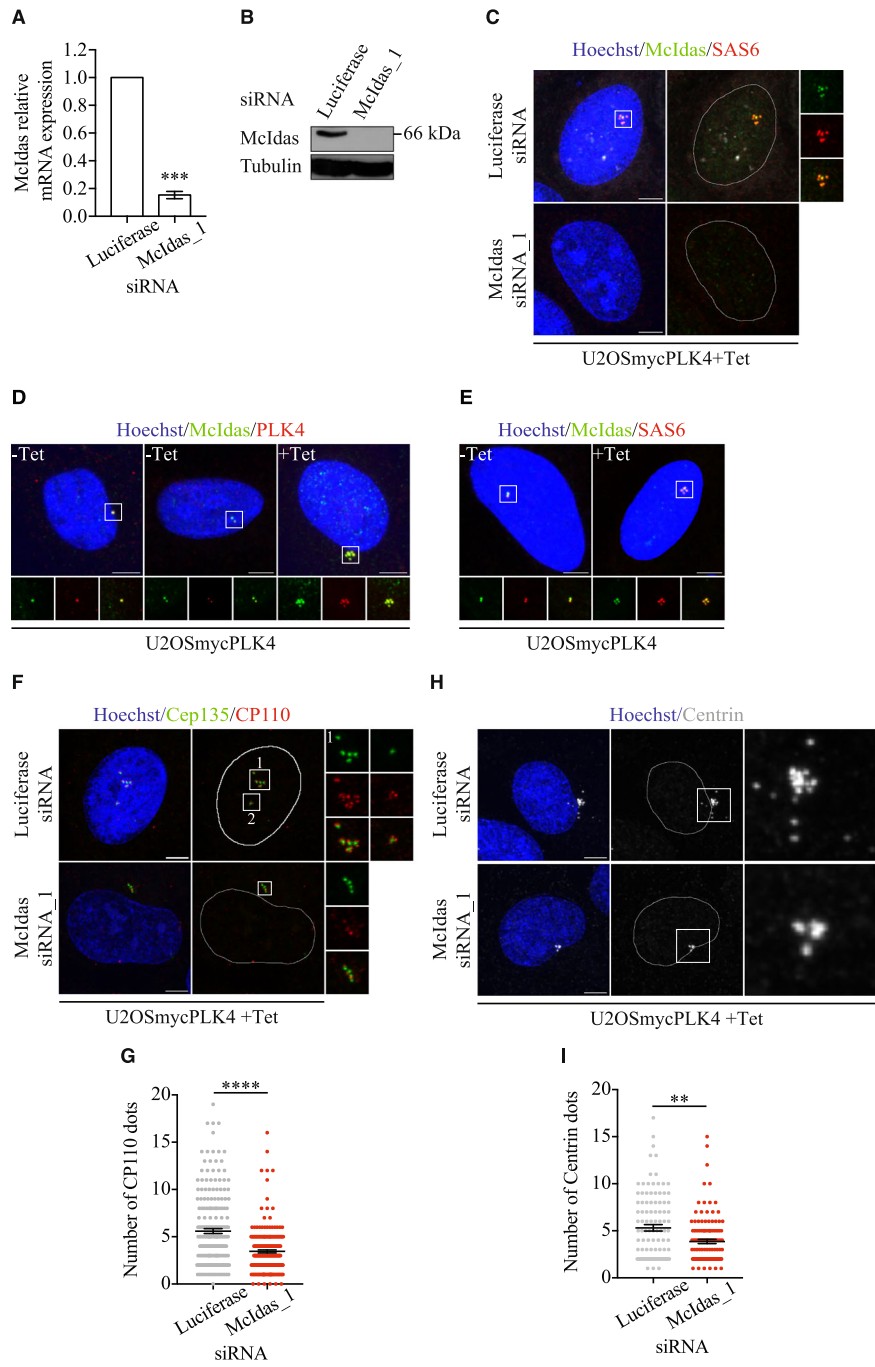

**Figure EV3. McIdas is required for PLK4-induced centriole amplification.**

(A, B) U2OS cells expressing myc-PLK4 under a tetracycline-dependent promoter were transfected with McIdas_1 or control siRNA oligos. A second transfection was performed 24 h later, followed by tetracycline treatment to induce myc-PLK4 overexpression. McIdas mRNA (A) and protein (B) levels were assessed by qPCR and western blot, respectively. *P*-value ($P = 0.0005$) was calculated using two-tailed Student's t-test: ***$P < 0.001$. (C) Cells were fixed and immunostained with antibodies against McIdas and SAS6. (D, E) U2OS cells expressing myc-PLK4 under a tetracycline-dependent promoter were treated with tetracycline or DMSO as a control and immunostained with antibodies against McIdas and PLK4 (D) or SAS6 (E). (F–I) U2OS cells expressing myc-PLK4 under a tetracycline-dependent promoter were transfected with McIdas_1 or control siRNA oligos. A second siRNA transfection was performed 24 h later followed by tetracycline treatment to induce myc-PLK4 overexpression. Cells were fixed and immunostained for CP110 and Cep135 (F) or Centrin (H) to count centriole numbers. (G, I) Quantification of centriole numbers in control and McIdas-depleted cells overexpressing PLK4. The number of CP110 or Centrin dots corresponds to the number of centrioles. Data are from two independent experiments. In each experiment, more than 100 cells were counted per condition. Error bars indicate ± SEM. *P*-values in (G) ($P < 0.0001$) and (I) ($P = 0.0025$) were calculated using the nonparametric two-tailed Mann-Whitney test: **$P < 0.01$, ****$P < 0.0001$. Nuclear boundaries are outlined and white boxes indicate regions shown as higher-magnification images. DNA was stained with Hoechst. Scale bars, 5 μm. Tet tetracycline.

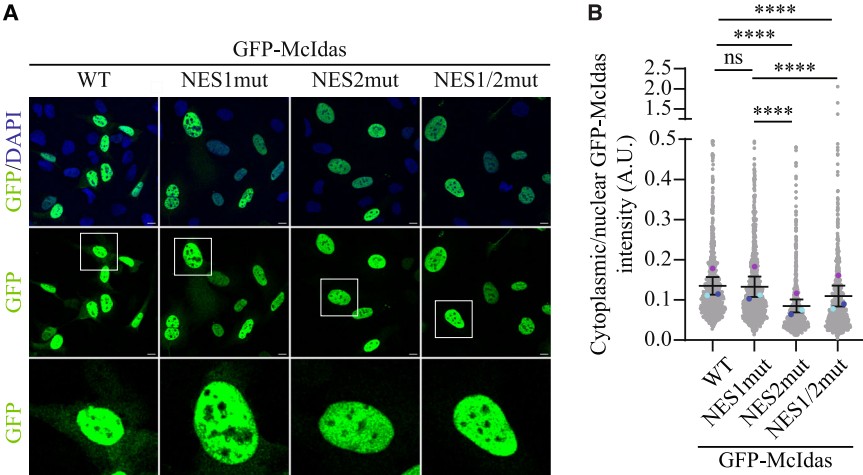

**Figure EV4. McIdas cytoplasmic localization is mediated by a nuclear export signal.**

(A) U2OS cells were transfected with vectors expressing GFP-McIdas WT or the GFP-McIdas NES mutants. After 48 h, cells were fixed and stained for GFP. (B) Quantification of the cytoplasmic to nuclear GFP signal intensity ratio for each condition. Small symbols represent individual cells and large symbols represent the mean values of three independent experiments. Error bars represent ± SEM. At least 100 cells were counted per condition in each experiment. $P$-values were calculated using two-tailed Student's t-test: ****$P < 0.0001$ and ns, not significant ($P$ values: WT-NES1mut, $P = 0.9811$; WT-NES2mut, $P < 0.0001$; WT-NES1/2mut, $P < 0.0001$; NES1mut-NES2mut, $P < 0.0001$ and NES1mut-NES1/2mut, $P < 0.0001$). White boxes indicate regions shown as higher-magnification images. DNA was stained with Dapi. Scale bars, 10 μm. NES Nuclear Export Signal, AU arbitrary units, ns not significant.

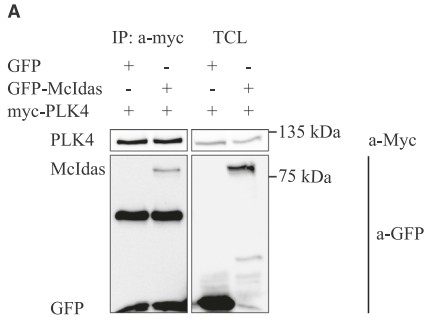

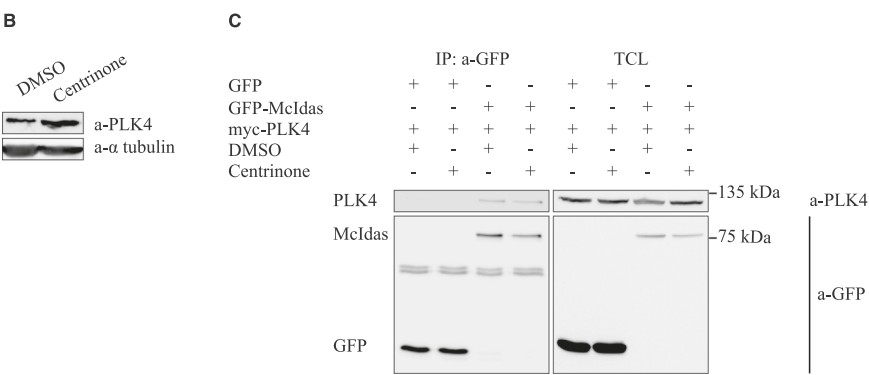

**Figure EV5.  McIdas interacts with PLK4 independently of PLK4 kinase activity.**

(A) U2OS cells expressing myc-PLK4 under a tetracycline-dependent promoter were transfected with either GFP-tagged McIdas or GFP alone as a control. Immunoprecipitation was performed using an antibody against myc, and McIdas was detected in PLK4 immunoprecipitates. (B) U2OS cells expressing myc-PLK4 under a tetracycline-dependent promoter were cultured in the presence of 100 nM centrinone or DMSO as a control. Total cell lysates were analyzed for PLK4 expression. (C) U2OS cells expressing myc-PLK4 under a tetracycline-dependent promoter were transfected with either GFP-tagged McIdas or GFP alone as a control. After 48 h cells were cultured in the presence of 100 nM centrinone or DMSO as a control, and total protein lysates were collected. Immunoprecipitation was performed using an antibody against GFP. Centrinone-mediated inhibition of PLK4 kinase activity did not affect its interaction with McIdas.

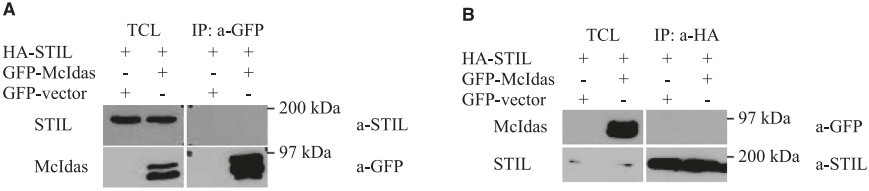

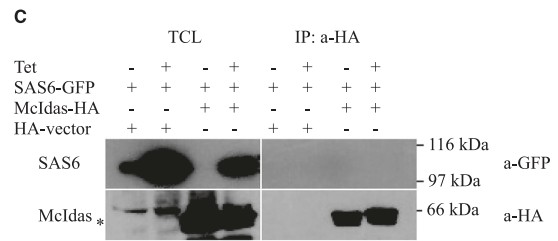

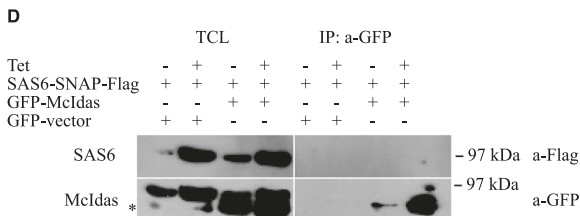

**Figure EV6.  McIdas is unable to interact with STIL and SAS6.**

(A, B) HEK 293 T cells were co-transfected with vectors expressing HA-STIL and GFP-McIdas or GFP alone as a control. Cell extracts were collected and immunoprecipitation was performed using anti-GFP (A) or anti-HA (B) antibodies. STIL was not detected in McIdas immunoprecipitates and vice versa. (C) HEK 293 T cells were co-transfected with vectors expressing SAS6-GFP under a tetracycline-dependent promoter and either McIdas-HA or HA alone as a control. Cell extracts were collected and immunoprecipitation was performed using an anti-HA antibody. (D) HEK 293 T cells were co-transfected with vectors expressing SAS6-SNAP-Flag under a tetracycline-dependent promoter and either GFP-McIdas or GFP alone as a control. Cell extracts were collected and immunoprecipitation was performed using an antibody against GFP. No detectable interaction between McIdas and SAS6 was observed. Asterisks indicate the specific bands in each experiment. TCL total cell extract, IP immunoprecipitation.

                                                                                              

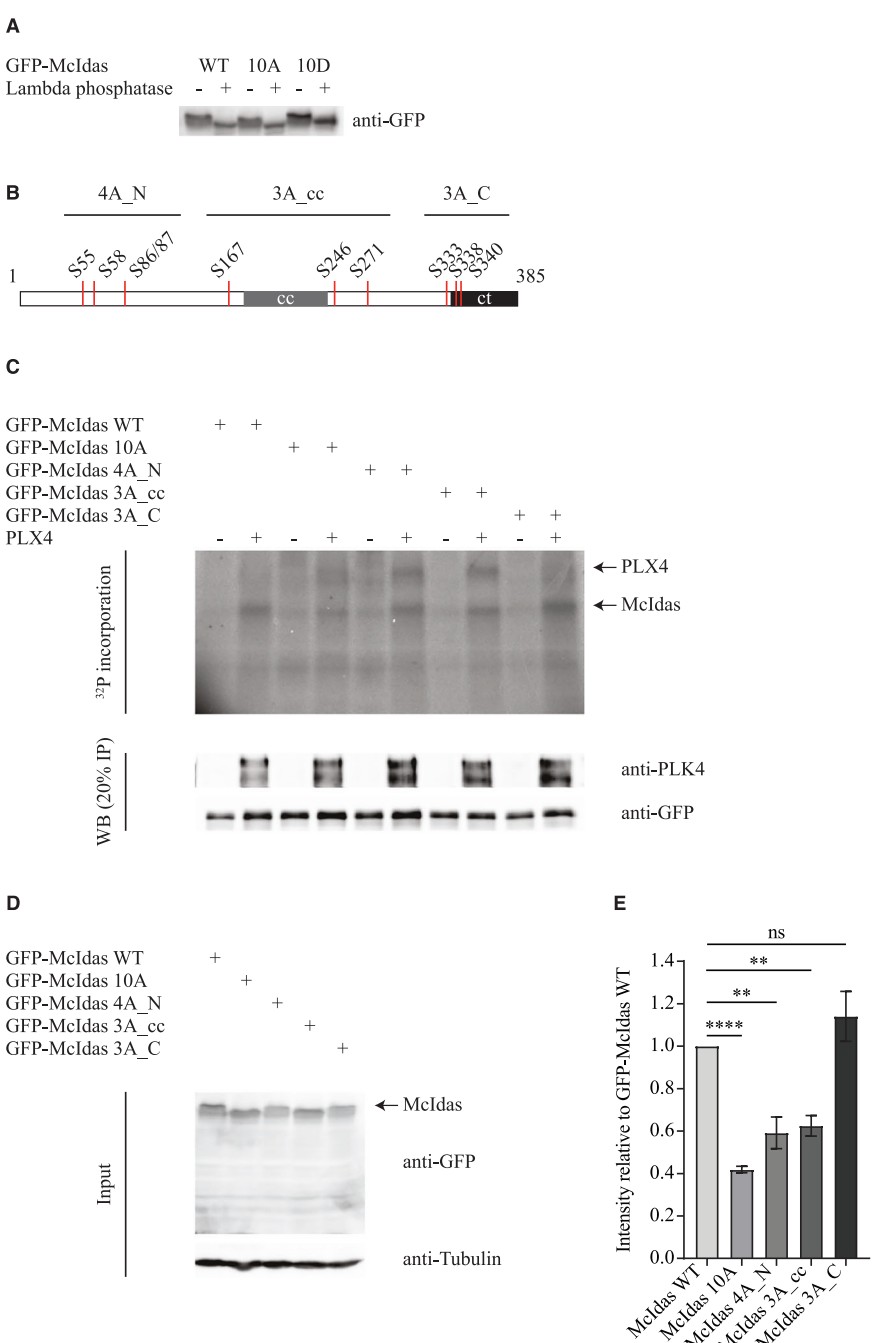

**Figure EV7. PLK4-dependent phosphorylation of McIdas is enriched in its N-terminal and coiled coil-containing regions.**

(A) HEK 293 T cells were transfected with vectors expressing GFP-McIdas WT or the mutants 10 A and 10D. Cell lysates were collected and incubated with lambda protein phosphatase. Western blot analysis was followed with an antibody against GFP. (B) Schematic representation of the PLK4-phosphorylated serine residues identified on the McIdas protein and the partial McIdas phospho-dead mutants generated. (C) In vitro kinase assays were performed using immunoprecipitated GFP-McIdas WT, GFP-McIdas 10 A or the indicated partial phospho-dead mutants. Each IP was incubated with recombinant Xenopus PLK4 (PLX4) and [γ-$^{32}$P] ATP. Samples were analyzed by autoradiography. Western blot analysis of 20% of each sample was performed using antibodies against PLK4 and GFP to verify protein loading. (D) Western blot analysis of total cell extracts used in (C) was performed with antibodies against GFP and α-tubulin. (E) Quantification of McIdas WT and mutant phosphorylation (from C) was performed using Image Lab software. Background signal in the control condition (no PLK4) was subtracted from the final intensity values. Values indicate phosphorylation signal intensity relative to McIdas WT (set as 1). Data are the mean values of three independent experiments and error bars indicate ± SEM. *P*-values were calculated using a two-tailed Student's t-test: **$P < 0.01$, ****$P < 0.0001$ and ns, not significant (*P* values: WT-10A, $P < 0.0001$; WT-4A_N, $P = 0.0054$; WT-3A_cc, $P = 0.0015$ and WT-3A_C: $P = 0.2988$).

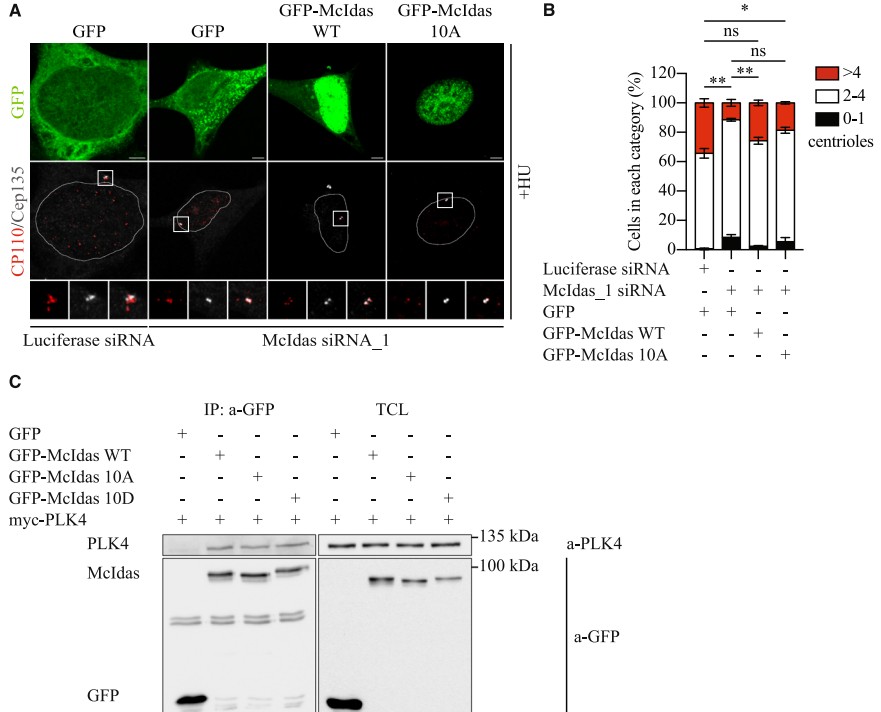

**Figure EV8. PLK4-specific phosphorylation of McIdas is essential for daughter centriole biogenesis but is independent of their interaction.**

(A) U2OS cells were transfected twice with McIdas siRNA_1 or control siRNA oligos. 24 h after the second siRNA transfection, cells were transfected with vectors expressing GFP-McIdas WT, GFP-McIdas 10A, or GFP alone as a control, followed by treatment with HU for an additional 48 h. Cells were fixed, and centriole numbers per cell were determined by staining with antibodies against Cep135 and CP110. (B) Quantification of centriole numbers per cell. The number of CP110 dots corresponds to the number of centrioles. Part of the immunofluorescence images in (A) and the corresponding quantification in (B) were also used in Fig. 2E and F, respectively, as part of the same experiment. Data are presented as the mean values of three independent experiments for GFP and GFP-McIdas WT or two for GFP-McIdas 10A and error bars indicate ± SEM. In each experiment at least 100 cells were counted per condition. Statistical values shown refer to the >4 centrioles category. *P*-values were calculated using two-tailed Student's t-test: *P <0.1, **P <0.01 (P values: Luciferase siRNA/GFP-McIdas siRNA-GFP, P = 0.0035; McIdas siRNA/GFP-McIdas siRNA/McIdas WT, P = 0.0087; Luciferase siRNA/GFP-McIdas siRNA/McIdas WT, P = 0.0673, Luciferase siRNA/GFP-McIdas siRNA/McIdas 10 A, P = 0.0247 and McIdas siRNA/GFP-McIdas siRNA/McIdas 10 A, P = 0.0637. (C) GFP-tagged McIdas WT, GFP-McIdas 10 A, GFP-McIdas 10D, or GFP alone as a control were transfected into an inducible U2OS myc-PLK4 overexpressing cell line. Immunoprecipitation was performed using an antibody against GFP, and PLK4 was detected in immunoprecipitates of both McIdas WT and mutant proteins. Nuclear boundaries are outlined and white boxes indicate regions shown as higher-magnification images. Scale bars, 5 μm. HU hydroxyurea, ns not significant, TCL total cell extract, IP immunoprecipitation.

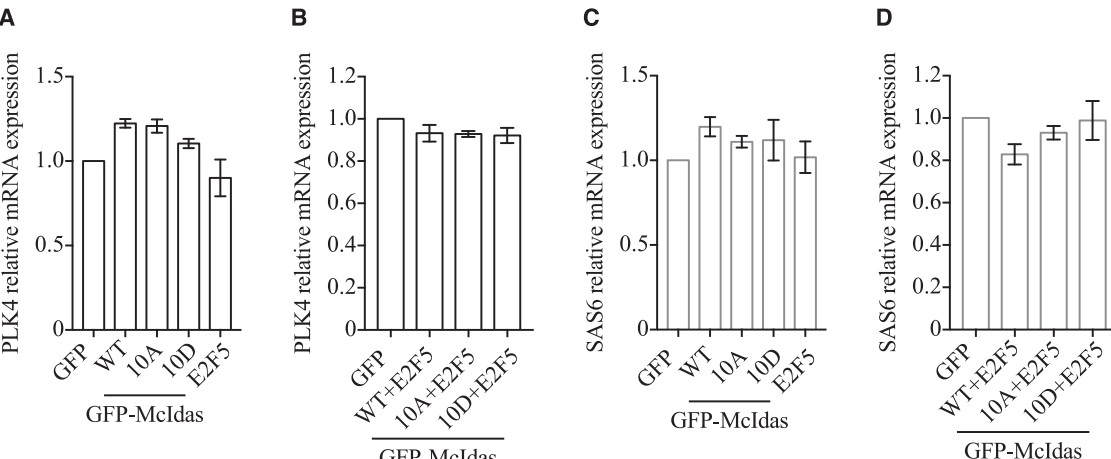

**Figure EV9. PLK4 and SAS6 are not transcriptionally activated by McIdas.**

(A, B) U2OS cells were transfected with vectors expressing the indicated genes and PLK4 mRNA expression was analyzed by quantitative real-time PCR (qPCR). (C, D) U2OS cells were transfected with the indicated genes and SAS6 mRNA expression was analyzed by quantitative real-time PCR (qPCR). Data are the mean values of at least three independent experiments and error bars indicate ± SEM.

