## [Peer Review File · EMBO Reports]

Mcdas localizes to centrioles and controls centriole numbers through PLK4-dependent phosphorylation

Marina Arbi, Margarita Skamnelou, Lydia Koufoudaki, Vasiliki Bakali, Spyridoula Bournaka, Sihem Zitouni, Stavroula Tsaridou, Ozge Karayel, Catherine Vasilopoulou, Aikaterini Tsika, Nikolaos Giakoumakis, Ourania Preza, Georgios Spyroulias, Matthias Mann, Monica Bettencourt-Dias, Stavros Taraviras, and Zoi Lygerou

Corresponding author(s): Marina Arbi (marmpi@upatras.gr)

Review Timeline:

Submission Date:	8th Nov 22
Editorial Decision:	1st Dec 22
Appeal Received:	4th Aug 25
Editorial Decision:	6th Oct 25
Revision Received:	22nd Dec 25
Accepted:	15th Jan 26

Editor: Ioannis Papaioannou / Esther Schnapp

Transaction Report:

Dear Dr. Arbi,

Thank you for submitting your research manuscript for consideration by EMBO reports. It has now been seen by three experts in the field, and their detailed reports are appended below.

As you will see, the referees acknowledge that the study is potentially interesting, but none of them are very positive about the manuscript as it stands. They all raise several significant concerns regarding methodological limitations, the experimental design, and interpretation of the findings. They point out that important controls, rescue experiments, and image quantification are missing, and they mention that the main conclusions of the study are not convincingly supported by the available data. All reviewers call for a considerable amount of additional experimentation to resolve these issues. Due to the many criticisms and technical problems of the current version, and the amount of work likely to be required to address them, I am afraid that we do not feel it would be productive to call for a revised version of your manuscript.

Given the potential interest of the findings, we would, however, have no objection to consider a new manuscript on the same topic if at some time in the future you obtained data that would considerably strengthen the study and address the referees' concerns in full. If you were to send a new manuscript that would entirely address the concerns put forward by the referees, it would be treated as a new submission rather than a revision, and it would be reviewed afresh, also with respect to any novel literature on the topic at the time of submission.

I am sorry to disappoint you on this occasion, but I nevertheless hope that you will find the referees' comments and suggestions helpful in your work. I would like to thank you once again for your interest in our journal and for the opportunity to consider your manuscript.

Yours sincerely

Ioannis Papaioannou, PhD
Editor
EMBO reports

Referee #1:

The manuscript by Arbi et al. describes a novel localization for the Mcidas protein to centrioles. The authors show that Mcidas localizes to centrioles in a cell cycle variable way and they perform nice high resolution measurements of the specific centriolar localization and find that it is in the middle of the centriole between Cep164 and Centrin/Cep135. They go on to show that in cells overexpressing (OE) Mcidas there are supernumerary centrioles and cells deficient for Mcidas there are fewer cells with supernumerary centrioles. They go on to propose that Mcidas physically interacts with Plk4 and that this interaction leads to phosphorylation of Mcidas by Plk4 at multiple sites. They test the functional consequences of these phosphorylation events and show broadly and show that the pan-phosphomimetic and non-phosphorylatable versions of Mcidas are both dysfunctional. In general, I have a significant number of issues with this paper and feel that the overall quality and rigor of the data is not up to the standards of EMBO. While I have no trouble believing that Mcidas could have multiple functions, this paper is written in a very deceptive way that essentially ignores everything that is known about this protein, even stuff published from this lab. Mcidas is well characterized to bind transcription factors and modulate transcription. While this is appropriately integrated in the introduction and discussion it is ignored in the experimental design and interpretation, which is very problematic. In fact this group showed that it binds geminin and is required for geminin nuclear localization. Yet the nuclear localization is essentially completely ignored (and not even observed with their some of their current reagents). Mcidas has been reported to drive the conversion of cells in culture into multiciliated cells and while that is much improved when coupled with E2F4 it is certainly possible and I would say likely that the over amplification seen here is related to this known phenotype rather than its localization to centrioles. Furthermore, if the function is indeed non-nuclear then there should be some interesting regulation of its nuclear to centriole shift in localization. All of these important possibilities are not considered in their experimental design. If there is indeed a distinct non-nuclear function for Mcidas, that would be interesting, but it would need to be shown in the context of blocking the nuclear function. The non-specificity of the phosphorylation data is very suggestive of something that is not in fact regulatory, which appears to be consistent with the functional data. For this to be meaningful the specific sites would need to be mapped (shown to be lost in Plk4 deficient cells) and shown to be important for function.

Comments:

The data quantification in general is very subjective (e.g. 4C). It is not clear what dictates a foci. There are clearly numerous Sas6 foci on both of these examples that are not considered centrioles. Some discussion of what qualifies as a centriole in their

analysis should be included.

Numerous groups, including this one have reported nuclear localization of Mcdas and this group previously reported that Mcdas is required for Geminin nuclear localization. Yet in their analysis and provided images there appears to be no nuclear localization of Mcdas throughout the cell cycle. This draws some concern regarding their reagents.

"Accumulation of Centrin was evident upon GFP-Mcdas overexpression in U2OS and hTERT-RPE1 cells, suggestive of a possible role of Mcdas in centriole formation (Supplementary Figure 1A & B, marked by an asterisk)." Mcdas is massively overexpressed and the centrin accumulation is not convincing as the foci are all different sizes.

In many of the supplemental figures overexpression of Mcdas leads to numerous aggregates which is not addressed.

Throughout the paper the authors use the term expression when in fact that are discussion localization. These two words are not interchangeable and all uses of expression that refer to protein localization should be changed.

The authors present evidence that in G1 Mcdas sometimes localizes to one centriole and sometimes both. This seems a bit odd and it would be nice to know the relative percentage of these events.

While IPs with overexpressed tagged proteins is never optimal, at the very least the IP should be done in reciprocal experiments.

There is no reference to Figure 5C in the text.

Figure 6E should be represented as relative intensity (to a control) which should allow for the data to be presented from all three replicates rather than a "representative" quantification which is essentially meaningless when we can see it in Figure 6C.

Minor: In the future please label your figures for the reviewers. Without numbers it was difficult to know which figure was which (especially the supplemental)

Referee #2:

Mcdas is known to be a nuclear protein, and its connection to the cell cycle regulator Geminin has been well established. Upon binding to Geminin, it inhibits the association between Geminin and Cdt1 and modulates DNA replication licensing. In this study by Marina Arb and co-workers, a new function of Mcdas has been demonstrated. A new localization of Mcdas has been identified at the middle region of centrioles and the distribution of the protein is also changed during the cell cycle. Overexpression or deletion of Mcdas significantly impacts centriole biogenesis. The authors also detected the phosphorylation of Mcdas by PLK4 at multiple sites. These PTM are critical for its role on centriole number control.

This is an interesting study that describes Mcdas as new PLK4 target. Interaction of Mcdas and PLK4 and phosphorylation of Mcdas by PLK4 in vitro are convincingly demonstrated. The localization of Mcdas with centrioles and the function of Mcdas at centrioles is less convincing and needs further validation and clarification. Many "functional" experiments are based on "artificial" setups: HU block and Mcdas or PLK4 overexpression. In addition, the authors deplete Mcdas using siRNA and show a defect in centriole duplication based on IF staining of Centrin and Sas6 as markers (Fig. 4), however, because of the missing rescue experiment this approach lacks an important control. Whether Mcdas phosphorylation by PLK4 has a role in centrosome duplication remains unclear since 10A and 10D mutants behave identical in Fig. 7. The 10-point mutations might inactivate the protein (do they localize to centrosomes? - unclear in Fig. 7A) or a phosphorylation cycle of Mcdas is necessary as discussed by the authors. Analyzing Mcdas mutant versions with less mutations is advisable.

In summary, following key improvements are needed:

1. Show Mcdas antibody specificity convincingly
2. Perform siRNA rescue experiment and confirm centriole duplication defect
3. Provide data for the function of Mcdas phosphorylation by PLK4.

Concerns and suggestions:

- The localization data would benefit from quantification (Fig. 1A). I am concerned about Fig. S1E: Mcdas signal intensity at centrosomes was only mildly affected by siRNA depletion of Mcdas (images are not shown - why?). However, depletion works very well when analyzed by IB (Fig. S2). How does this fit together? This raises concerns about the specificity of the Mcdas antibody and the validity of this study. It is very important that the authors resolve this discrepancy and show convincingly that the centriole signals observed with the Mcdas antibody reflect the localization of this protein.
- I am concerned about the image quality in Fig. 2E.

- On p. 6 the authors talk about that MclDas expression is significantly increased at procentrioles during early mitosis (Fig. 2D). How can they make such a conclusion without quantification of the signal and statistics? Furthermore, there is a certain discrepancy between Fig. 2E and Fig. 2J. According to Fig. 2E, MclDas in G2 is at the newly synthesized centriole close to Sas6 but not at the distal part of the mother centriole. Fig. 2J, S/G2 cells shows something different: MclDas is at the mother centriole and co-localizes with the entire daughter.
- Fig. 4: the authors should perform a rescue experiment with a siRNA resistant MclDas construct.
- When MclDas siRNA was applied to different cell lines, apart from the confirmation by qPCR (Fig. S2A, S2C, Fig. 3, Fig. 4), IB and IF should also be carried out to check the protein level and localization.
- What will happen when MclDas is overexpressed? Would it induce amplification of centrioles?
- Fig 4A and 4C: Since High background was detected especially for SAS-6, at least another centrosome marker should be used to confirm the centriole numbers (or EM analysis).

The minor comments:

- Start a new paragraph on p. 3: PLK4 (Polo-like kinase 4), STIL () ... Same on p.3: The expression levels and the interactions Same on p. 7: Given that PLK4 overexpression induces ... (by the way: genes are overexpressed, therefore PLK4 is italic). Same p. 9: Next, we mutated these 10 serine residues
- A statistical analysis should be carried out to determine whether the differences observed are significant (Fig. S1D and F and Fig. S2C).
- p. 8: "highlighting that MclDas is important for centrosomal Sas6 recruitment." I would say: MclDas is important for PLK4 induced centrosome amplification.
- Fig. 5E, Fig. 6E and Fig. S4E: it would be nice to show the average values from different repeats and show the standard deviation of each value.
- The length and the thickness of the scale bar should be standardized (Fig. 2E and 2J)
- Fig 3B and 3D: under GFP overexpressing, about 34% of cells contain more than four centrioles due to HU treatment. In siRNA control cells, almost half of them had the same phenotype. The authors should explain the different frequencies in both control experiments, which are supposed to be comparable.
- Fig 5A: additional unlabeled bands appear in the IB. The authors should clarify the identity of these bands.
- Fig 5E: the title of y axis seems wrong; it should be 1-179.
- Fig S4C and S4E: the quantification method could be mentioned in the legend of this figure, because relatively high background signal could be observed from non-PLX4 treated samples.
- Fig 7B: the authors should check the expression level of different constructs, i.e., wt, 10A and 10D. The constructs may have different transfection efficiency, different expression level or different stability in cells.

Referee #3:

Summary:

The authors claim that MclDas localizes to the centrosome where it regulates centriole duplication. While the presented data indeed suggests MclDas may have functions at the centrosome, this work comes short of supporting the main claim. Although the addressed question is of general interest for the molecular biology community, the work required to adequately back the main conclusions is beyond what can be expected from significant revision. This work is therefore un-suited for publication in EMBO Reports.

Major shortcomings of the manuscript:

The first main claim of the study is that MclDas localizes to centrosomes, which the authors support by showing images of immunolabelling experiments and MclDas over-expression (Figure1). A key point to back this claim, is to demonstrate specific antibody detection of endogenous MclDas at the centrosome. Although the authors state they have confirmed this point by siRNA depletion of MclDas, the presented data isn't convincing. The first issue is that MclDas RNA levels are shown, but not protein levels by western blot. The second issue is that no images of centrosomes stained with MclDas antibody are shown for this experiment, and the quantification only shows a slight reduction in MclDas centrosomal signal. This leads to multiple important questions: is it that the siRNA does not deplete the protein ? Or is the antibody staining centrosomes in a non-specific manner ? How are the centrosomal levels quantified ? The authors later claim that centrosomal MclDas localization varies during the cell-cycle. It would therefore also be important to assess how MclDas siRNA affects cell-cycle progression here. In regard of all these questions, it remains unconvincing that MclDas is indeed a centriole-associated protein.

The second main claim of the manuscript is that MclDas regulates centriole duplication. The authors aim to demonstrate this in different contexts, but the presented data is not convincing. To claim MclDas is involved in centriole biogenesis, the main point is to demonstrate that changing MclDas levels affects centriole numbers in unperturbed cycling cells. This is the point the authors aim to make in Figure 4. They deplete MclDas by siRNA, and then count centrioles specifically in EdU positive S-phase cells,

using Centrin or SAS6 to detect all centrioles or nascent centrioles respectively. In both case, the immunostainings show lots of what seems to be non-specific signal, raising the question of how the authors decided to count a spot as a centriole or not. Colocalisation of multiple centriolar markers would be necessary to confirm centriolar staining. Additionally, if Mcdas does regulate centriole biogenesis, the expectation is that centrosome numbers should be affected in siRNA depletion experiments or Mcdas OE experiments. No data throughout the paper convincingly shows this, although the authors do state that cell lines over-expressing GFP-McIDAs may have more centrioles, but without providing convincing images or quantifications (Supplementary Figure 1A/B).

As Mcdas involvement in centriole biogenesis in unperturbed cells seems hard to demonstrate, the authors aimed to prove its involvement in two different contexts in which centrioles are over-duplicated: HU-induced S-phase lengthening, and PLK4 over-expression. Here, many points weaken the claim that Mcdas affects centriole over-duplication, which is tested by either over-expressing or down-regulating Mcdas. First, for both mechanisms of over-duplication, cell-cycle progression could have a strong impact on levels of over-duplication. While the authors state (but without showing the data) that Mcdas over-expression does not affect cell-cycle progression upon HU treatment, the impact on cell-cycle progression of Mcdas siRNA is never assessed, making conclusions hard to take. Second, the effect-size of the perturbations (either centriole gain or loss), is always very mild, and sometimes even milder than control perturbations. As an example in the HU induced overduplication, the levels of over-duplication are already different between GFP over-expression and Luciferase siRNA which are two control perturbations. Finally, the choice of showing in some cases the frequency of cells with a given centriole number, and in some others the number of centrioles per cells, only makes conclusions even harder to take. Together, the presented data does not convincingly support that Mcdas is involved in centriole biogenesis regulation.

The fact that the data comes short of supporting the manuscript's main claim is the principal reason for not recommending this paper is published in EMBO reports. Additionally, other experiments presented in this manuscript also raise doubts as to the soundness of the conclusions taken. In particular, the study of Mcdas centriolar localization dynamics is weakened by images that show varying quality of immunostainings. Furthermore, the study of PLK4 and Mcdas molecular interactions also lack conclusiveness and physiological relevance. Although the immunoprecipitation of over-expressed tagged proteins suggest possible interaction, and in vitro phosphorylation assays suggest possible phosphorylation, these observations would require additional controls (IP in the other direction ? Effect of the commercially available specific Plk4 kinase inhibitor centrinone ?). Furthermore, the physiological relevance of this phosphorylation suggested in the title and in the main text claims from Figure 7, are again not backed by convincing data showing centriole biogenesis modifications.

** As a service to authors, EMBO Press provides authors with the ability to transfer a manuscript that one journal cannot offer to publish to another journal, without the author having to upload the manuscript data again. To transfer your manuscript to another EMBO Press journal using this service, please click on Link Not Available

Response to reviewers' comments

Please find below a point-by-point responses to the comments raised by the reviewers. In line with their suggestions, we have performed several additional experiments, which are now included in the revised manuscript. These new experiments address all the key points raised, including a rescue experiment demonstrating the direct role of McIdas and its phosphorylation by PLK4 in centriole biogenesis, an analysis of the specificity of the McIdas antibody and siRNA oligos, and a more comprehensive characterization of McIdas localization at centrioles. One of the most significant findings in our revised manuscript is the identification and characterization of a novel nuclear export signal in the McIdas protein that is critical for its centrosomal localization and function in centriole biogenesis. Furthermore, additional experiments separating the centrosomal role of McIdas from its previously described transcriptional role revealed that McIdas can regulate centriole numbers independently of its transcriptional activity. Taken together, these findings strongly support our conclusion that McIdas localizes to centrosomes and directly impacts centriole duplication, suggesting that McIdas may serve as an important link between the cell cycle and multiciliogenesis. We would like to thank the reviewers for their comments and suggestions, which we believe have significantly strengthened our study.

Answer to Reviewer 1 comments:

The manuscript by Arbi et al. describes a novel localization for the Mcidas protein to centrioles. The authors show that Mcidas localizes to centrioles in a cell cycle variable way and they perform nice high resolution measurements of the specific centriolar localization and find that it is in the middle of the centriole between Cep164 and Centrin/Cep135. They go on to show that in cells overexpressing (OE) Mcidas there are supernumerary centrioles and cells deficient for Mcidas there are fewer cells with supernumerary centrioles. They go on to propose that Mcidas physically interacts with Plk4 and that this interaction leads to phosphorylation of Mcidas by Plk4 at multiple sites. They test the functional consequences of these phosphorylation events and show broadly that the pan-phosphomimetic and non-phosphorylatable versions of Mcidas are both dysfunctional. In general, I have a significant number of issues with this paper and feel that the overall quality and rigor of the data is not up to the standards of EMBO. While I have no trouble believing that Mcidas could have multiple functions, this paper is

written in a very deceptive way that essentially ignores everything that is known about this protein, even stuff published from this lab.

Mcidas is well characterized to bind transcription factors and modulate transcription. While this is appropriately integrated in the introduction and discussion it is ignored in the experimental design and interpretation, which is very problematic. In fact this group showed that it binds geminin and is required for geminin nuclear localization. Yet the nuclear localization is essentially completely ignored (and not even observed with their some of their current reagents). Mcidas has been reported to drive the conversion of cells in culture into multiciliated cells and while that is much improved when coupled with E2F4 it is certainly possible and I would say likely that the over amplification seen here is related to this known phenotype rather than its localization to centrioles. Furthermore, if the function is indeed non-nuclear than there should be some interesting regulation of its nuclear to centriole shift in localization. All of these important possibilities are not considered in their experimental design. If there is indeed a distinct non-nuclear function for Mcidas, that would be interesting, but it would need to be shown in the context of blocking the nuclear function.

The non-specificity of the phosphorylation data is very suggestive of something that is not in fact regulatory, which appears to be consistent with the functional data. For this to be meaningful the specific sites would need to be mapped (shown to be lost in Plk4 deficient cells) and shown to be important for function.

We would like to thank Reviewer 1 for the helpful and detailed comments. One of the main concerns raised was the possibility that the centriole amplification observed in our studies could be attributed to the known nuclear/transcriptional role of McIdas. Indeed, McIdas has previously been shown to form a complex with E2F4/5 and DP1 transcription factors to upregulate genes important for multiciliogenesis, including cMyb and FoxJ1. To address this issue, we performed new experiments to separate the transcriptional and centrosomal functions of McIdas. We overexpressed GFP-McIdas WT in U2OS cells and as expected we showed that it upregulates cMyb and FoxJ1 mRNA levels, an effect enhanced by co-expression of E2F5 (revised Figure 7D-G) and does not upregulate PLK4 and SAS6 (revised Supplementary Figure 9). Importantly, we also examined the GFP-McIdas 10A phospho-dead mutant, which is impaired in inducing centriole amplification. We found that this mutant still retains transcriptional activity and is capable of upregulating cMyb and FoxJ1 mRNA expression, especially when co-expressed with E2F5. These findings show that McIdas transcriptional

activity is intact even when centriole amplification is impaired, indicating that McIDAS role in centriole biogenesis is independent of its transcriptional activity. Furthermore, we identified a putative Nuclear Export Signal on McIDAS, which is required for its cytoplasmic and centriolar localization, as well as its ability to induce centriole amplification. We have now included these results in our revised manuscript, which we believe provide strong support for a distinct cytoplasmic role of McIDAS in regulating centriole number. We have revised the manuscript throughout, to distinguish this newly described centrosomal localization and function from the known, nuclear localization and function of McIDAS.

Please find below our point-by-point responses to all the comments raised by Reviewer 1.

Comments

1. The data quantification in general is very subjective (e.g. 4C). It is not clear what dictates a foci. There are clearly numerous Sas6 foci on both of these examples that are not considered centrioles. Some discussion of what qualifies as a centriole in their analysis should be included.

Following the reviewer's suggestion, we strengthened our analysis in Figure 3 by performing double immunostaining on centrosomes from control and McIDAS-depleted asynchronous U2OS cells. In addition, we included a detailed description in our manuscript of the criteria used to define centrioles in our analysis. Specifically, we used antibodies against SAS6 and Cep170, a centriole component of the subdistal appendages, to examine SAS6 recruitment at centrosomes. Foci were defined as centrioles only if they were positive for both markers. We focused on S-phase cells and quantified the number of SAS6 foci on Cep170-positive centrosomes. Consistent with our initial findings, we observed a significant increase in the number of cells with 0 or 1 SAS6 focus in McIDAS-depleted centrosomes, compared to control S-phase cells, where the majority exhibited two SAS6 foci (Figure 3C & D).

Moreover, the same approach was used in Figure 3A & B to quantify centriole numbers based on centrin foci, along with Cep192, a structural component of the pericentriolar material (PCM) (see also response to Reviewer 2 below).

2. Numerous groups, including this one have reported nuclear localization of McIDAS and this group previously reported that McIDAS is required for Geminin nuclear localization. Yet in their analysis and provided images there appears to be no nuclear

localization of McIdas throughout the cell cycle. This draws some concern regarding their reagents.

In our previous work (Arbi *et al*, 2016; Pefani *et al*, 2011), and consistent with the work from other labs, we demonstrated that McIdas is a nuclear protein, which co-operates with transcription factors to drive gene expression programs essential for multiciliate cell differentiation. In those studies, we used PFA fixation and observed endogenous and ectopically expressed McIdas localization in the nucleus. In the current study, we used the same, previously validated antibody, but instead used methanol fixation throughout our experiments to better preserve centriolar structures and assess McIdas localization at centrosomes. Methanol fixation is not optimal for preserving nucleoplasmic proteins and it is thus not surprising that endogenous nuclear McIdas signal under these conditions is difficult to detect. In experiments where McIdas was overexpressed, we additionally used pre-extraction prior to methanol fixation to remove soluble proteins and better visualize stable cellular structures, such as the centrosome (Figure 1C and Supplementary Figure 1C & D). Under these conditions, we were able to detect McIdas in both the nucleus and the centrosome of U2OS cells (Figures 2A & E, 4C & E, 7A and Supplementary Figures 1C & D, 4A, 8A), as well as in HeLa cells stably expressing GFP-McIdas (Figure 1C). The centrosomal signal can be masked by nucleoplasmic signal, especially when specific centrosomal markers are not used in parallel. It is thus not surprising that McIdas centrosomal localization had previously gone unnoticed. We have now included this point in the Results section, noting the fixation methods used and clarified why nuclear localization is not consistently observed under our current experimental conditions.

3. "Accumulation of Centrin was evident upon GFP-McIdas overexpression in U2OS and hTERT-RPE1 cells, suggestive of a possible role of McIdas in centriole formation (Supplementary Figure 1A & B, marked by an asterisk)." Mcidas is massively overexpressed and the centrin accumulation is not convincing as the foci are all different sizes.

Supplementary Figure 1C (previously Supplementary Figure 1A & B) displays maximal projection images. Individual centrin foci are apparent in control GFP-overexpressing cells but not when GFP-McIdas is overexpressed. This suggests that McIdas may play a role in the

accumulation of centrin, and thus in centriole numbers. This finding is consistent with the analysis presented in Figure 2A & B, where increased McIDAS expression directly induces centriole amplification. We acknowledge, however, that this sentence may have been unclear, and we have now revised it in the manuscript as follows: “Accumulation of centrin was evident upon GFP-McIDAS overexpression in U2OS and hTERT-RPE1 cells.”. While overexpression can lead to variability, the observed increase in centrin signal is consistent with McIDAS-induced centriole amplification under these conditions. We have also clarified in the corresponding figure legend that the images shown are maximal projections.

4. In many of the supplemental figures overexpression of McIDAS leads to numerous aggregates which is not addressed.

As explained above, we follow pre-extraction and methanol fixation, which removes soluble cellular proteins and can enhance the appearance of protein aggregates. We have now explained better our experimental conditions in the revised manuscript.

5. Throughout the paper the authors use the term expression when in fact that are discussion localization. These two words are not interchangeable and all uses of expression that refer to protein localization should be changed.

We corrected this as suggested, particularly in the text related to Figure 1 & Supplementary Figure 1.

6. The authors present evidence that in G1 McIDAS sometimes localizes to one centriole and sometimes both. This seems a bit odd and it would be nice to know the relative percentage of these events.

As demonstrated in Figure 1, McIDAS becomes detectable during the G1 phase of the cell cycle, with its expression increasing as the cell cycle progresses and dropping during mitosis (Figure 1B). In G1 phase, we observed centrosomes that exhibited either one or two McIDAS-dot staining. We have now properly integrated this finding into the new Supplementary Figure 1A,

which shows immunofluorescence images with McIDAS staining at one or both centrioles during G1. Moreover, we quantified the frequency of these events and added a new graph to Supplementary Figure 1B, showing that approximately half of the G1 centrosomes analyzed exhibited one McIDAS-dot staining, while the other half showed staining at both centrioles. Notably, in cases with two McIDAS-positive centrioles, one dot was typically brighter and larger than the other. This gradual and asymmetric accumulation pattern has also been observed for other centrosomal proteins, including SAS6 (Keller *et al*, 2014; Kim *et al*, 2013). We have shown that in G1 cells with one McIDAS centriolar focus, this corresponded to the mother centriole. We have explained these points in more detail in the revised manuscript.

7. While IPs with overexpressed tagged proteins is never optimal, at the very least the IP should be done in reciprocal experiments.

Following the reviewer's suggestion, we conducted reciprocal immunoprecipitation experiments in U2OS cells expressing myc-PLK4 under a tetracyclin-dependent promoter, transfected with GFP-McIDAS or GFP alone as a control. In Figure 5A, we demonstrated that myc-PLK4 is present in GFP-McIDAS immunoprecipitates. Additionally, in the new Supplementary Figure 5A, we show that GFP-McIDAS is also present in myc-PLK4 immunoprecipitates.

Furthermore, to strengthen this finding, we determined whether endogenous McIDAS can interact with endogenous PLK4. We collected total cell extracts from U2OS cells and immunoprecipitated the endogenous McIDAS protein. In the new Figure 5B, we validate the interaction between the endogenously expressed proteins (see also comments to Reviewer 3).

8. There is no reference to Figure 5C in the text.

In the previous version of Figure 5C, we presented a western blot analysis of the total cell extracts used in the *in vitro* kinase assays to examine whether McIDAS is phosphorylated by PLK4. We have now included this (now Figure 5D) in the main text, in the section discussing this *in vitro* kinase assay.

9. Figure 6E should be represented as relative intensity (to a control) which should allow for the data to be presented from all three replicates rather than a "representative" quantification which is essentially meaningless when we can see it in Figure 6C.

In the revised manuscript, the data in Figure 6E showing the quantification of the phosphorylation of WT McIdas or the phospho-dead mutant are now presented as the mean of three independent experiments, as suggested by the reviewer. The fold change was determined relative to GFP-McIdas WT to better describe the differences in phosphorylation efficiency between the WT and the 10A mutant.

10. Minor: In the future please label your figures for the reviewers. Without numbers it was difficult to know which figure was which (especially the supplemental)

Figures have been labeled based on the reviewer's suggestion.

Answer to Reviewer 2 comments:

McIdas is known to be a nuclear protein, and its connection to the cell cycle regulator Geminin has been well established. Upon binding to Geminin, it inhibits the association between Geminin and Cdt1 and modulates DNA replication licensing. In this study by Marina Arbi and co-workers, a new function of McIdas has been demonstrated. A new localization of McIdas has been identified at the middle region of centrioles and the distribution of the protein is also changed during the cell cycle. Overexpression or deletion of McIdas significantly impacts centriole biogenesis. The authors also detected the phosphorylation of McIdas by PLK4 at multiple sites. These PTM are critical for its role on centriole number control.

This is an interesting study that describes McIdas as new PLK4 target. Interaction of McIdas and PLK4 and phosphorylation of McIdas by PLK4 in vitro are convincingly demonstrated. The localization of McIdas with centrioles and the function of McIdas at centrioles is less convincing and needs further validation and clarification. Many "functional" experiments are based on "artificial" setups: HU block and McIdas or PLK4 overexpression. In addition, the authors deplete McIdas using siRNA and show a

defect in centriole duplication based on IF staining of Centrin and Sas6 as markers (Fig. 4), however, because of the missing rescue experiment this approach lacks an important control. Whether McIDAS phosphorylation by PLK4 has a role in centrosome duplication remains unclear since 10A and 10D mutants behave identical in Fig. 7. The 10-point mutations might inactivate the protein (do they localize to centrosomes? - unclear in Fig. 7A) or a phosphorylation cycle of McIDAS is necessary as discussed by the authors. Analyzing McIDAS mutant versions with less mutations is advisable.

In summary, following key improvements are needed:

- 1. Show McIDAS antibody specificity convincingly**
- 2. Perform siRNA rescue experiment and confirm centriole duplication defect**
- 3. Provide data for the function of McIDAS phosphorylation by PLK4.**

We would also like to thank Reviewer 2 for the helpful suggestions and comments, which we believe have significantly improved our manuscript.

In our revised manuscript, we now include additional data in Supplementary Figure 2 demonstrating the specificity of the McIDAS antibody and the efficiency of McIDAS depletion. This includes qPCR, western blotting, and immunofluorescence images. Additionally, we conducted a mutational analysis showing that McIDAS centrosomal localization is dependent on a nuclear export signal located at the C-terminus of the McIDAS coiled-coil domain. Mutation of this motif abolished McIDAS centrosomal localization, supporting the specificity of the signal.

We would also like to clarify the rationale behind examining McIDAS specific function in centriole duplication by performing the experiments in HU-treated cells. McIDAS loss has previously been shown to cause accumulation of cells in S phase, while its overexpression leads to abnormal multipolar spindles (Pefani et al., 2011). To investigate McIDAS' direct role in the centrosome cycle, we performed overexpression and depletion experiments in HU-treated cells to uncouple the DNA replication cycle from the centrosome cycle, focusing solely on McIDAS function in the centriole duplication cycle. This system has been extensively used to study the centrosome cycle independently from the DNA replication cycle (Tachibana *et al*, 2005; Tachibana & Nigg, 2006; Xu *et al*, 2017), as HU treatment blocks the DNA replication cycle and arrests cells in S-phase, but the centrosome duplication cycle proceeds. As cells are arrested in S-phase under all tested conditions, findings are not confounded by any cell cycle effects. We have now better explained the rationale of the HU experiments in the revised manuscript.

In addition to this system, we showed that McIDAS is also essential for centriole biogenesis in unperturbed cells (Figure 3).

Furthermore, we now include rescue experiments (updated Figure 2E & F and Supplementary Figure 8A & B) demonstrating that McIDAS WT, but not the phospho-dead mutant, can rescue the centriole duplication defect observed upon McIDAS loss.

Please find below our detailed, point-by-point response to all comments raised by Reviewer 2.

Concerns and suggestions:

1. The localization data would benefit from quantification (Fig. 1A).

In the revised manuscript, in Figure 1, we quantified the localization data to demonstrate how McIDAS protein levels at centrosomes change throughout the cell cycle. We measured McIDAS fluorescence intensity at centrosomes and plotted these values in a new graph in Figure 1B. The results in Figure 1B are consistent with the representative images in Figure 1A, showing that McIDAS is present at centrosomes as early as the G1 phase, with its expression increasing as the cell cycle progresses before dropping in the later stages of mitosis. Notably, during G1 phase, McIDAS can be detected at either one or both centrioles. We quantified the frequency of these events and included the data into the new Supplementary Figure 1A & B (see also our response to comment #6 from Reviewer 1).

2. I am concerned about Fig. S1E: McIDAS signal intensity at centrosomes was only mildly affected by siRNA depletion of McIDAS (images are not shown - why?). However, depletion works very well when analyzed by IB (Fig. S2). How does this fit together? This raises concerns about the specificity of the McIDAS antibody and the validity of this study. It is very important that the authors resolve this discrepancy and show convincingly that the centriole signals observed with the McIDAS antibody reflect the localization of this protein.

Previous work from our group verified and published the specificity of the McIDAS antibody used in this study, as well as the efficiency of the McIDAS siRNA_1 oligo in depleting McIDAS expression in cells (Pefani *et al.*, 2011).

To further address this concern, we performed additional experiments and added a new figure (Supplementary Figure 2) to demonstrate both the specificity of the McIDAS antibody for

centrosomal detection and the effectiveness of two different siRNA oligos in depleting McIDAS. For both siRNA oligos, we now provide qPCR and western blot data showing reduction of McIDAS mRNA and total protein levels, respectively. We also included the corresponding immunofluorescence images showing McIDAS signal intensity at centrosomes. While McIDAS protein, both total levels by western blotting (Supplementary Figure 2B & C and 2G & H) and at centrosomes by immunofluorescence (Supplementary Figure 2D & E and 2I & J), is consistently reduced upon siRNA treatment, it is not completely lost - likely due to partial knockdown efficiency. However, McIDAS depletion in the cell line stably expressing mycPLK4 under a tetracycline-dependent promoter with McIDAS_1 siRNA oligos (now Supplementary Figure 3A-C, previously Supplementary Figure 2B) appeared to be more efficient than in U2OS cells shown elsewhere, suggesting possible differences in knockdown efficiency between cell lines. We also analyzed lysates from cells overexpressing an HA-tagged McIDAS construct and confirmed that the antibody recognizes a band corresponding to the expected molecular weight.

The specificity of the phenotype is further supported by the data presented in Supplementary Figure 1D: HeLa cells stably expressing GFP-tagged McIDAS or GFP alone as a control were treated with the proteasome inhibitor MG132, resulting in the accumulation of McIDAS at centrosomes. McIDAS accumulation could be detected using both an antibody against GFP and the McIDAS antibody, further confirming the specificity of McIDAS antibody staining and its validity for detecting McIDAS centrosomal localization. Moreover, the new mutant analysis in Figure 4, aimed at characterizing how McIDAS is localized at centrosomes, further supports the specificity of McIDAS localization, as centrosomal localization is lost when NES2 is mutated. Taken together, these data support the specificity of the McIDAS antibody and the effectiveness of the siRNA oligos used and demonstrate that the centrosomal signal observed by immunofluorescence reflects endogenous McIDAS localization. We have included these points in the revised manuscript.

- 3. I am concerned about the image quality in Fig. 2E.**
- 4. On p. 6 the authors talk about that McIDAS expression is significantly increased at procentrioles during early mitosis (Fig. 2D). How can they make such a conclusion without quantification of the signal and statistics?**
- 5. Furthermore, there is a certain discrepancy between Fig. 2E and Fig. 2J. According to Fig. 2E, McIDAS in G2 is at the newly synthesized centriole close to Sas6 but not at the**

distal part of the mother centriole. Fig. 2J, S/G2 cells shows something different: McIdas is at the mother centriole and co-localizes with the entire daughter.

In the revised manuscript, we updated and combined the data showing McIdas dynamic localization at centrosomes during the cell cycle to present them more comprehensively and to address the concerns raised by the reviewers. In the updated Figure 1, we included the previously presented data showing that during part of the G1 phase, McIdas can be detected at only one of the two centrioles. To determine which centriole this is, we performed co-staining with Cep164, which marks the distal appendages of the mother centriole, and concluded that McIdas is initially detected at the mother centriole.

This is consistent with further analysis of McIdas sub-centrosomal localization using expansion microscopy (now shown in Figure 1H): during the G1 phase, McIdas can be found at the mother centriole or at both centrioles, and as cell cycle progresses, McIdas becomes prominently localized at the daughter centrioles, compared to the mother centrioles. This observation is also supported by the data now presented in Supplementary Figure 1E-G. As suggested by the reviewer, we plotted McIdas, Cep164, and centrin signal intensities relative to distance for both the mother and daughter centrioles. These plots consistently demonstrate that McIdas signal intensity is increased at daughter centrioles at this stage. The previous analysis (former Figure 2E) has now been removed.

6. Fig. 4: the authors should perform a rescue experiment with a siRNA resistant McIdas construct.

In the revised manuscript, we performed a rescue experiment using an siRNA resistant McIdas construct. In these experiments, McIdas was expressed in cells treated with McIdas siRNA to assess its ability to restore centriole amplification. As shown in the revised Figure 2E & F, the centriole overduplication phenotype was significantly rescued by transient expression of the RNAi-resistant GFP-McIdas, but not by GFP alone, in McIdas-depleted U2OS cells. These results demonstrate that McIdas is sufficient to rescue the centriole overduplication phenotype, confirming the specificity of the observed phenotype.

7. When McIdas siRNA was applied to different cell lines, apart from the conformation by qPCR (Fig. S2A, S2C, Fig. 3, Fig. 4), IB and IF should also be carried out to check the protein level and localization.

In our revised manuscript, we have included the new Supplementary Figure 2, which demonstrates the validation of the McIdas siRNA oligos used in U2OS cells throughout our analysis of McIdas function in centriole duplication (see also response to point 2 above). For both siRNA oligos, we now provide qPCR and western blot data showing reduction of McIdas mRNA and total protein levels, respectively. We also included the corresponding immunofluorescence images showing McIdas fluorescence intensity at centrosomes upon its depletion. In the western blot analysis, total protein extracts from cells overexpressing an HA-tagged McIdas construct were used to validate the expected molecular weight of McIdas. McIdas total protein levels are significantly reduced, although not completely lost, which is consistent with the immunofluorescence staining.

Additionally, we initially examined whether McIdas depletion also affects PLK4-mediated centriole amplification in U2OS cells inducibly overexpressing PLK4. In the new Supplementary Figure 3A-C, in addition to qPCR and western blot analyses conforming McIdas knockdown, we also included immunofluorescence images showing effective depletion of McIdas from centrosomes. We believe these updated data clearly demonstrate that McIdas localizes to centrosomes and is efficiently depleted by the siRNA oligos used in our study.

8. What will happen when McIdas is overexpressed? Would it induce amplification of centrioles?

McIdas was initially identified as a factor affecting normal cell cycle progression (Pefani *et al.*, 2011). Previous experiments from our group demonstrated that McIdas loss leads to the accumulation of cells in S phase, while its overexpression causes abnormal multipolar spindles. In this study, we aimed to investigate the direct role of McIdas in the centrosome cycle. To achieve this, we performed a centrosome duplication assay in which cells were treated with a high concentration of HU, allowing the uncoupling of the DNA replication cycle from the centrosome duplication cycle. Using this assay, we focused solely on McIdas function in the centriole duplication cycle and demonstrated that McIdas overexpression increases centriole amplification (Figure 2A & B).

This effect of McIDAS on the centriole duplication cycle was further supported by our new findings related to McIDAS localization at centrosomes via a newly characterized nuclear export signal (NES) in the McIDAS protein (Figure 4E & F). When this NES was mutated, McIDAS centrosomal localization was impaired, and McIDAS NES2 mutant could not induce centriole overduplication, in contrast to GFP-McIDAS WT or the NES1 mutant.

We also show that GFP-McIDAS overexpression in non-HU treated HeLa cells leads to increased centrin signals (Supplementary Figure 1C), which is consistent with inducing centrosome overduplication. However, we have not quantified number of centrioles in such asynchronous cells, due to possible confounding effects from cell cycle effects of McIDAS overexpression.

9. Fig 4A and 4C: Since High background was detected especially for SAS-6, at least another centrosome marker should be used to confirm the centriole numbers (or EM analysis).

As also suggested by Reviewer 1, in the updated Figure 3, we have included Cep170 and Cep192 as additional centrosomal markers for the quantification of SAS6 and centrin foci, respectively. In addition, we have added a description in the manuscript outlining the criteria used to define centrioles in our analysis. Foci were defined as centrioles only if they were positive for both markers. Finally, the results now represent the mean values from three independent experiments, rather than two as previously shown (see also our response to comment #1 from Reviewer 1).

The minor comments:

1. Start a new paragraph on p. 3: PLK4 (Polo-like kinase 4), STIL () ... Same on p.3: The expression levels and the interactions Same on p. 7: Given that PLK4 overexpression induces ... (by the way: genes are overexpressed, therefore PLK4 is italic). Same p. 9: Next, we mutated these 10 serine residues

We have implemented these changes in the respective paragraphs/pages based on the reviewer's recommendations.

2. A statistical analysis should be carried out to determine whether the differences observed are significant (Fig. S1D and F and Fig. S2C).

The statistical analysis comparing McIdas mRNA levels between control and McIdas-depleted cells has been performed for all graphs indicated by the reviewer. In the revised manuscript, these analyses are now included in the new Supplementary Figure 2A & F, as well as in the Supplementary Figure 3A.

3. p. 8: "highlighting that McIdas is important for centrosomal Sas6 recruitment." I would say: McIdas is important for PLK4 induced centrosome amplification.

We have revised this sentence according to the reviewer's suggestion.

4. Fig. 5E, Fig. 6E and Fig. S4E: it would be nice to show the average values from different repeats and show the standard deviation of each value.

The data in Figure 6E and Supplementary Figure 7E (previously Supplementary Figure 4E) are now presented as the mean values of three independent experiments with error bars indicating \pm SEM, as suggested by the reviewer. In Figure 5, we have removed the graph (previously Figure 5E), as the main point was to demonstrate that McIdas is phosphorylated at multiple phosphorylation sites, making quantification unnecessary.

5. The length and the thickness of the scale bar should be standardized (Fig. 2E and 2J)

The thickness of the scale bars has been standardized, and the length of the scale bars is indicated in the respective figure legend.

6. Fig 3B and 3D: under GFP overexpressing, about 34% of cells contain more than four centrioles due to HU treatment. In siRNA control cells, almost half of them had the

same phenotype. The authors should explain the different frequencies in both control experiments, which are supposed to be comparable.

The observed differences in the frequency of control cells with more than 4 centrioles arise from the different experimental conditions used in the overexpression and RNAi experiments. In the overexpression experiment, U2OS cells were cultured and treated with HU for a total of 72 hours. However, in the RNAi experiment, cells were transfected with siRNA oligos, and the HU-induced centriole amplification was initiated 5h after the second transfection, with cells treated with HU for a total of 48h. Each experimental condition was always compared to its corresponding control condition in every experiment. These points have been clarified in the respective figure legends.

7. Fig 5A: additional unlabeled bands appear in the IB. The authors should clarify the identity of these bands.

The additional bands observed in the immunoblot correspond to the heavy and light chains of the IgGs. This has now been indicated in the figure and mentioned in the corresponding figure legend.

8. Fig 5E: the title of y axis seems wrong; it should be 1-179.

This graph has been removed from our revised manuscript.

9. Fig S4C and S4E: the quantification method could be mentioned in the legend of this figure, because relatively high background signal could be observed from non-PLX4 treated samples.

The data in the new Supplementary Figure 7E (previously Supplementary Figure 4E) are now presented as the mean values of three independent experiments with error bars indicating \pm SEM, as suggested by the reviewers (see also comments to Reviewer 1 & 2 above). We have

now included in the corresponding figure legend the method used for quantification, as follows: quantification of McIdas WT or partial mutant phosphorylation was performed using Image Lab software. In each experiment, background signal in the control condition (without PLK4) was calculated and subtracted from the final intensity measurements.

10. Fig 7B: the authors should check the expression level of different constructs, i.e., wt, 10A and 10D. The constructs may have different transfection efficiency, different expression level or different stability in cells.

We initially showed that the phospho-dead mutant of McIdas (10A) failed to induce centriole amplification upon HU treatment, in contrast to the centriole overduplication phenotype observed upon GFP-McIdas WT overexpression. Interestingly, the phospho-mimetic mutant GFP-McIdas 10D also did not induce centriole amplification compared to GFP-McIdas WT-overexpressing cells (Figure 7A & B). As suggested by the reviewer, to investigate whether these phenotypic differences could be attributed to variations in expression levels among the different constructs used, we analyzed total protein extracts from these cells. In the new Figure 7C, we demonstrate that the expression levels of the constructs are comparable.

Additionally, in Supplementary Figure 7A, we show a western blot analysis of total lysates collected from HEK 293T cells expressing GFP-McIdas WT protein or the mutant forms 10A and 10D. No significant differences were observed in the expression levels among the different proteins, with only a slight increase observed in 10D.

Answer to Reviewer 3 comments:

Major shortcomings of the manuscript:

1. The first main claim of the study is that McIdas localizes to centrosomes, which the authors support by showing images of immunolabelling experiments and McIdas overexpression (Figure1). A key point to back this claim, is to demonstrate specific antibody detection of endogenous McIdas at the centrosome. Although the authors state they have confirmed this point by siRNA depletion of McIdas, the presented data isn't convincing. The first issue is that McIdas RNA levels are shown, but not protein levels by western blot. The second issue is that no images of centrosomes stained with McIdas antibody are

shown for this experiment, and the quantification only shows a slight reduction in McIDAS centrosomal signal. This leads to multiple important questions: is it that the siRNA does not deplete the protein? Or is the antibody staining centrosomes in a non-specific manner? How are the centrosomal levels quantified? The authors later claim that centrosomal McIDAS localization varies during the cell-cycle. It would therefore also be important to assess how McIDAS siRNA affects cell-cycle progression here. In regard of all these questions, it remains unconvincing that McIDAS is indeed a centriole-associated protein.

We agree that the concerns raised by the reviewer regarding the specificity of the McIDAS antibody and the efficiency of the siRNA oligos used for McIDAS depletion are important. To address these points, we have included a new figure in the revised manuscript (see also response to Reviewer 2, comments #2 & #7). In the updated Supplementary Figure 2, for both siRNA oligos used in this study, we now include not only the qPCR analysis showing McIDAS mRNA levels in control and McIDAS-depleted cells, but also western blot analysis of McIDAS protein levels and immunofluorescence images used to quantify McIDAS intensity at centrosomes. We also analyzed total protein extracts from cells overexpressing an HA-tagged McIDAS construct to confirm the expected molecular weight of McIDAS protein. Although McIDAS depletion is not complete, our combined data consistently demonstrate a reproducible reduction in both total McIDAS protein levels and centrosomal localization. Please note that data shown in Figure 1A, B, D, E, F, G and H concern the endogenous protein in U2OS and RPE1 cells.

Additional data supporting the specificity of our antibody and centrosomal McIDAS localization are provided in Figure 1C and Supplementary Figure 1C & D, where GFP-McIDAS can be detected at centrosomes in different cell lines, either stably expressed or exogenously overexpressed. Notably, treatment with the proteasome inhibitor MG132 in HeLa cells stably expressing GFP-tagged McIDAS blocked McIDAS degradation, leading to its accumulation at centrosomes. The signal was detected using both GFP and McIDAS antibodies and differs from that observed in control GFP-expressing cells. Finally, we would like to highlight the new findings from our mutant analysis shown in Figure 4, where McIDAS localization is significantly reduced when its NES is mutated. We believe that our data overall demonstrate the specificity of the McIDAS antibody and the effectiveness of the siRNA oligos used in the current study for McIDAS depletion.

Regarding the question of how McIDAS siRNA affects cell cycle progression, previous studies from our group originally identified McIDAS as a cell cycle regulator that binds to Geminin and inhibits its association with Cdt1 (Pefani *et al.*, 2011). It was shown that McIDAS depletion

affects normal cell cycle progression and leads to the accumulation of cells in S phase, as they are unable to enter mitosis or progress to the following G1 phase. Therefore, to examine McIDAS' role independently of its function in cell cycle regulation, we used a HU-induced centrosome duplication assay to uncouple DNA replication from centriole duplication. HU treatment arrests cells in S-phase and allows multiple rounds of centriole duplication. Under these conditions, we examined the effects of McIDAS depletion in U2OS cells. As shown in the updated Figure 2C & D, McIDAS depletion impaired centriole duplication in HU-treated cells. Consistently, we showed that McIDAS is also required for canonical centriole duplication, as demonstrated in Figure 3, where both centriole numbers and SAS6 recruitment are reduced upon McIDAS loss.

2. The second main claim of the manuscript is that McIDAS regulates centriole duplication. The authors aim to demonstrate this in different contexts, but the presented data is not convincing. To claim McIDAS is involved in centriole biogenesis, the main point is to demonstrate that changing McIDAS levels affects centriole numbers in unperturbed cycling cells. This is the point the authors aim to make in Figure 4. They deplete McIDAS by siRNA, and then count centrioles specifically in EdU positive S-phase cells, using Centrin or SAS6 to detect all centrioles or nascent centrioles respectively. In both cases, the immunostainings show lots of what seems to be non-specific signal, raising the question of how the authors decided to count a spot as a centriole or not. Colocalisation of multiple centriolar markers would be necessary to confirm centriolar staining.

This is an important point that has been raised by all the reviewers and we have now addressed it in the updated Figure 3 (see also responses to comments #1 and #9, raised by Reviewer 1 and 2, respectively). We have verified our initial results by performing double immunostainings on centrosomes from control and McIDAS-depleted asynchronous U2OS cells, using antibodies against centrin and Cep192 (Figure 3A & B) or SAS6 and Cep170 (Figure 3C & D). Throughout our analysis, we defined foci as centrioles only if they were positive for both centriole markers. Our data consistently demonstrate that McIDAS depletion affects centriole duplication in cells. We have now better explained these points in the revised manuscript.

3. Additionally, if McIDAS does regulate centriole biogenesis, the expectation is that centrosome numbers should be affected in siRNA depletion experiments or McIDAS OE experiments. No data throughout the paper convincingly shows this, although the authors do state that cell lines over-expressing GFP- McIDAS may have more centrioles, but without providing convincing images or quantifications (Supplementary Figure 1A/B). As McIDAS involvement in centriole biogenesis in unperturbed cells seems hard to demonstrate, the authors aimed to prove its involvement in two different contexts in which centrioles are overduplicated: HU-induced S-phase lengthening, and PLK4 over-expression. Here, many points weaken the claim that McIDAS affects centriole over-duplication, which is tested by either over-expressing or down-regulating McIDAS. First, for both mechanisms of over-duplication, cell-cycle progression could have a strong impact on levels of over-duplication. While the authors state (but without showing the data) that McIDAS over-expression does not affect cell-cycle progression upon HU treatment, the impact on cell-cycle progression of McIDAS siRNA is never assessed, making conclusions hard to take.

As mentioned above, previous studies from our group demonstrated that McIDAS loss leads to accumulation of cells in S phase, while its overexpression causes abnormal multipolar spindles (Pefani *et al.*, 2011). We would like to clarify that the reason of performing the experiments using HU-induced centriole overduplication was not due to inability to observe centriole duplication effects in unperturbed cells, but rather to investigate McIDAS' role independently of its function in cell cycle regulation. The HU-induced centrosome duplication assay enables uncoupling of the DNA replication cycle from the centriole duplication cycle, as HU treatment arrests cells in S phase and allows multiple rounds of centriole duplication. This system has been extensively used in the centrosome field to assess effects on the centrosome cycle, independently from other cell cycle effects (Tachibana *et al.*, 2005; Tachibana & Nigg, 2006; Xu *et al.*, 2017), and it is precisely to make sure that our findings are not confounded by any cell cycle-related effects. Under these conditions, we observed that McIDAS depletion in U2OS cells significantly impaired centriole duplication.

Consistently, we also demonstrate that McIDAS is required for canonical centriole duplication, as shown in the updated Figure 3. There, additional centriole markers were used and both centriole numbers and SAS6 recruitment are significantly reduced upon McIDAS depletion.

Moreover, to address concerns about cell cycle progression under HU treatment, a FACS analysis was performed on the samples used for GFP-McIDAS overexpression in HU-treated

U2OS cells (Figure 2A). The data show that indeed, as expected, cells arrest in early S phase upon HU treatment and there are no significant changes in cell cycle profiles between conditions, indicating that the observed increase in centriole number upon GFP-McIdas overexpression is cell cycle-independent.

U2OS cells were treated with 4 mM hydroxyurea (HU) and 24 hours later were transfected with vectors expressing GFP-tagged McIdas or GFP alone as a control. Cells were analyzed by flow cytometry for DNA content.

4. Second, the effect-size of the perturbations (either centriole gain or loss), is always very mild, and sometimes even milder than control perturbations. As an example in the HU induced overduplication, the levels of over-duplication are already different between GFP over-expression and Luciferase siRNA which are two control perturbations.

We have already addressed this concern (see also our response to question #6 raised by Reviewer 2). The observed differences in the frequency of control cells with more than 4 centrioles arise from the different experimental conditions used in the overexpression and siRNA experiments. Each experimental condition was always compared to its corresponding control condition within the same experiment.

5. Finally, the choice of showing in some cases the frequency of cells with a given centriole number, and in some others the number of centrioles per cells, only makes conclusions even harder to take. Explain here why Together, the presented data does not convincingly support that McIdas is involved in centriole biogenesis regulation.

Throughout the manuscript, we calculated centriole numbers and grouped them into categories: 2 to 4 centrioles, which represents the expected range during a normal centriole cycle, and 0 to 1 or more than 4 centrioles, which reflect conditions that perturb centriole numbers. In our revised manuscript, we have now consistently presented our data using this categorization, except for the experiments examining the effects in U2OS cells inducibly overexpressing PLK4. In these specific experiments, centriole numbers are significantly increased upon PLK4 overexpression, and we aimed to include this information and how it changes upon McIDAS knockdown (updated Supplementary Figure 3G & I). Additionally, in Figure 7B, we calculated centriole numbers in conditions where McIDAS WT or phospho-mutants were overexpressed, as the effects on centriole numbers are most clearly observed in cells with more than 4 centrioles.

6. Additionally, other experiments presented in this manuscript also raise doubts as to the soundness of the conclusions taken. In particular, the study of McIDAS centriolar localization dynamics is weakened by images that show varying quality of immunostainings.

In our manuscript, we present all the data for McIDAS localization at centrioles and its changes throughout the cell cycle in the updated Figure 1 and Supplementary Figure 1. We now show this in a more comprehensive way, including quantification of the localization data in Figure 1A to demonstrate how McIDAS centrosomal levels change during the cell cycle. This confirms that McIDAS is present at centrosomes, from the G1 phase, with its expression increasing as the cell cycle progresses, before dropping in the later stages of mitosis. We further analyzed the centrosomal localization pattern of McIDAS during G1 phase, where it can be detected at either one or both centrioles. We quantified the frequency of these events and included the data in the updated Supplementary Figure 1A & B. We believe that our revised figures clearly demonstrate that McIDAS localizes to the centre of centrioles (Figure 1D-G) and our expansion microscopy images (Figure 1H), together with the analysis of mitotic cells (Supplementary Figure 1E-G), show that McIDAS is initially recruited to the mother centriole, showing a gradual enrichment at newly synthesized centrioles as the cell cycle progresses. In the revised manuscript, we provide further evidence of the specific localization of McIDAS to centrosomes by showing that a mutant form of McIDAS which lacks a nuclear export signal fails to correctly localize to

centrosomes (Figure 4C & D). This McIdas NES2 mutant was unable to induce centriole overduplication, in contrast to McIdas WT or the McIdas NES1 mutant (Figure 4E & F).

7. Furthermore, the study of PLK4 and McIdas molecular interactions also lack conclusiveness and physiological relevance. Although the immunoprecipitation of over-expressed tagged proteins suggest possible interaction, and in vitro phosphorylation assays suggest possible phosphorylation, these observations would require additional controls (IP in the other direction ? Effect of the commercially available specific Plk4 kinase inhibitor centrinone ?).

As also suggested by Reviewer 1 (see also our response to comment #7 from Reviewer 1), in our revised manuscript we have conducted reciprocal immunoprecipitation experiments in U2OS cells expressing myc-PLK4 under a tetracycline-dependent promoter, transfected with GFP-McIdas or GFP as a control. In Figure 5A, we previously showed the presence of myc-PLK4 in GFP-McIdas immunoprecipitates. In the new Supplementary Figure 5A, we show the reciprocal IP, where GFP-McIdas is also present in myc-PLK4 immunoprecipitates, confirming the specificity of the interaction between McIdas and PLK4. To further corroborate this interaction, we also examined whether endogenous McIdas interacts with endogenous PLK4. Total cell extracts from U2OS cells were immunoprecipitated for endogenous McIdas and as shown in the new Figure 5B, we confirmed the interaction between the endogenously expressed proteins.

Since McIdas interacts with and is phosphorylated by PLK4, we also tested whether the kinase activity of PLK4 is required for this interaction, using centrinone - a specific PLK4 kinase inhibitor. In the new Supplementary Figure 5B & C, we show that centrinone had no effect on McIdas-PLK4 binding, indicating that McIdas binding to PLK4 is independent of PLK4 kinase activity.

8. Furthermore, the physiological relevance of this phosphorylation suggested in the title and in the main text claims from Figure 7, are again not backed by convincing data showing centriole biogenesis modifications.

To further support the importance of McIdas phosphorylation by PLK4 during centriole duplication, we performed additional rescue experiments to determine the specificity of the phospho-dead mutant of McIdas. We previously showed that GFP-McIdas WT protein was sufficient to rescue the centriole overduplication phenotype observed in McIdas-depleted cells (Figure 2E & F). We depleted again McIdas in HU-treated U2OS cells and overexpressed the GFP-McIdas 10A construct. As shown in the updated Supplementary Figure 8A & B, the GFP-McIdas 10A phospho-dead mutant was defective in rescuing the centriole overduplication phenotype, compared to GFP-McIdas WT. This result now shows the specificity of the McIdas phospho-dead mutant in centriole duplication phenotype and strengthens our conclusion that the phosphorylation of McIdas by PLK4 is important for centriole biogenesis.

References

- Arbi M, Pefani DE, Kyrousi C, Lalioti ME, Kalogeropoulou A, Papanastasiou AD, Taraviras S, Lygerou Z (2016) GemC1 controls multiciliogenesis in the airway epithelium. *EMBO Rep* 17: 400-413
- Keller D, Orpinell M, Olivier N, Wachsmuth M, Mahen R, Wyss R, Hachet V, Ellenberg J, Manley S, Gonczy P (2014) Mechanisms of HsSAS-6 assembly promoting centriole formation in human cells. *J Cell Biol* 204: 697-712
- Kim TS, Park JE, Shukla A, Choi S, Murugan RN, Lee JH, Ahn M, Rhee K, Bang JK, Kim BY *et al* (2013) Hierarchical recruitment of Plk4 and regulation of centriole biogenesis by two centrosomal scaffolds, Cep192 and Cep152. *Proc Natl Acad Sci U S A* 110: E4849-4857
- Pefani DE, Dimaki M, Spella M, Karantzelis N, Mitsiki E, Kyrousi C, Symeonidou IE, Perrakis A, Taraviras S, Lygerou Z (2011) Idas, a novel phylogenetically conserved geminin-related protein, binds to geminin and is required for cell cycle progression. *J Biol Chem* 286: 23234-23246
- Tachibana KE, Gonzalez MA, Guarguaglini G, Nigg EA, Laskey RA (2005) Depletion of licensing inhibitor geminin causes centrosome overduplication and mitotic defects. *EMBO Rep* 6: 1052-1057
- Tachibana KE, Nigg EA (2006) Geminin regulates multiple steps of the chromosome inheritance cycle. *Cell Cycle* 5: 151-154
- Xu X, Huang S, Zhang B, Huang F, Chi W, Fu J, Wang G, Li S, Jiang Q, Zhang C (2017) DNA replication licensing factor Cdc6 and Plk4 kinase antagonistically regulate centrosome duplication via Sas-6. *Nat Commun* 8: 15164

Dear Dr. Arbi,

Thank you for your patience while your revised ms was re-reviewed at EMBO reports. We have now received the comments from 2 referees that are pasted below, as well as cross-comments from referee 2 on referee 1's report, which are attached to this email.

I would like to invite you to address all final comments as suggested by both referees. Please send us a point-by-point response to all final comments with your final ms submission.

Also a few editorial requests will need to be addressed before we can proceed with the official acceptance of your manuscript:

- Please upload your ms text file in Word format.
- Please reduce the number of keywords to 5.
- Please add a Data Availability Statement providing links and access to data generated in this study and deposited in public databases to the end of the Methods. The specific URL for the PRIDE dataset PXD037043 must be provided in the data availability statement.
- Please correct the conflict of interest subheading to "Disclosure and Competing Interests Statement".
- Please remove the author credits from the ms file. All credits need to be entered during online ms submission.
- Please delete DATA NOT SHOWN on page 6 and page 8, as per journal policy. Either show the data or re-write.
- Please co-submit with your final ms a fully completed author checklist, which you can download from our author guidelines <<https://www.embopress.org/page/journal/14693178/authorguide>>. The completed author checklist will also be part of your transparent peer-review file.
- All main figures need to be uploaded as individual production quality Figure files with their legends in the ms. Your suppl. figures could be either EV figures, then they also need to be uploaded as individual figure files with their legends in the ms following the main figure legends. EV figures are embedded in the main ms text online. Otherwise, the suppl. figures could stay in the PDF that would need to be titled Appendix and each legend should follow its figure (and be removed from the ms). The Appendix file would also need a title page with a ToC and page numbers. The correct nomenclature of the Appendix figures should be Appendix Figure S1, etc. The callouts in the ms need to be updated accordingly. "Supplementary" should not be used - the callouts for the suppl. figures need updating to either Figure EV1, etc. or Appendix Figure S1. etc. For more information please check our guide to authors online.
- All Materials and Methods need to be described in the main text. The Methods section should include a separate file called Reagents and Tools Table (listing key reagents, experimental models, software and relevant equipment and including their sources and relevant identifiers) and a Methods and Protocols section in which authors should describe their methods using a step-by-step protocol format with bullet points, to facilitate the adoption of the methodologies across labs. More information on how to adhere to this format as well as downloadable templates (.docx) for the Reagents and Tools Table can be found in our author guidelines: < <https://www.embopress.org/page/journal/14693178/authorguide#manuscriptpreparation>>.
- The manuscript sections should be in the following order: Title page - Abstract & Keywords - Introduction - Results - Discussion - Methods - Data Availability - Acknowledgments - Disclosure Statement & Competing Interests - References - Figure Legends - (Main Tables with legends if applicable) - Expanded View Figure Legends (if applicable).
- Our routine image analysis of to be accepted ms detected a reuse between Figure 2E and Appendix Fig 8A that is not listed in the figure legends. Please clarify and if the same images are used this must be stated in the legend.
- We need the entire figure set at a higher resolution. Currently most Cells and blots are over pixelated with no background definition. Please send us High resolution individual figure files.

Figure Legends - Comments

- Please note that the exact p values are not provided in the legends of figures 1B, 2B, D, F, H; 3B, D; 4D, F; 6E, 7B, D, E, F, G. Please provide exact p-values as reasonable.
- Please note that the scale bar needs to be defined for figures 1D, E.

I would like to suggest to modify the last sentence of the abstract to:

This novel, direct role of MclDas in centriole duplication connects its functions in cell cycle regulation and multiciliogenesis.

EMBO press papers are accompanied online by A) a short (1-2 sentences) summary of the findings and their significance, B) 2-3 bullet points highlighting key results and C) a synopsis image that is exactly 550 pixels wide and 200-600 pixels high (the height is variable). The synopsis image should provide a sketch of the major findings, like a graphical abstract. Please note that text needs to be readable at the final size. Please send us this information along with the final manuscript.

Referee #1:

The manuscript from Arbi et al. has been significantly improved with the addition of some important new experiments. However, there are a lot of changes making this essentially a new submission. There are still some considerable issues that I think need to be addressed prior to publications.

Figure 4. I have seen many images of gamma tubulin myself and in papers and I have never seen the kind of nuclear membrane localization that is shown in Fig 4. Please explain.

FigS2I-J. This was a point of contention in the first round and it is still concerning. It is not that strong of a knockdown in localization and the Sas6 looks equally knocked down. This experiment should have an internal control that stays the same. It is simply not very convincing and central to interpretations.

The quantification of centriole numbers is still very difficult to understand. At almost every image of centriole quantification I have no idea how many centrioles there are. The authors state that they have added two color staining to verify which foci are centrioles but then they go on to say that they count CP110 foci which don't costain with Cep135. While I appreciate that these are likely procentrioles and won't have Cep135 that defeats the purpose of costaining. In Figure 2E there are 50 CP110 foci but they only count the ones near Cep135? The reader must put a lot of faith into how this is quantified. Right now it is not very convincing.

Full gels should be shown for IP/Westerns in the supplemental data.

The point of the centrinone experiment was not to test whether it affected the interaction but to test if it effects the phosphorylation. This key experiment was not done.

"Consistently, incubation of GFP-MclDas WT- or 10A- or 10D-containing extracts with lambda protein phosphatase released these proteins from their phosphate groups, suggesting that phosphorylation is responsible for the observed differences in their mobility (Supplementary Figure 7A)." I am very confused by the interpretation of this result. They claim that the shift in mobility is due to the phosphorylation since the band gets smaller with phosphatase treatment. However it shifts equally with the Alanine switch which should be non phosphorylatable and with the phosphomimetic 10D which should not be affected by the phosphatase. Please explain your interpretation.

Minor:

FigS1D. There are no boxes for the inlay pictures. Not sure what is being looked at.

Referee #2:

The authors have added the siRNA resistant control and it is now convincing that MclDas associated with centrioles, is important for centriole duplication and is a PLK4 substrate. The authors have identified a NES in MclDas that is important for centriole

localization. However, the manuscript does not show how Mcdas phosphorylated by PLK4 promotes centriole duplication on a molecular level.

Taken together, the authors have improved the manuscript in the revised version to a level that justified publication in EMBO Reports.

Point-by-point response

Please find below a point-by-point response to the comments raised by the referees, as well as to the editorail requests. We would like to thank both the referees and the editor for their additional comments and suggestions.

Answer to Referee #1 comments:

Referee #1:

The manuscript from Arbi et al. has been significantly improved with the addition of some important new experiments. However, there are lot of changes making this essentially a new submission. There are still some considerable issues that I think need to be addressed prior to publications.

1. Figure 4. I have seen many images of gamma tubulin myself and in papers and I have never seen the kind of nuclear membrane localization that is shown in Fig 4. Please explain.

In Figure 4C & 4E, nuclear boundaries are outlined with a white line and white boxes mark regions showing GFP-McIdas (green), Centrin (red) and γ -tubulin (grey) staining as higher-magnification images on the right (Figure 4C) or below (Figure 4E). We have now clarified this in the corresponding figure legend, as well as in all other figures where such insets are used, by adding: *“Nuclear boundaries are outlined and white boxes indicate regions shown as higher-magnification images.”*

2. FigS2I-J. This was a point of contention in the first round and it is still concerning. It is not that strong of a knockdown in localization and the Sas6 looks equally knocked down. This experiment should have an internal control that stays the same. It is simply not very convincing and central to interpretations.

To further address the concern regarding the efficiency of McIdas depletion, we used two independent siRNA oligos throughout the study and assessed knockdown at multiple levels:

McIdas mRNA, total protein and centrosomal signal. As shown in Supplementary Figure 2, for both siRNAs, McIdas mRNA levels (Supplementary Figure 2A & F) and total protein levels (Supplementary Figure 2B-C & 2G-H) are consistently reduced, although not completely lost, likely due to partial knockdown efficiency. Importantly, centrosomal McIdas intensity (Supplementary Figure 2D-E & 2I-J) is also significantly reduced, although residual signal may persist, as often observed for centrosome-associated proteins due to local protein stability or retention.

Regarding SAS6, its reduced centrosomal signal upon McIdas depletion (as also shown in Figure 3C & D) is consistent with impaired centriole biogenesis rather than a technical artifact. Because McIdas depletion directly affects the centriole duplication pathway, downstream components involved in daughter centriole formation (including SAS6) are expected to be reduced as a biological consequence of the perturbation. Nevertheless, the use of two independent siRNAs, together with consistent reductions in McIdas mRNA, total protein, centrosomal signal, and daughter centriole formation, supports the conclusion that McIdas depletion reproducibly impairs centriole biogenesis.

3. The quantification of centriole numbers is still very difficult to understand. At almost every image of centriole quantification I have no idea how many centrioles there are. The authors state that they have added two color staining to verify which foci are centrioles but then they go onto to say that they count CP110 foci which don't costain with Cep135. While I appreciate that these are likely procentrioles and won't have Cep135 that defeats the purpose of costaining. In Figure 2E there are 50 CP110 foci but they only count the ones near Cep135? The reader must put a lot of faith into how this is quantified. Right now it is not very convincing.

Throughout our study, the criteria used to define foci as centrioles were positivity for two centriole markers. In most cases, one proximal (e.g. CEP135) and one distal (e.g. CP110 or Centrin) marker were used. Although these markers do not fully co-localize due to their known differential distribution along the centriole, they are sufficiently close to clearly distinguish specific centrosomal dots from non-specific cytoplasmic signal, as shown for example in Figure 2E. Centriole numbers were subsequently quantified using the distal markers (CP110 or Centrin), which provide sufficient spatial resolution to reliably count individual centrioles across all z-stacks. This is a widely used approach in the centrosome field. While this was

already described in the manuscript where centriole numbers were quantified, for clarity and consistency we have now explicitly stated the marker used for centriole quantification in all relevant figure legends (Figure 2, 3, 4 & 7 and Supplementary Figure 3 & 8).

4. Full gels should be shown for IP/Westerns in the supplemental data.

Full uncropped gels are now included in the source data files.

5. The point of the centrinone experiment was not to test whether it affected the interaction but to test if it effects the phosphorylation. This key experiment was not done.

Centrinone was used in our study to test whether PLK4 kinase activity is required for the McIDAS-PLK4 interaction. As shown in Supplementary Figure 5C, centrinone treatment did not disrupt McIDAS-PLK4 binding, indicating that the interaction does not depend on PLK4 kinase activity under the tested conditions. With respect to phosphorylation, we assessed PLK4-dependent phosphorylation using *in vitro* kinase assays with a phospho-dead McIDAS mutant. As shown in Figure 6C & E, phosphorylation of McIDAS by PLK4 was markedly reduced when the phospho-dead mutant was used, supporting the specificity of the PLK4-mediated phosphorylation events identified by mass spectrometry. While we did not directly assay centrinone-sensitive phosphorylation of McIDAS, we believe that these experiments demonstrate that PLK4 kinase activity is required for McIDAS phosphorylation at the identified sites.

6. "Consistently, incubation of GFP-McIDAS WT- or 10A- or 10D-containing extracts with lambda protein phosphatase released these proteins from their phosphate groups, suggesting that phosphorylation is responsible for the observed differences in their mobility (Supplementary Figure 7A)." I am very confused by the interpretation of this result. They claim that the shift in mobility is due to the phosphorylation since the band gets smaller with phosphatase treatment. However it shifts equally with the Alanine switch which should be non phosphorylatable and with the phosphomimetic 10D which should not be affected by the phosphatase. Please explain your interpretation.

As shown in Supplementary Figure 7A, lambda phosphatase treatment causes a faster mobility shift not only for GFP-McIdas WT, but also for the 10A and 10D mutants. We agree with the referee that this point was not sufficiently clarified in the original manuscript. We have now revised the text to clarify that the mobility shift observed for both 10A and 10D indicates that these mutants still carry additional phosphorylation sites (beyond the ten identified PLK4-specific residues). This suggests that McIdas is phosphorylated at multiple sites, including PLK4-dependent and additional PLK4-independent sites, which together likely contribute to the mobility differences.

Minor:

1. FigS1D. There are no boxes for the inlay pictures. Not sure what is being looked at.

White boxes have now been added to Supplementary Figure 1D (subpanel: DMSO, HeLa GFP-McIdas), corresponding to the higher-magnification images shown for McIdas staining.

Referee #2:

The authors have added the siRNA resistant control and it is now convincing that McIdas associated with centrioles, is important for centriole duplication and is a PLK4 substrate. The authors have identified a NES in McIdas that it is important for centriole localization. However, the manuscript does not show how McIdas phosphorylated by PLK4 promotes centriole duplication on a molecular level.

Taken together, the authors have improved the manuscript in the revised version to a level that justified publication in EMBO Reports.

We would like to thank Referee 2 for the constructive feedback and suggestions throughout the revision process. This work is the first to demonstrate a direct role for McIdas in centriole duplication, linking its previously described functions in the cell cycle and multiciliogenesis. We show that McIdas is present at centrioles from early G1 (similar to PLK4), interacts with and is phosphorylated by PLK4 and that its depletion impairs PLK4-induced centriole

amplification. Together, these findings support a model in which McIdas cooperates with PLK4 during the initiation of centriole duplication.

We agree with the referee that further studies will be required to define the underlying molecular mechanism. We have therefore added a paragraph to the Discussion outlining mechanistic insights and highlighting key open questions raised by our findings, as follows: *“Together, these findings support a model in which McIdas cooperates with PLK4 during the initiation of centriole duplication. It will be interesting to determine whether McIdas is required for the recruitment or stabilization of PLK4 at the proximal end of the mother centriole together with the CEP192 and CEP63/CEP152 complex (Brown et al, 2013; Cizmecioglu et al, 2010; Dzhindzhev et al, 2014; Hatch et al, 2010; Kim et al, 2013; Sir et al, 2011; Sonnen et al, 2013). Notably, Cep63, which interacts with CEP152 to recruit PLK4 during centriole biogenesis in cycling cells, has a paralogue in multiciliated cells, DEUPI, which is a transcriptional target of McIdas (Ma et al, 2014; Zhao et al, 2013). Furthermore, our data show that McIdas-PLK4 binding is independent of PLK4-mediated phosphorylation, raising the interesting possibility that these two events contribute to McIdas function through distinct mechanisms. Whether McIdas phosphorylation directly facilitates procentriole assembly and how these processes are coordinated temporally during the centriole duplication cycle, remain important questions for future investigation.”*

Answer to editorial requests:

1. Please upload your ms text file in Word format.

The manuscript text file has been submitted in Word format

2. Please reduce the number of keywords to 5.

The number of keywords have been reduced to 5.

3. Please add a Data Availability Statement providing links and access to data generated in this study and deposited in public databases to the end of the Methods. The specific

URL for the PRIDE dataset PXD037043 must be provided in the data availability statement.

A Data Availability Statement has now been included at the end of the Methods section. The datasets generated in this study are available as follows:

LC-MS/MS proteomics data: PRIDE PXD037043

(<https://www.ebi.ac.uk/pride/archive/projects/PXD037043>)

4. Please correct the conflict of interest subheading to "Disclosure and Competing Interests Statement".

This has been corrected as suggested.

5. Please remove the author credits from the ms file. All credits need to be entered during online ms submission.

The author credits have been removed from the manuscript file.

6. Please delete DATA NOT SHOWN on page 6 and page 8, as per journal policy. Either show the data or re-write.

The phrase "data not show" has been removed from pages 6 and 8.

7. Please co-submit with your final ms a fully completed author checklist, which you can download from our author guidelines

<<https://www.embopress.org/page/journal/14693178/authorguide>>. The completed author checklist will also be part of your transparent peer-review file.

A fully completed author checklist has been co-submitted with the final manuscript.

8. All main figures need to be uploaded as individual production quality Figure files with their legends in the ms. Your suppl. figures could be either EV figures, then they also need

to be uploaded as individual figure files with their legends in the ms following the main figure legends. EV figures are embedded in the main ms text online.

Otherwise, the suppl. figures could stay in the PDF that would need to be titled Appendix and each legend should follow its figure (and be removed from the ms). The Appendix file would also need a title page with a ToC and page numbers. The correct nomenclature of the Appendix figures should be Appendix Figure S1, etc. The callouts in the ms need to be updated accordingly. "Supplementary" should not be used - the callouts for the suppl. figures need updating to either Figure EV1, etc. or Appendix Figure S1. etc. For more information please check our guide to authors online.

The Supplementary Figures are now presented as Expanded View Figures, uploaded as individual figure files, and their legends are included in the main manuscript following the main figure legends.

**9. All Materials and Methods need to be described in the main text. The Methods section should include a separate file called Reagents and Tools Table (listing key reagents, experimental models, software and relevant equipment and including their sources and relevant identifiers) and a Methods and Protocols section in which authors should describe their methods using a step-by-step protocol format with bullet points, to facilitate the adoption of the methodologies across labs. More information on how to adhere to this format as well as downloadable templates (.docx) for the Reagents and Tools Table can be found in our author guidelines: <
[The previously titled Materials and Methods section has been revised and is now presented as the Methods section. This includes a detailed Reagents and Tools Table \(uploaded as a separate file\) and a Methods and Protocol section.](https://www.embopress.org/page/journal/14693178/authorguide#manuscriptpreparation&>;

>>> Please note that it is EMBO Reports policy for the transcript of the editorial process (containing referee reports and your response letter) to be published as an online supplement to each paper. If you do NOT want this, you will need to inform the Editorial Office via email immediately. More information is available here: <https://link.springer.com/partners/embo-press/editorial-policies#Peer%20review>